# Fermionic Non-Invertible Symmetries in (1+1)d: Gapped and Gapless Phases, Transitions, and Symmetry TFTs

Lakshya Bhardwaj,[1] Kansei Inamura,[2] and Apoorv Tiwari[3]

[1]*Mathematical Institute, University of Oxford, Woodstock Road, Oxford, OX2 6GG, United Kingdom*
[2]*Institute for Solid State Physics, University of Tokyo, Kashiwa, Chiba 277-8581, Japan*
[3]*Niels Bohr International Academy, Niels Bohr Institute,*
*University of Copenhagen, Blegdamsvej 17, DK-2100, Copenhagen, Denmark*

We study fermionic non-invertible symmetries in (1+1)d, which are generalized global symmetries that mix fermion parity symmetry with other invertible and non-invertible internal symmetries. Such symmetries are described by fermionic fusion supercategories, which are fusion $\pi$-supercategories with a choice of fermion parity. The aim of this paper is to flesh out the categorical Landau paradigm for fermionic symmetries. We use the formalism of Symmetry Topological Field Theory (SymTFT) to study possible gapped and gapless phases for such symmetries, along with possible deformations between these phases, which are organized into a Hasse phase diagram. The phases can be characterized in terms of sets of condensed, confined and deconfined generalized symmetry charges, reminiscent of notions familiar from superconductivity. Many of the gapless phases also serve as phase transitions between gapped phases. The associated fermionic conformal field theories (CFTs) can be obtained by performing generalized fermionic Kennedy-Tasaki (KT) transformations on bosonic CFTs describing simpler transitions. The fermionic non-invertible symmetries along with their charges and phases discussed here can be obtained from those of bosonic non-invertible symmetries via fermionization or Jordan-Wigner transformation, which is discussed in detail.

## CONTENTS

## I. INTRODUCTION

Over the past few years, there has been a lot of interest in understanding generalized global symmetry structures [1] involving non-invertible or categorical symmetries [2–157], see e.g. [36, 93, 100, 158–160] for recent reviews. Various applications of these symmetries have been proposed. This paper explores applications of generalized symmetries to understand possible gapped and gapless phases of quantum systems. We focus on finite non-invertible symmetries in 1+1 spacetime dimension. In [2–6], this program has been carried out for bosonic non-invertible symmetries, which are described by fusion categories [10–12, 161]. Here we extend their analysis to fermionic non-invertible symmetries, which are non-invertible symmetries involving a $\mathbb{Z}_2^f$ fermion parity symmetry generated by $(-1)^F$ (which is taken to be non-anomalous). As we will discuss, such symmetries are described by fusion $\pi$-supercategories with a choice of fermion parity.

We will see that the structure of phases for both fermionic and bosonic symmetries is similar, and can be summarized as follows. Let $\mathcal{S}$ be a bosonic or fermionic symmetry in (1+1)d. Local operators may form various representations of $\mathcal{S}$,[1] which we refer to as charges of $\mathcal{S}$. The set of charges is labeled as $\mathcal{Z}(\mathcal{S})$. Two operators with charges $q_1, q_2 \in \mathcal{Z}(\mathcal{S})$ may be mutually non-local, i.e. their correlation function may shift when one of them is transported in a circle around the other. The mutual non-locality is a function only of $q_1$ and $q_2$, independent

---

[1] More precisely, at least in the bosonic case, point-like operators form representations of the tube algebra of $\mathcal{S}$ [2, 7, 92].

of the precise identity of the operators involved. Now, a gapped/gapless phase $\mathcal{P}$ with symmetry $\mathcal{S}$ is characterized by a subset $Q_\mathcal{P} \subset \mathcal{Z}(\mathcal{S})$, which are the charges of operators condensed in the phase $\mathcal{P}$. Note that the charges in $Q_\mathcal{P}$ must all be mutually local. Just like in the Meissner effect, any operator with a charge mutually non-local with some charge in $Q_\mathcal{P}$ gets confined and is invisible in the infrared (IR). On the other hand, any charge not in $Q_\mathcal{P}$ and mutually local with all charges in $Q_\mathcal{P}$ remains deconfined.

Let us label the set of deconfined charges in phase $\mathcal{P}$ as $D_\mathcal{P}$. The deconfined charges are the charges that must be carried by gapless excitations in phase $\mathcal{P}$. As long as we perform $\mathcal{S}$-symmetric deformations on a system in phase $\mathcal{P}$ without condensing or uncondensing charges, we cannot gap out all excitations with a charge $q \in D_\mathcal{P}$. Consequently, a gapped phase is a phase $\mathcal{P}$ having no deconfined charges $D_\mathcal{P} = 0$. In such a phase all gapless excitations are uncharged and hence can be gapped out via $\mathcal{S}$-symmetric deformations.

Starting from an arbitrary phase $\mathcal{P}$, we can further condense a subset $Q_{\mathcal{P}',\mathcal{P}} \subset D_\mathcal{P}$ of charges to deform the phase $\mathcal{P}$ to a phase $\mathcal{P}'$ for which the condensed charges are $Q_{\mathcal{P}'} = Q_\mathcal{P} \cup Q_{\mathcal{P}',\mathcal{P}}$ and the set of deconfined charges $D_{\mathcal{P}'}$ is the subset of $D_\mathcal{P}$ which is mutually local with $Q_{\mathcal{P}',\mathcal{P}}$. In this way, all possible gapped/gapless phases with symmetry $\mathcal{S}$ and possible $\mathcal{S}$-preserving deformations between them are captured [5, 162], which can be arranged into a phase diagram that is partially ordered, or in other words, a Hasse diagram [6]. The partial order describes various possible sequential condensation patterns of charges.

Some of the gapless phases may be viewed as certain transitions between other gapped and gapless phases. For example, if a gapless phase can be deformed to two gapped phases, then it can act as a transition between the two gapped phases. As discussed in [5] for the bosonic case, some of these transitions can be described by degenerate gapless states (or universes) in the IR, with each gapless state described by either a well-known conformal field theory (CFT) like Ising CFT or 3-state Potts CFT, or a CFT obtained by gauging a discrete symmetry of these CFTs. Here in the fermionic case we will find that some such transitions can be described similarly by degenerate gapless states, with each gapless state described either by a well-known bosonic CFT like Ising or 3-state Potts, or a fermionic CFT like the Majorana CFT, or finite gaugings/bosonizations/fermionizations thereof.

In this way, the essential input for understanding phases and transitions for $\mathcal{S}$ symmetry boils down to the set of charges for $\mathcal{S}$ and their mutual (non-)locality properties. For a bosonic symmetry $\mathcal{S}$, it was understood in [2, 7, 92] that the corresponding charges can be identified with anyons of a bosonic $(2+1)$d topological field theory (TFT) $\mathfrak{Z}(\mathcal{S})$ associated to $\mathcal{S}$, which is referred to as the Symmetry TFT (SymTFT) for $\mathcal{S}$ [163]. The mutual (non-)locality properties of the charges are encoded in the mutual braidings of these anyons. Mathemati-

cally the anyons and their braidings are described by a modular tensor category (MTC) $\mathcal{Z}(\mathcal{S})$ referred to as the Drinfeld center of $\mathcal{S}$. The possible sets $Q_\mathcal{P}$ of condensed charges are encapsulated mathematically as possible condensable algebras $\mathcal{A}_\mathcal{P}$ in the category $\mathcal{Z}(\mathcal{S})$ [162], using which gapped and gapless phases $\mathcal{P}$ for symmetry $\mathcal{S}$ were studied in [3–6]. A deformation from a phase $\mathcal{P}_1$ to a phase $\mathcal{P}_2$ is possible only if the first condensable algebra is a subalgebra of the second $\mathcal{A}_{\mathcal{P}_1} \subset \mathcal{A}_{\mathcal{P}_2}$. Here, a deformation is an $\mathcal{S}$-symmetric perturbation that does not close the energy gap of $\mathcal{S}$-charges, i.e., gapped degrees of freedom with $\mathcal{S}$-charges remain gapped under deformations. A gapped phase $\mathcal{P}$ corresponds to a condensable algebra $\mathcal{A}_\mathcal{P}$ that is Lagrangian (or maximal).

In this paper, we observe that the charges for a fermionic symmetry $\mathcal{S} = \mathcal{S}_f$ are also encoded in a bosonic $(2+1)$d TFT. In other words, just as for bosonic symmetries, the SymTFT $\mathfrak{Z}(\mathcal{S}_f)$ for a fermionic symmetry $\mathcal{S}_f$ is also a bosonic TFT. This has been proposed for general non-invertible symmetries in [164]. For invertible fermionic symmetries, this has been proposed and studied in [8]. Returning to the general non-invertible case, the anyons of the SymTFT may be viewed as forming the Drinfeld center $\mathcal{Z}(\mathcal{S}_f)$ for the fusion supercategory $\mathcal{S}_f$.[2] The gapped and gapless phases for fermionic symmetry $\mathcal{S}_f$ are thus characterized by condensable algebras in $\mathcal{Z}(\mathcal{S}_f)$, and deformations between the phases are characterized by the inclusion of condensable algebras.

A quick argument for the SymTFT of a fermionic symmetry being bosonic is as follows. Since we take the fermion parity to be non-anomalous, we can always bosonize (or perform GSO orbifolding) using it. This converts the fermionic (non-invertible) symmetry $\mathcal{S}_f$ to a bosonic (non-invertible) symmetry $\mathcal{S}$, for which the SymTFT $\mathfrak{Z}(\mathcal{S})$ is bosonic. Now recall that gauging of bosonic symmetries does not modify SymTFT [8]. Since bosonization is analogous to gauging [165–169], we do not expect it to modify the SymTFT either, hence leading to the conclusion $\mathfrak{Z}(\mathcal{S}_f) = \mathfrak{Z}(\mathcal{S})$.

A consequence of the above is that all the gapless and gapped phases, and transitions, for a fermionic symmetry $\mathcal{S}_f$ are fermionizations of the gapless and gapped phases, and transitions, for the bosonized symmetry $\mathcal{S}$. That being said, the fermionization/bosonization procedure may not be straightforward to implement. The fermionization procedure for general non-invertible symmetries was discussed in detail in [9]. Here we extend that analysis to describe how the fermionization procedure is implemented on the phases.

The paper is organized as follows. In Section II, we discuss the general structures of fermionic non-invertible symmetries in 1+1d, which are described by fusion $\pi$-

---

[2] Precisely, the Drinfeld center $\mathcal{Z}(\mathcal{S}_f)$ contains a transparent fermion in addition to anyons of the SymTFT. In other words, $\mathcal{Z}(\mathcal{S}_f)$ describes the bosonic SymTFT stacked with a trivial fermionic TFT, see Section III for more details.

supercategories with a choice of fermion parity. We will see that the choice of a fermion parity in a fermionic $\pi$-superfusion category is affected by stacking a 2d fermionic invertible TFT known as the Arf TFT. We also discuss how the fermionic symmetries are obtained as fermionizations of bosonic non-invertible symmetries described by ordinary fusion categories. In Section III, we develop a general framework of the symmetry TFT for fermionic non-invertible symmetries. It turns out that the symmetry TFT for a fermionic non-invertible symmetry is a bosonic TFT that admits a fermionic topological boundary. Such fermionic boundaries are characterized by fermionic Lagrangian algebras up to stacking with the Arf TFT. In Section IV, we discuss generalized charges of fermionic non-invertible symmetries. As in the case of bosonic symmetries, generalized charges are labeled by anyons of the symmetry TFT. Their structure can be obtained by fermionizing the charges for bosonic symmetries. In Section V, we apply the symmetry TFT construction to study fermionic gapped and gapless phases with fermionic symmetries. We also discuss phase transitions between these fermionic phases. We describe how these phases and transitions can be obtained by fermionizing phases and transitions for bosonic symmetries.

**Note added**: While nearing completion of this paper, we became aware of a related work [149], which studies the symmetry TFTs for 1+1d fermionic systems with finite invertible (i.e., group-like) symmetries. We also learned that there is another related paper [170], which has some overlaps with our results. The symmetry TFTs for fermionic symmetries are also discussed from the point of view of 4d Crane-Yetter TFT and the 4-category of braided fusion categories in [171, 172].

## II. FERMIONIC SYMMETRIES IN 1+1 DIMENSIONS

In this section, we describe the general structure of finite non-invertible symmetries of 1+1d fermionic systems. In particular, we clarify that such a symmetry is described by a fusion $\pi$-supercategory with a choice of fermion parity. This is analogous to the fact that finite non-invertible symmetries of 1+1d bosonic systems are described by fusion categories [10–12, 161]. The bosonic and fermionic symmetries are related by operations of fermionization and bosonization. An interesting feature of fermionic symmetries is that they appear in pairs related by stacking with the 2d fermionic invertible TFT known as the Arf TFT.

### A. Fusion Supercategories

Let us begin with a brief review of fusion supercategories. For more details of fusion supercategories, we refer the reader to [173–176]. See also [9, 17, 49, 177, 178]

for these symmetries in the context of two-dimensional quantum field theories.

A fusion supercategory $\mathcal{S}$ consists of topological lines and topological point-like defects between topological lines. A trivial (i.e., invisible or identity) line is denoted by 1, while a trivial point-like defect on a topological line $X$ is denoted by $1_X$. Topological point-like defects between topological lines $X$ and $Y$ form a finite dimensional $\mathbb{C}$-vector space,[3] which is denoted by $\mathrm{Hom}(X, Y)$. The vector space $\mathrm{Hom}(X, Y)$ is equipped with a $\mathbb{Z}_2$-grading that represents the fermion parity of point-like defects. The $\mathbb{Z}_2$-grading of $f \in \mathrm{Hom}(X, Y)$ is denoted by $|f|$, which is 0 if $f$ is bosonic and 1 if $f$ is fermionic. This $\mathbb{Z}_2$-grading is compatible with the fusion of point-like defects in the sense that

$$|g \circ f| = |f| + |g| \mod 2 \qquad \text{(II.1)}$$

for any homogeneous $f \in \mathrm{Hom}(X, Y)$ and $g \in \mathrm{Hom}(Y, Z)$.[4] Note that $1_X$ is bosonic for all $X$. A point-like defect $f \in \mathrm{Hom}(X, Y)$ is called an isomorphism if there exists a point-like defect $f^{-1} \in \mathrm{Hom}(Y, X)$ such that $f^{-1} \circ f = 1_X$ and $f \circ f^{-1} = 1_Y$. We note that an isomorphism $f$ can be either bosonic or fermionic. When there is an isomorphism between topological lines $X$ and $Y$, we say that $X$ and $Y$ are isomorphic to each other and write $X \cong Y$. In the study of fermionic symmetries, it is often useful to track objects only up to bosonic isomorphism.

The fusion of topological lines $X$ and $Y$ defines a tensor product $X \otimes Y$. This tensor product is compatible with the $\mathbb{Z}_2$-grading of point-like defects living on topological lines. More specifically, the $\mathbb{Z}_2$-grading of a point-like defect $f \otimes g \in \mathrm{Hom}(X \otimes X', Y \otimes Y')$ is given by

$$|f \otimes g| = |f| + |g| \mod 2 \qquad \text{(II.2)}$$

for any homogeneous $f \in \mathrm{Hom}(X, X')$ and $g \in \mathrm{Hom}(Y, Y')$. Here, the tensor product $f \otimes g$ of point-like defects is defined by

$$f \otimes g := (f \otimes 1_{Y'}) \circ (1_X \otimes g) = (-1)^{|f||g|}(1_{X'} \otimes g) \circ (f \otimes 1_Y), \qquad \text{(II.3)}$$

where the sign $(-1)^{|f||g|}$ encodes the anti-commutation relation of fermionic point-like defects. Pictorially, this fermionic anti-commutation relation can be depicted as

$$f \uparrow \quad \uparrow g \; = \; (-1)^{|f||g|} g \uparrow \quad \uparrow f. \qquad \text{(II.4)}$$
$$\quad X \quad Y \qquad\qquad\quad X \quad Y$$

Any topological line in a fusion supercategory can be decomposed into a finite direct sum of indecomposable topological lines. The space $\mathrm{Hom}(x, x)$ of topological

---

[3] In this paper, all vector spaces are $\mathbb{C}$-vector spaces.
[4] A point-like defect is said to be homogeneous if it has a definite $\mathbb{Z}_2$-grading.

point-like defects on an indecomposable topological line $x$ is isomorphic to either $\mathbb{C}^{1|0}$ or $\mathbb{C}^{1|1}$, where $\mathbb{C}^{p|q}$ denotes the $(p + q)$-dimensional super vector space of superdimension $(p, q)$. An indecomposable topological line $x$ is called an $m$-type line if $\mathrm{Hom}(x, x)$ is isomorphic to $\mathbb{C}^{1|0}$ [175], i.e. if $x$ cannot have a fermionic point-like defect on it. On the other hand, an indecomposable topological line $x$ is called a $q$-type line if $\mathrm{Hom}(x, x)$ is isomorphic to $\mathbb{C}^{1|1}$ [175], i.e. if $x$ can have a fermionic point-like defect on it. We note that the trivial line $1$ is $m$-type.

The fusion of indecomposable topological lines $x$ and $y$ can be decomposed as

$$x \otimes y \cong \bigoplus_z N_{xy}^z z, \quad N_{xy}^z \in \mathbb{Z}_{\geq 0}, \qquad \text{(II.5)}$$

where the direct sum on the right-hand side is taken over (representatives of isomorphism classes of) indecomposable topological lines. Here, the number of (isomorphism classes of) indecomposable topological lines is supposed to be finite. The equation (II.5) is known as the fusion rules. The fusion $x \otimes y$ of topological lines $x$ and $y$ is also denoted as $xy$ in subsequent sections. In this paper, we write fusion rules up to bosonic isomorphisms unless otherwise stated. Furthermore, bosonic isomorphisms will also be written simply as an equality by abuse of notation.

A network of topological lines can be locally deformed by using the $F$-move defined by

$$\vcenter{\hbox{\includegraphics{fmove_left}}} = \sum_v \sum_{\rho,\sigma} (F_w^{xyz})_{(u;\mu,\nu),(v;\rho,\sigma)} \vcenter{\hbox{\includegraphics{fmove_right}}},$$
$$\text{(II.6)}$$

where $(F_w^{xyz})_{(u;\mu,\nu),(v;\rho,\sigma)}$ is a complex number called the $F$-symbol. The first summation on the right-hand side of eq. (II.6) is taken over representatives of isomorphism classes of simple objects, while the second summation is taken over bases of the spaces of topological point-like defects. Precisely, the left-hand side of eq. (II.6) is an element of a vector space

$$V_w^{xyz} \cong \bigoplus_u \mathrm{Hom}(u, x \otimes y) \otimes_{\mathrm{Hom}(u,u)} \mathrm{Hom}(w, u \otimes z),$$
$$\text{(II.7)}$$

while the right-hand side of eq. (II.6) is an element of

$$V_w^{xyz} \cong \bigoplus_v \mathrm{Hom}(w, x \otimes v) \otimes_{\mathrm{Hom}(v,v)} \mathrm{Hom}(v, y \otimes z). \quad \text{(II.8)}$$

The tensor product over $\mathrm{Hom}(u, u)$ and $\mathrm{Hom}(v, v)$ is implicit in the diagrams, see [175, Section 8] for more details.[5] The $F$-symbols have to satisfy the consistency condition known as the fermionic pentagon identity [174–176], which is the fermionic analogue of the ordinary pentagon identity, where the fermionic anti-commutation relation (II.4) is taken into account.

---

[5] This complication does not play any role in this paper.

Another important property of a topological line is that it can be bent freely to the left and right. This implies that the pair $(x, x^*)$ of a topological line $x$ and its orientation reversal $x^*$ is equipped with bosonic topological point-like defects called the evaluation and coevaluation morphisms, which are denoted by $\mathrm{ev}_x^{L/R}$ and $\mathrm{coev}_x^{L/R}$ in the following diagrams:

$$\text{(II.9)}$$

These point-like defects enable us to define the quantum dimension of each topological line as follows:

$$\dim(x) = \vcenter{\hbox{\includegraphics{loop_left}}} x = x \vcenter{\hbox{\includegraphics{loop_right}}}. \qquad \text{(II.10)}$$

The quantum dimension is a topological point-like defect on the trivial line $1$, which can be canonically identified with a complex number because we have $\mathrm{Hom}(1, 1) = \mathbb{C}$. In this paper, we suppose that the left and right quantum dimensions agree with each other as in eq. (II.10), i.e., we suppose that a fusion supercategory is spherical.

We note that a topological defect may or may not be isomorphic to its orientation reversal. A topological defect $x$ is said to be self-dual if it is isomorphic to its orientation reversal $x^*$. When $x$ is a self-dual $m$-type topological line, the isomorphism $x \cong x^*$ is unique up to scalar multiplication. This isomorphism can be either bosonic or fermionic. An example of a fermionic symmetry that has a self-dual $m$-type line $x$ with a fermionic isomorphism $x \cong x^*$ is a $\mathbb{Z}_2 \times \mathbb{Z}_2^f$ symmetry with a Gu-Wen anomaly, which we will encounter in the next subsection.

## B. Fermionic Fusion Supercategories

Any fermionic system has a $\mathbb{Z}_2$ symmetry called a fermion parity symmetry $\mathbb{Z}_2^f$, which we suppose to be non-anomalous in this paper. In addition, any fermionic system in 1+1d has another "$\mathbb{Z}_2$ symmetry" denoted by $\mathbb{Z}_2^\pi$, which is generated by a 0+1d fermionic invertible topological field theory (i.e., a quantum mechanical system).[6] On the other hand, a fusion supercategory introduced in the previous subsection does not necessarily contain both $\mathbb{Z}_2^f$ and $\mathbb{Z}_2^\pi$ subgroups. This implies that not every fusion supercategory can describe the symmetry of a fermionic system in 1+1d. Fusion supercategories realized as symmetries of 1+1d fermionic systems will be

---

[6] In most of the literature, $\mathbb{Z}_2^\pi$ is not regarded as the symmetry of a 1+1d fermionic system. This is presumably because the action of $\mathbb{Z}_2^\pi$ is almost trivial. Nevertheless, it turns out that $\mathbb{Z}_2^\pi$ plays a crucial role in the description of fermionic symmetries. We will further comment on this point later.

called fermionic fusion supercategories or fermionic symmetries in this paper. In what follows, instead of giving a precise definition of the fermionic fusion supercategory, we discuss some basic structures that every fermionic symmetry should have. See [171, 172] for a more precise definition of the fermionic fusion supercategory.

As mentioned above, a fermionic fusion supercategory contains a $\mathbb{Z}_2^\pi$ subgroup generated by a 0+1d fermionic invertible TFT. Here, we recall that 0+1d fermionic invertible TFTs are classified by $\mathbb{Z}_2$: the trivial class is given by a bosonic state, while the non-trivial class is given by a fermionic state. The generator $\pi$ of $\mathbb{Z}_2^\pi$ can be regarded as the worldline of a local fermion in 0+1d, which implies that a $\pi$ line can have a (topological) fermionic endpoint $\mathcal{O}_\pi \in \text{Hom}(1, \pi)$. This endpoint $\mathcal{O}_\pi$ defines a fermionic isomorphism between the trivial line $1$ and the $\pi$ line. If we fuse a $\pi$ line with an indecomposable topological line $x$, we obtain another indecomposable topological line $\pi x := \pi \otimes x$ that is isomorphic to $x$ via a fermionic isomorphism $\mathcal{O}_\pi \otimes 1_x$. When $x$ is $m$-type, this is the unique (up to scalar) isomorphism between $x$ and $\pi x$. On the other hand, when $x$ is $q$-type, there is also a bosonic isomorphism $\mathcal{O}_\pi \otimes f_x$, where $f_x \in \text{Hom}(x, x)$ is a fermionic isomorphism from $x$ to itself. In particular, an indecomposable topological line $x$ is $q$-type if and only if there is a bosonic point-like defect (acting as a bosonic isomorphism) between $x$ and $\pi x$. A fusion supercategory that is equipped with a $\pi$ line is called a fusion $\pi$-supercategory [173].

In a fusion $\pi$-supercategory, the quantum dimension of any q-type object $x$ satisfies $\dim(x) \geq \sqrt{2}$. This is because when $x$ is q-type, a fermionic automorphism of $x \otimes x^*$ implies that $x \otimes x^*$ contains $1$ and $\pi$ at the same time, meaning that $\dim(x)^2 \geq \dim(1 \oplus \pi) = 2$. On the other hand, the quantum dimension of any m-type object $y$ satisfies $\dim(y) \geq 1$ as in the case of bosonic symmetries. Here, we supposed that the quantum dimensions are non-negative real numbers, which is satisfied if the fusion (super)category is unitary.

A fermionic fusion supercategory is a fusion $\pi$-supercategory together with a choice of a fermion parity line $(-1)^F$.[7] In this paper, we suppose that the fermion parity symmetry generated by $(-1)^F$ is non-anomalous, meaning that the $F$-symbols involving only $(-1)^F$ and $1$ are all trivial. In particular, the isomorphism between $(-1)^F \otimes (-1)^F$ and $1$ has to be bosonic because otherwise the $F$-symbols cannot be trivial due to the fermionic anti-commutation relation. We note that the choice of a fermion parity line $(-1)^F$ is not unique for a given fusion $\pi$-supercategory. Specifically, if $(-1)^F$ is eligible to be a fermion parity line, $\pi(-1)^F$ is equally eligible to be a

fermion parity line. As we will see in Section II C, different choices of a fermion parity line give rise to symmetries of different fermionic systems.

The fermion parity line $(-1)^F$ and the $\pi$ line has a canonical junction $J$ between them. This junction is specified by the condition that $(-1)^F$ acts as $-1$ on the fermionic endpoint $\mathcal{O}_\pi$:

$$\pi \xrightarrow{\quad} J \underset{(-1)^F}{\overset{\mathcal{O}_\pi}{\bigcirc}} = (-1) \; \pi \xrightarrow{\quad} \mathcal{O}_\pi. \qquad \text{(II.11)}$$

More explicitly, the canonical junction $J$ can be written as

$$\underset{(-1)^F \quad \pi}{\overset{J}{\diagup\diagdown}} = (-1) \; \underset{(-1)^F \quad \pi}{\overset{\mathcal{O}_\pi \quad \mathcal{O}_\pi^{-1}}{\diagup\diagdown}}. \qquad \text{(II.12)}$$

The action of the $\pi$ line on the $(-1)^F$-twisted sector is also defined by using the same canonical junction as follows:

$$(-1)^F \xrightarrow{\quad} J^{-1} \overset{\pi}{\underset{}{\bigcirc}} \mathcal{O} = (-1) \; (-1)^F \xrightarrow{\quad} \mathcal{O}. \qquad \text{(II.13)}$$

The above equation shows that the $\pi$ line acts as $-1$ on point-like operators in the $(-1)^F$-twisted sector. On the other hand, the $\pi$ line acts as $+1$ on local or untwisted sector operators. Equivalently, $\pi$ acts as $-1$ on the Ramond (R) sector, while it acts as $+1$ on the Neveu-Schwarz (NS) sector. This is because point-like operators living at the end of $(-1)^F$ correspond to states in the R sector, while local operators correspond to states in the NS sector.[8] The $\pi$ line thus measures the spin structure. In later sections, point-like operators living at the end of $(-1)^F$ are sometimes called local operators in the R sector.

### C. Stacking with Arf TFT

As already discussed briefly above, given a fermionic fusion supercategory $\mathcal{S}_f$ with a choice of fermion parity, we obtain another fermionic fusion supercategory $\widetilde{\mathcal{S}}_f$ whose underlying fusion $\pi$-supercategory is the same as $\mathcal{S}_f$ but the choice of fermion parity is modified by $\pi$

$$(-1)^F_{\widetilde{\mathcal{S}}_f} = \pi(-1)^F_{\mathcal{S}_f}. \qquad \text{(II.14)}$$

Note that $\mathcal{S}_f$ and $\widetilde{\mathcal{S}}_f$ may be equivalent fermionic symmetries, meaning that there may be a supertensor autoequivalence of the fusion $\pi$-supercategory underlying $\mathcal{S}_f$ that maps $(-1)^F_{\mathcal{S}_f}$ to $\pi(-1)^F_{\mathcal{S}_f}$.

---

[7] The choice of a fermion parity line is not arbitrary. For example, in the case of invertible symmetries, $(-1)^F$ should generate a central $\mathbb{Z}_2^f$ subgroup. Even in the case of more general non-invertible symmetries, the fermion parity subgroup $\mathbb{Z}_2^f$ has to be central in an appropriate sense [171, 172].

[8] The spin structure induced on a small circle around the end of $(-1)^F$ is R (i.e., non-bounding), while the spin structure induced on a small circle around a local operator is NS (i.e., bounding).

Given a (1+1)d fermionic system $\mathfrak{T}_f$ with $\mathcal{S}_f$ symmetry, there exists a closely related system $\widetilde{\mathfrak{T}}_f$ which carries $\widetilde{\mathcal{S}}_f$ symmetry. $\widetilde{\mathfrak{T}}_f$ is obtained by stacking $\mathfrak{T}_f$ with the Arf TFT

$$\widetilde{\mathfrak{T}}_f = \mathrm{Arf} \boxtimes \mathfrak{T}_f. \tag{II.15}$$

Let us recall that the Arf TFT is an invertible 2d TFT carrying fermionic parity symmetry generated by[9]

$$(-1)^F_{\mathrm{Arf}} = \pi. \tag{II.16}$$

After stacking the two systems together we modify the $(-1)^F$ to be the diagonal of the $(-1)^F$ symmetries of the two systems, which results in the change of fermionic symmetry from $\mathcal{S}_f$ to $\widetilde{\mathcal{S}}_f$.

We thus say, as a figure of speech, that the fermionic symmetry $\widetilde{\mathcal{S}}_f$ is obtained from the fermionic symmetry $\mathcal{S}_f$ by stacking with the Arf TFT and write

$$\widetilde{\mathcal{S}}_f = \mathcal{S}_f \boxtimes \mathrm{Arf}. \tag{II.17}$$

### D. Fermionization and Bosonization

In general, a fermionic fusion supercategory is obtained by the fermionization of a bosonic (i.e., ordinary) fusion category that contains a non-anomalous $\mathbb{Z}_2$ subgroup. This is because any fermionic system with a non-anomalous $\mathbb{Z}_2^f$ symmetry can be obtained by the fermionization of its bosonization [167–169], which has a dual non-anomalous $\mathbb{Z}_2$ symmetry.

Let us discuss this in more detail as it plays a crucial role in the rest of the paper. Consider a bosonic theory $\mathfrak{T}$ in (1+1)d with a non-anomalous $\mathbb{Z}_2$ symmetry. Let us call the topological line operator implementing the $\mathbb{Z}_2$ symmetry by $P$. To fermionize this system, we first regard the bosonic system $\mathfrak{T}$ trivially as a fermionic system and stack the Arf TFT on top of it. The combined system is now

$$\mathrm{Arf} \boxtimes \mathfrak{T}. \tag{II.18}$$

We now gauge the $\mathbb{Z}_2$ symmetry generated by the topological line operator

$$(-1)^F_{\mathrm{Arf}} P = \pi P \tag{II.19}$$

of the combined system, which we label $\mathbb{Z}_2^{\pi P}$. The fermionization $\mathfrak{T}_f$ of $\mathfrak{T}$ is defined as the theory obtained after this gauging

$$\mathfrak{T}_f := \frac{\mathrm{Arf} \boxtimes \mathfrak{T}}{\mathbb{Z}_2^{\pi P}} \tag{II.20}$$

---

[9] In fact, the only indecomposable topological lines (upto bosonic isomorphisms) in the Arf TFT are 1 and $\pi$.

with the fermionic parity symmetry implemented by

$$(-1)^F = \pi\eta, \tag{II.21}$$

where $\eta = \widehat{\pi P}$ is the topological line operator implementing the dual $\mathbb{Z}_2$ symmetry of the gauged theory. This procedure implementing

$$\mathfrak{T} \longrightarrow \mathfrak{T}_f \tag{II.22}$$

is referred to as **fermionization** in this paper. Understanding fermionization as the condensation of an algebra object $1 \oplus \pi P$ was first proposed in [17, Appendix G]. This fermionization procedure can be regarded as the Jordan-Wigner transformation in a generalized sense. Indeed, just as the traditional Jordan-Wigner transformation [179], our fermionization maps a trivial bosonic TFT with an unbroken $\mathbb{Z}_2$ symmetry to a trivial fermionic TFT, while it maps a bosonic TFT with a spontaneously broken $\mathbb{Z}_2$ symmetry to the Arf TFT. The inverse procedure implementing

$$\mathfrak{T}_f \longrightarrow \mathfrak{T} \tag{II.23}$$

is referred to as **bosonization**.

When the original bosonic system $\mathfrak{T}$ has a fusion category symmetry $\mathcal{S}$, its fermionization $\mathfrak{T}_f$ has the corresponding fusion supercategory symmetry, which we denote by $\mathcal{S}_f$. The fermionic fusion supercategory $\mathcal{S}_f$ depends on the choice of a $\mathbb{Z}_2$ subgroup of $\mathcal{S}$. The general relation between $\mathcal{S}$ and $\mathcal{S}_f$ was studied in [9].

Let us describe how the fermionization acts on various kinds of local operators in $\mathfrak{T}$: [167–169]

1. Consider an uncharged local operator $\mathcal{O}$ of $\mathfrak{T}$ in the untwisted sector of $\mathbb{Z}_2$ symmetry. With respect to $\mathbb{Z}_2^{\pi P}$ symmetry of $\mathrm{Arf} \boxtimes \mathfrak{T}$, the operator $\mathcal{O}$ is also uncharged and untwisted. Under a $\mathbb{Z}_2$ gauging, an untwisted uncharged operator goes to an untwisted uncharged operator for the dual $\mathbb{Z}_2$ symmetry. Thus, after gauging of $\mathbb{Z}_2^{\pi P}$, the operator $\mathcal{O}$ is uncharged under $\eta$ and is in the untwisted sector of $\eta$ symmetry. Equivalently, $\mathcal{O}$ is uncharged under $(-1)^F = \pi\eta$ and is in the untwisted sector of $(-1)^F$ symmetry. In other words, $\mathcal{O}$ becomes a bosonic operator in the NS sector of the fermionization $\mathfrak{T}_f$.

2. Consider a charged local operator $\mathcal{O}_e$ of $\mathfrak{T}$ in the untwisted sector of $\mathbb{Z}_2$ symmetry. With respect to $\mathbb{Z}_2^{\pi P}$ symmetry of $\mathrm{Arf} \boxtimes \mathfrak{T}$, the operator $\mathcal{O}_e$ is also charged and untwisted. Under a $\mathbb{Z}_2$ gauging, an untwisted charged operator goes to an uncharged operator in the twisted sector for the dual $\mathbb{Z}_2$ symmetry, i.e. it is an operator uncharged under dual $\mathbb{Z}_2$ symmetry but attached to the topological line operator $\widehat{P}$ generating the dual $\mathbb{Z}_2$ symmetry. Thus, after gauging of $\mathbb{Z}_2^{\pi P}$, the operator $\mathcal{O}_e$ is uncharged under $\eta$ and is attached to $\eta$ line. Equivalently, $\mathcal{O}_e$

is a bosonic operator attached to $\eta$ line. Let us convert it into an operator

$$\mathcal{O}_e^\pi := \mathcal{O}_e \otimes \mathcal{O}_\pi \qquad (\text{II}.24)$$

obtained by fusing $\mathcal{O}_e$ with the topological local operator $\mathcal{O}_\pi$, which let us recall is the canonical fermionic operator arising at the end of $\pi$ line.

$$\mathcal{O}_e^\pi = \left. \begin{array}{ccc} \bullet\,\mathcal{O}_e & \bullet\,\mathcal{O}_\pi & \bullet\,\mathcal{O}_e^\pi \\ \Big| & \Big| & \Big| \\ \eta & \pi & (-1)^F \end{array} \right. = \qquad (\text{II}.25)$$

The operator $\mathcal{O}_e^\pi$ is attached to $\pi\eta = (-1)^F$ line and will be referred to as the fermionization of the operator $\mathcal{O}_e$ of $\mathfrak{T}$. In other words, the fermionization converts $\mathcal{O}_e$ into a fermionic operator in the R sector.

3. Consider an uncharged local operator $\mathcal{O}_m$ of $\mathfrak{T}$ in the twisted sector of $\mathbb{Z}_2$ symmetry. This can be converted into an operator

$$\mathcal{O}_{m,\pi} := \mathcal{O}_m \otimes \mathcal{O}_\pi \qquad (\text{II}.26)$$

of the Arf $\boxtimes \mathfrak{T}$ theory, which is attached to the $\pi P$ line and is charged under $\pi P$ since $\mathcal{O}_\pi$ is charged under $(-1)_{\text{Arf}}^F = \pi$ of the Arf factor. Under a $\mathbb{Z}_2$ gauging, a twisted charged operator goes to a twisted charged operator for the dual $\mathbb{Z}_2$ symmetry. Thus, after gauging of $\mathbb{Z}_2^{\pi P}$, the operator $\mathcal{O}_{m,\pi}$ is charged under $\eta$ and is attached to $\eta$ line. Equivalently, $\mathcal{O}_{m,\pi}$ is charged under $(-1)^F = \pi\eta$ and hence a fermionic operator attached to $\eta$ line. Then, the operator

$$\mathcal{O}_{m,\pi}^\pi := \mathcal{O}_{m,\pi} \otimes \mathcal{O}_\pi \qquad (\text{II}.27)$$

is a bosonic operator attached to $(-1)^F$ line and will be referred to as the fermionization of the operator $\mathcal{O}_m$ of $\mathfrak{T}$. In other words, the fermionization converts $\mathcal{O}_m$ into a bosonic operator in the R sector.

4. Consider a charged local operator $\mathcal{O}_f$ of $\mathfrak{T}$ in the twisted sector of $\mathbb{Z}_2$ symmetry. This can be converted into an operator

$$\mathcal{O}_{f,\pi} := \mathcal{O}_f \otimes \mathcal{O}_\pi \qquad (\text{II}.28)$$

of the Arf $\boxtimes \mathfrak{T}$ theory, which is attached to the $\pi P$ line and is uncharged under $\pi P$. Under a $\mathbb{Z}_2$ gauging, a twisted uncharged operator goes to an untwisted charged operator for the dual $\mathbb{Z}_2$ symmetry. Thus, after gauging of $\mathbb{Z}_2^{\pi P}$, the operator $\mathcal{O}_{f,\pi}$ is a fermionic operator in the NS sector, i.e. charged under $(-1)^F$ but unattached to $(-1)^F$ line. The operator $\mathcal{O}_{f,\pi}$ will be referred to as fermionization of the operator $\mathcal{O}_f$ of $\mathfrak{T}$.

Succinctly the fermionization rules for local operators are

$$\begin{aligned} \text{Untwisted, Uncharged} &\longrightarrow \text{NS sector, Boson} \\ \text{Untwisted, Charged} &\longrightarrow \text{R sector, Fermion} \\ \text{Twisted, Unharged} &\longrightarrow \text{R sector, Boson} \\ \text{Twisted, Charged} &\longrightarrow \text{NS sector, Fermion} \end{aligned} \qquad (\text{II}.29)$$

**Alternate Convention for Fermionization.** There is another convention for fermionization that one may adopt, which implements

$$\mathfrak{T} \longrightarrow \widetilde{\mathfrak{T}_f} \qquad (\text{II}.30)$$

but we do not adopt this convention in this paper. In this convention, the fermionized theory $\widetilde{\mathfrak{T}_f}$ is again

$$\widetilde{\mathfrak{T}}_f = \frac{\text{Arf} \boxtimes \mathfrak{T}}{\mathbb{Z}_2^{\pi P}} \qquad (\text{II}.31)$$

but the fermionic parity symmetry is implemented by

$$(-1)^F = \eta \qquad (\text{II}.32)$$

instead of $\pi\eta$.

The alternate fermionization acts on local operators according to

$$\begin{aligned} \text{Untwisted, Uncharged} &\longrightarrow \text{NS sector, Boson} \\ \text{Untwisted, Charged} &\longrightarrow \text{R sector, Boson} \\ \text{Twisted, Unharged} &\longrightarrow \text{R sector, Fermion} \\ \text{Twisted, Charged} &\longrightarrow \text{NS sector, Fermion} \end{aligned} \qquad (\text{II}.33)$$

The two fermionic theories are related by stacking of Arf TFT

$$\widetilde{\mathfrak{T}}_f = \text{Arf} \boxtimes \mathfrak{T}_f. \qquad (\text{II}.34)$$

Applying the original bosonization map (II.23) to $\widetilde{\mathfrak{T}_f}$ we obtain a bosonic theory $\widetilde{\mathfrak{T}}$. The map

$$\mathfrak{T} \longrightarrow \widetilde{\mathfrak{T}} \qquad (\text{II}.35)$$

is the bosonic gauging of the $\mathbb{Z}_2$ symmetry of $\mathfrak{T}$ [167–169, 180]

$$\widetilde{\mathfrak{T}} = \mathfrak{T}/\mathbb{Z}_2. \qquad (\text{II}.36)$$

Finally, the operation that maps

$$\mathfrak{T}_f \longrightarrow \widetilde{\mathfrak{T}} \qquad (\text{II}.37)$$

is known as the **Gliozzi-Scherk-Olive (GSO) projection** [181]. The relation between four systems $\mathfrak{T}$, $\mathfrak{T}_f$, $\widetilde{\mathfrak{T}}$, and $\widetilde{\mathfrak{T}_f}$ can be summarized in a diagram shown in Figure 1.

### E.  Examples

Let us now discuss simple examples of fermionic fusion supercategories and the corresponding bosonic fusion categories whose fermionization they are.

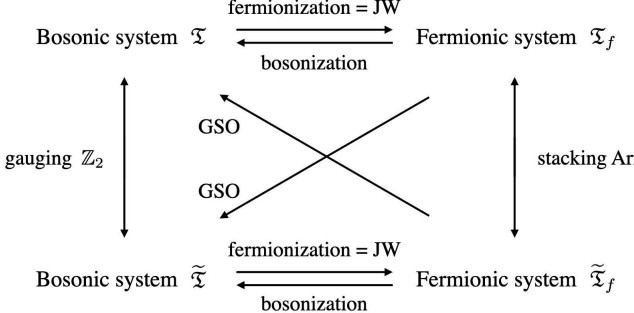

Figure 1. The fermionization and bosonization in 1+1d. JW stands for the Jordan-Wigner transformation and GSO stands for the GSO projection.

### 1. $\mathbb{Z}_2^f$ symmetry

The simplest example of a fermionic fusion supercategory is the one that describes only the fermion parity symmetry. The non-trivial indecomposable lines of this fusion $\pi$-supercategory are $(-1)^F$, $\pi$, and their fusion $\pi(-1)^F$.[10] These topological lines form a fusion $\pi$-supercategory $\mathsf{sVec}_{\mathbb{Z}_2} := \mathsf{sVec} \boxtimes \mathsf{Vec}_{\mathbb{Z}_2}$, where $\mathsf{sVec}$ is the fusion $\pi$-supercategory (of super vector spaces) generated by $\pi$ and $\mathsf{Vec}_{\mathbb{Z}_2}$ is the ordinary fusion category (of $\mathbb{Z}_2$ graded vector spaces) generated by $(-1)^F$. When no confusion can arise, the symmetry described by $\mathsf{sVec}_{\mathbb{Z}_2}$ will be simply denoted by $\mathbb{Z}_2^f$ and called the fermion parity symmetry. The fermion parity symmetry $\mathbb{Z}_2^f$ described by fusion $\pi$-supercategory $\mathsf{sVec}_{\mathbb{Z}_2}$ is the fermionization of a non-anomalous $\mathbb{Z}_2$ symmetry described by ordinary fusion category $\mathsf{Vec}_{\mathbb{Z}_2}$.

### 2. $\mathbb{Z}_4^f$ and $\mathbb{Z}_4^{\pi f}$ symmetries

Another simple example of a fermionic fusion supercategory is the one that describes the spin $\mathbb{Z}_4$ symmetry, which is a $\mathbb{Z}_4$ symmetry whose $\mathbb{Z}_2$ subgroup is the fermion parity symmetry $\mathbb{Z}_2^f$. By definition, a generator $P$ of the spin $\mathbb{Z}_4$ symmetry satisfies $P^2 = (-1)^F$. The fusion $\pi$-supercategory that describes this symmetry is $\mathsf{sVec}_{\mathbb{Z}_4} := \mathsf{sVec} \boxtimes \mathsf{Vec}_{\mathbb{Z}_4}$, where $\mathsf{sVec}$ is again the fusion $\pi$-supercategory generated by $\pi$ and $\mathsf{Vec}_{\mathbb{Z}_4}$ is the ordinary fusion category generated by $P$. The spin $\mathbb{Z}_4$ symmetry, described by $\mathsf{sVec}_{\mathbb{Z}_4}$ with the choice $(-1)^F = P^2$, is simply denoted by $\mathbb{Z}_4^f$

There is also a variant of the spin $\mathbb{Z}_4$ symmetry, which we denote by $\mathbb{Z}_4^{\pi f}$. This is a $\mathbb{Z}_4$ symmetry whose generator $P$ satisfies $P^2 = \pi(-1)^F$ rather than $P^2 = (-1)^F$. The two symmetries are related by stacking with Arf

---

[10] An indecomposable topological line $x$ is said to be non-trivial if there does not exist a bosonic isomorphism between $x$ and $1$.

TFT in the sense of section II C

$$\mathbb{Z}_4^{\pi f} = \mathbb{Z}_4^f \boxtimes \mathrm{Arf} \qquad (\text{II.38})$$

The $\mathbb{Z}_4^f$ symmetry is the fermionization of a non-anomalous $\mathbb{Z}_4$ symmetry described by ordinary fusion category $\mathsf{Vec}_{\mathbb{Z}_4}$. Let us for brevity denote the generator of this bosonic $\mathbb{Z}_4$ symmetry also by $P$. We want to fermionize with respect to the non-anomalous $\mathbb{Z}_2$ generated by $P^2$. The fact that we have a $\mathbb{Z}_4$ symmetry means that we have a trivalent junction formed by two incoming $P$ lines and one outgoing $P^2$ line.

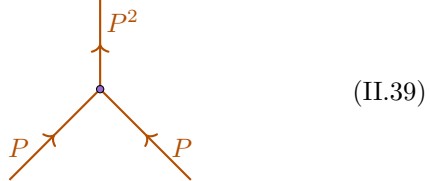

$$(\text{II.39})$$

The local operator describing this trivalent junction may be viewed as an uncharged (due to non-anomalous nature of $\mathbb{Z}_4$ symmetry) and twisted sector operator for this $\mathbb{Z}_2$ symmetry which, as discussed in (II.29), fermionizes to an R-sector bosonic operator attached to two incoming $P$ lines. Thus the fermionized symmetry is such that two $P$ lines fuse to form $(-1)^F$ with a bosonic operator sitting at the corresponding trivalent junction, or in other words the fermionized symmetry is $\mathbb{Z}_4^f$.

The $\mathbb{Z}_4^{\pi f}$ symmetry is the fermionization of $\mathbb{Z}_2 \times \mathbb{Z}_2$ symmetry with mixed 't Hooft anomaly, which is obtained by bosonically gauging the $\mathbb{Z}_2$ subgroup of the non-anomalous bosonic $\mathbb{Z}_4$ symmetry. Let us denote the generators of this $\mathbb{Z}_2 \times \mathbb{Z}_2$ by $P$ and $P'$. We want to fermionize with respect to the non-anomalous $\mathbb{Z}_2$ generated by $P'$. The fact that we have a mixed 't Hooft anomaly means that a trivalent junction formed by two incoming $P$ lines and one outgoing identity line is charged under $P'$.

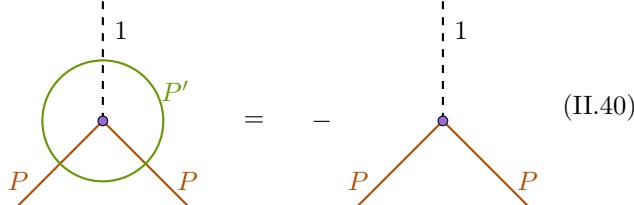

$$(\text{II.40})$$

The local operator describing this trivalent junction is thus a charged untwisted sector operator for $P'$ which, as discussed in (II.29), fermionizes to an R-sector fermionic operator attached to two incoming $P$ lines. Thus the fermionized symmetry is such that two $P$ lines fuse to form $\pi(-1)^F$ with a bosonic operator sitting at the corresponding trivalent junction, or in other words the fermionized symmetry is $\mathbb{Z}_4^{\pi f}$.

We emphasize that $\mathbb{Z}_4^f$ and $\mathbb{Z}_4^{\pi f}$ are different fermionic symmetries because there is no autoequivalence of the underlying fusion $\pi$-supercategory $\mathsf{sVec}_{\mathbb{Z}_4}$ that maps $(-1)^F$

to $\pi(-1)^F$. Equivalently, the bosonization of $\mathbb{Z}_4^f$ and that of $\mathbb{Z}_4^{\pi f}$ are different symmetries because they have different group structures and anomalies.

The fact that $\mathbb{Z}_4^f$ and $\mathbb{Z}_4^{\pi f}$ symmetries are different is also consistent with the classification of 1+1d SPT phases with these symmetries. It is known that 1+1d SPT phases with $\mathbb{Z}_4^f$ symmetry are classified by a second group cohomology $H^2(\mathbb{Z}_4, \mathrm{U}(1))$ as a set [182, 183]. Since $H^2(\mathbb{Z}_4, \mathrm{U}(1))$ vanishes, $\mathbb{Z}_4^f$ symmetry admits only one SPT phase, which is a trivial phase. This classification implies that the stacking of this SPT phase and the Arf TFT is not an SPT phase with $\mathbb{Z}_4^f$ symmetry. This can be understood as a consequence of the fact that stacking the Arf TFT changes the symmetry from $\mathbb{Z}_4^f$ to $\mathbb{Z}_4^{\pi f}$.

### 3. $\mathsf{Rep}(S_3)^f$ and $\mathsf{Rep}(S_3)^{\pi f}$ symmetries

Simple examples of non-invertible fermionic symmetries are given by fusion $\pi$-supercategory

$$\mathsf{sVec} \boxtimes \mathsf{Rep}(S_3) \qquad (\text{II}.41)$$

$\mathsf{Rep}(S_3)$ denotes the fusion category formed by representations of $S_3$, which has two non-trivial topological lines $P$ and $E$, where $P$ is an invertible line corresponding to a one-dimensional irreducible representation of $S_3$, while $E$ is a non-invertible line corresponding to a two-dimensional irreducible representation of $S_3$. The fusion rules of these topological lines are given by

$$P^2 \cong 1, \quad PE \cong EP \cong E, \quad E^2 \cong 1 \oplus P \oplus E. \quad (\text{II}.42)$$

This supercategory gives rise to two fermionic symmetries $\mathsf{Rep}(S_3)^f$ for which

$$(-1)^F = P \qquad (\text{II}.43)$$

and $\mathsf{Rep}(S_3)^{\pi f}$ for which

$$(-1)^F = \pi P \qquad (\text{II}.44)$$

$\mathsf{Rep}(S_3)^f$ is obtained by fermionizing $\mathsf{Rep}(S_3)$ with respect to $P$. To see this, note that a trivalent junction involving two $E$ lines and a $P$ line fermionizes to a trivalent junction involving two $E$ lines and a $(-1)^F$ line, which means that the fermionized symmetry has the same fusion rules as $\mathsf{Rep}(S_3)$ with fermion parity being $P$. In general, the fermionization of $\mathsf{Rep}(G)$ symmetry is described by $\mathsf{sRep}(G) := \mathsf{sVec} \boxtimes \mathsf{Rep}(G)$ [9].

On the other hand, $\mathsf{Rep}(S_3)^{\pi f}$ is obtained by fermionizing $S_3$ group symmetry. Let us label elements of $S_3$ as

$$S_3 = \{1, a, a^2, b, ab, a^2 b\}, \quad a^3 = b^2 = 1, \quad ba = a^2 b \qquad (\text{II}.45)$$

We can fermionize with respect to $b$. The non-simple line $a \oplus a^2$ before fermionization becomes a simple line

$E$ after fermionization, since $b$ exchanges $a$ and $a^2$. Let $1_a$ and $1_{a^2}$ be identity local operators along $a$ and $a^2$ respectively. Then the local operator $1_a - 1_{a^2}$ lives on $a \oplus a^2$ and may be regarded as a charged twisted operator with respect to the $\mathbb{Z}_2$ symmetry being fermionized. The fermionization of this operator is a fermionic operator providing a trivalent junction between two $E$ lines and one $(-1)^F$ line, or equivalently a bosonic operator providing a trivalent junction between two $E$ lines and one $\pi(-1)^F$ line. This reproduces fusion rules for $\mathsf{Rep}(S_3)^{\pi f}$.

Lattice models with $\mathsf{Rep}(S_3)^f$ and $\mathsf{Rep}(S_3)^{\pi f}$ symmetries were studied recently in [151, Appendix G].

### 4. $\mathbb{Z}_2 \times \mathbb{Z}_2^f$ symmetry with a Gu-Wen anomaly

A simple example of an anomalous fermionic symmetry is $\mathbb{Z}_2 \times \mathbb{Z}_2^f$ symmetry with a Gu-Wen anomaly [184]. To describe this symmetry, we first recall the classification of anomalies of $\mathbb{Z}_2 \times \mathbb{Z}_2^f$ symmetry in 1+1 dimensions. Anomalies of $\mathbb{Z}_2 \times \mathbb{Z}_2^f$ symmetry in 1+1d are classified by $\Omega^3_{\mathrm{spin}}(B\mathbb{Z}_2) \cong \mathbb{Z}_8$ [185–188]. An anomaly $\nu \in \mathbb{Z}_8$ is called a bosonic anomaly when $\nu = 4$ because the anomaly in this case descends from the anomaly of an ordinary $\mathbb{Z}_2$ symmetry, which can be realized in purely bosonic systems. An anomaly $\nu$ is called a Gu-Wen anomaly when $\nu = 2 \bmod 4$ because this anomaly can be realized on the boundary of a (2+1)d Gu-Wen SPT phase [184]. An anomaly $\nu$ is called a beyond Gu-Wen anomaly when $\nu$ is odd, which can be realized on the boundary of a (2+1)d SPT phase beyond Gu-Wen type [186]. A $\mathbb{Z}_2 \times \mathbb{Z}_2^f$ symmetry with an anomaly $\nu = 1$ is realized in the quantum field theory of a massless Majorana fermion, where the first $\mathbb{Z}_2$ of $\mathbb{Z}_2 \times \mathbb{Z}_2^f$ is the fermion parity symmetry of the left moving fermion $(-1)^{F_L}$. In a UV lattice Majorana model, a $\nu = 1$ anomaly is realized as an LSM anomaly between lattice translations (where a translation unit cell contains a single Majorana operator) and fermion parity [189, 190]. In the IR, translation by a single unit cell flows to $(-1)^{F_L}$. Since the anomaly is additive under the stacking of QFTs, a $\mathbb{Z}_2 \times \mathbb{Z}_2^f$ symmetry with an anomaly $\nu \in \mathbb{Z}_8$ is realized in the field theory of $\nu$ massless Majorana fermions.

The fusion $\pi$-supercategory that describes an anomalous $\mathbb{Z}_2 \times \mathbb{Z}_2^f$ symmetry depends on the anomaly $\nu \in \mathbb{Z}_8$. When the anomaly is bosonic, i.e., when $\nu = 4$, the fusion $\pi$-supercategory is just the product $\mathsf{sVec} \boxtimes \mathsf{Vec}^\omega_{\mathbb{Z}_2} \boxtimes \mathsf{Vec}_{\mathbb{Z}_2}$, where $\mathsf{Vec}^\omega_{\mathbb{Z}_2}$ represents an ordinary $\mathbb{Z}_2$ symmetry with a non-trivial anomaly $\omega \neq 0 \in H^3(\mathbb{Z}_2, \mathrm{U}(1)) \cong \mathbb{Z}_2$. We do not discuss this bosonic anomaly in this paper. The fusion $\pi$-supercategory that describes a $\mathbb{Z}_2 \times \mathbb{Z}_2^f$ symmetry with a Gu-Wen anomaly will be discussed shortly, while the fusion $\pi$-supercategory that describes a $\mathbb{Z}_2 \times \mathbb{Z}_2^f$ symmetry with a beyond Gu-Wen anomaly will be discussed in the next example.

A $\mathbb{Z}_2 \times \mathbb{Z}_2^f$ symmetry with a Gu-Wen anomaly $\nu = 2 \bmod 4$ is the fermionization of $\mathbb{Z}_4$ symmetry with an

anomaly $\omega = 2 \in H^3(\mathbb{Z}_4, \mathrm{U}(1)) \cong \mathbb{Z}_4$ [166, 169]. This fermionic symmetry is generated by three topological lines $\pi$, $\eta_+$, and $\eta_- := \eta_+(-1)^F$, which obey the following fusion rules: [167]

$$\eta_+^2 \cong \eta_-^2 \cong \pi. \qquad (\text{II.46})$$

Here, the isomorphisms in the above equation are all bosonic. Equivalently, we have fermionic isomorphisms $\eta_+^2 \cong 1$ and $\eta_-^2 \cong 1$, which means that the junction of two incoming $\eta_i$ lines ($i = +, -$) is fermionic. This also implies that $\eta_+$ and $\eta_-$ are self-dual with fermionic isomorphisms $\eta_+ \cong \eta_+^*$ and $\eta_- \cong \eta_-^*$.

The fermionic pentagon equation implies that the $F$-symbols $(F_{\eta_+}^{\eta_+\eta_+\eta_+})_{11}$ and $(F_{\eta_-}^{\eta_-\eta_-\eta_-})_{11}$ are either $+i$ or $-i$ [167]. Furthermore, since the fermion parity symmetry generated by $(-1)^F = \eta_+\eta_-$ is non-anomalous, $(F_{\eta_+}^{\eta_+\eta_+\eta_+})_{11}$ and $(F_{\eta_-}^{\eta_-\eta_-\eta_-})_{11}$ should have the opposite signs. Thus, without loss of generality, we can choose

$$(F_{\eta_+}^{\eta_+\eta_+\eta_+})_{11} = +i, \quad (F_{\eta_-}^{\eta_-\eta_-\eta_-})_{11} = -i. \qquad (\text{II.47})$$

The fusion $\pi$-supercategory that describes this symmetry is $\mathcal{S}_+ \boxtimes_{\mathrm{sVec}} \mathcal{S}_-$, where $\mathcal{S}_i$ is the fusion $\pi$-supercategory generated by $\pi$ and $\eta_i$, and $\boxtimes_{\mathrm{sVec}}$ denotes the Deligne tensor product over sVec, which essentially means that we identify together the two $\pi$ lines coming from the two $\mathcal{S}_i$ factors. We note that $\mathbb{Z}_2 \times \mathbb{Z}_2^f$ symmetries with anomalies $\nu = 2$ and $\nu = 6$ are described by the same fusion $\pi$-supercategory $\mathcal{S}_+ \boxtimes_{\mathrm{sVec}} \mathcal{S}_-$. The only difference between these symmetries is the choice of the generator of $\mathbb{Z}_2$ symmetry. More specifically, if we choose $\eta_+$ as the generator of $\mathbb{Z}_2$, the full $\mathbb{Z}_2 \times \mathbb{Z}_2^f$ symmetry has an anomaly $\nu = 2$. On the other hand, if we choose $\eta_-$ as the generator of $\mathbb{Z}_2$, the full $\mathbb{Z}_2 \times \mathbb{Z}_2^f$ symmetry has an anomaly $\nu = 6$. In the example of $\nu$ massless Majorana fermions, $\eta_+$ and $\eta_-$ correspond respectively to the fermion parity symmetries of the left and right mover.

We note that $\mathbb{Z}_2 \times \mathbb{Z}_2^f$ symmetry with a Gu-Wen anomaly is invariant under stacking the Arf TFT. This is because its bosonization, i.e., a $\mathbb{Z}_4$ symmetry with an anomaly $\omega = 2$, is invariant under gauging the non-anomalous $\mathbb{Z}_2$ subgroup. cf. Figure 1

### 5. $\mathbb{Z}_2 \times \mathbb{Z}_2^f$ symmetry with a beyond Gu-Wen anomaly

A $\mathbb{Z}_2 \times \mathbb{Z}_2^f$ symmetry with a beyond Gu-Wen anomaly has two $q$-type objects $q$ and $q' := q(-1)^F$ [17]. These $q$-type objects obey the following Ising-like fusion rules [9]:

$$q^2 \cong (q')^2 \cong 1 \oplus \pi. \qquad (\text{II.48})$$

Here, the isomorphisms in the above equation are supposed to be homogeneous (e.g., bosonic). Each $q$-type object together with $\pi$ generates a fusion $\pi$-supercategory whose underlying category is either the Ising category $\mathsf{Ising}_+$ or the other $\mathbb{Z}_2$ Tambara-Yamagami category

$\mathsf{Ising}_-$.[11] We denote these fusion $\pi$-supercategories by $\mathcal{S}_{\mathsf{Ising}_+}$ and $\mathcal{S}_{\mathsf{Ising}_-}$ respectively. When $\nu = 1$ or $7$, both $q$-type objects $q$ and $q'$ generate $\mathcal{S}_{\mathsf{Ising}_+}$, corresponding to the fact that $\mathbb{Z}_2 \times \mathbb{Z}_2^f$ symmetry with this anomaly is the fermionization of $\mathsf{Ising}_+$ symmetry [166, 169]. On the other hand, when $\nu = 3$ or $5$, both $q$-type objects $q$ and $q'$ generate $\mathcal{S}_{\mathsf{Ising}_-}$, corresponding to the fact that $\mathbb{Z}_2 \times \mathbb{Z}_2^f$ symmetry with this anomaly is the fermionization of $\mathsf{Ising}_-$ symmetry [169]. In particular, $\mathbb{Z}_2 \times \mathbb{Z}_2^f$ symmetries with anomalies $\nu = 1$ and $\nu = 7$ are described by the same fusion $\pi$-supercategory. Similarly, $\mathbb{Z}_2 \times \mathbb{Z}_2^f$ symmetries with anomalies $\nu = 3$ and $\nu = 5$ are described by the same fusion $\pi$-supercategory. In the example of $\nu$ massless Majorana fermions, two $q$-type objects $q$ and $q'$ are the fermion parity lines for the left mover and the right mover. We note that the fermion parity lines for the left mover and the right mover generate the same fusion $\pi$-supercategory even though they have different anomalies. These anomalies are distinguished by more subtle data that are necessary to mathematically define fermionic symmetry [172].

We emphasize that the q-type objects $q$ and $q'$ are non-invertible in the sense that there are no topological lines $q^{-1}$ and $(q')^{-1}$ such that $qq^{-1} \cong q^{-1}q \cong 1$ and $q'(q')^{-1} \cong (q')^{-1}q' \cong 1$. In particular, the quantum dimensions of $q$ and $q'$ are both $\sqrt{2}$. However, the non-invertibility of these topological lines is rather mild. Indeed, the fusion rules (II.48) imply that the corresponding symmetry operators $\mathcal{D}_q$ and $\mathcal{D}_{q'}$ acting on NS sector local operators obey the $\mathbb{Z}_2$ group-like fusion rule up to normalization:

$$(\mathcal{D}_q)^2 = (\mathcal{D}_{q'})^2 = 2. \qquad (\text{II.49})$$

Here, the last equality follows from the fact that the $\pi$ line acts as $+1$ on NS sector local operators. Therefore, the symmetry operators divided by their quantum dimensions

$$(-1)^{F_L} = \mathcal{D}_q/\sqrt{2}, \quad (-1)^{F_R} = \mathcal{D}_{q'}/\sqrt{2} \qquad (\text{II.50})$$

look as if they generate $\mathbb{Z}_2$ symmetries. As such, the symmetry generated by $q$ and $q'$ is often called an anomalous $\mathbb{Z}_2 \times \mathbb{Z}_2^f$ symmetry in the literature even though the generator of this symmetry is non-invertible.

Naively, $\mathcal{D}_q$ and $\mathcal{D}_{q'}$ square to zero in the R sector because the $\pi$ line acts as $-1$ on R sector operators. To avoid this, we can modify the symmetry operators in the R sector by putting a fermionic point-like defect on them, which is possible because $q$ and $q'$ are $q$-type objects. As a result, these symmetry operators anti-commute with

---

[11] The underlying category of a fusion supercategory $\mathcal{S}$ is a fusion category consisting of all objects and all bosonic morphisms of $\mathcal{S}$ [173]. The categories $\mathsf{Ising}_\pm$ are the only fusion categories with the Ising fusion rules [191].

$(-1)^F$ in the R sector. This anti-commutation relation was observed in the example of massless Majorana fermions [166, 168, 192].

It should be noted that the $q$- and $q'$-twisted sectors are well-defined as vector spaces of integral dimensions, while the $(-1)^{F_L}$- and $(-1)^{F_R}$-twisted sectors are not because of the factor of $\sqrt{2}$ in eq. (II.50). In particular, the dimensions of the $q$- and $q'$-twisted sectors are even integers because the fermionic automorphisms of $q$ and $q'$ imply the same numbers of bosonic states and fermionic states. Accordingly, the dimensions of $(-1)^{F_L}$- and $(-1)^{F_R}$-twisted sectors are formally given by integer multiples of $\sqrt{2}$. In the literature, the factor of $\sqrt{2}$ is also understood as the formal dimension of a single Majorana fermion in 0+1d.

In some cases, treating this symmetry as if it is invertible appears totally fine. For example, the above $\mathbb{Z}_2 \times \mathbb{Z}_2^f$ symmetry with a beyond Gu-Wen anomaly falls into the classification of anomalies of invertible symmetries via bordism groups $\Omega_{\mathrm{spin}}^3(B\mathbb{Z}_2) \cong \mathbb{Z}_8$ [185–188]. Nevertheless, in this paper, we regard this symmetry as a non-invertible symmetry due to its non-invertible fusion rules (II.48).

We note that a $\mathbb{Z}_2 \times \mathbb{Z}_2^f$ symmetry with a beyond Gu-Wen anomaly is invariant under stacking the Arf TFT. This is because its bosonization, which is either $\mathsf{Ising}_+$ or $\mathsf{Ising}_-$, is invariant under gauging its non-anomalous $\mathbb{Z}_2$ subgroup.

Another point to be noted is that regarding this fermionic symmetry as an invertible symmetry leads to the notion that the bosonic $\mathsf{Ising}_\pm$ symmetries are not intrinsically non-invertible. However, this conclusion should be avoided as technically the fermionic symmetry is non-invertible, and hence the bosonic $\mathsf{Ising}_\pm$ symmetries should be regarded as intrinsically non-invertible symmetries.

### 6. Bosonic Symmetries as Special Cases of Fermionic Symmetries

A bosonic symmetry described by an ordinary fusion category $\mathcal{S}$ trivially gives rise to two fermionic symmetries. Both of them are based on the fusion $\pi$-supercategory

$$\mathsf{sVec} \boxtimes \mathcal{S} \tag{II.51}$$

One of the fermionic symmetries $\mathcal{S}_f$ is obtained by choosing $(-1)^F$ to be trivial, i.e. $(-1)^F = 1$, while the other fermionic symmetry $\widetilde{\mathcal{S}}_f$ is obtained by choosing $(-1)^F = \pi$.

$\mathcal{S}_f$ is obtained by fermionizing $\mathcal{S}$ with respect to trivial $\mathbb{Z}_2$ symmetry generated by $P = 1$, and $\widetilde{\mathcal{S}}_f$ is obtained by fermionizing the multi-fusion category

$$\mathcal{S} \boxtimes \mathrm{Mat}_2(\mathsf{Vec}) \tag{II.52}$$

where $\mathrm{Mat}_2(\mathsf{Vec})$ is a multi-fusion category formed by $2 \times 2$ matrices valued in the category $\mathsf{Vec}$ of finite dimensional vector spaces. The $\mathbb{Z}_2$ symmetry for the fermionization is generated by off-diagonal matrix with each entry being 1-dimensional vector spaces, which we may label as $1_{01} \oplus 1_{10}$.

Let $\mathfrak{T}$ be a bosonic system with symmetry $\mathcal{S}$. The symmetry $\mathcal{S}_f$ is carried by the system $\mathfrak{T}_f$ obtained by stacking $\mathfrak{T}$ with trivial fermionic 2d TFT. On the other hand, $\widetilde{\mathcal{S}}_f$ is carried by the system $\widetilde{\mathfrak{T}}_f$ obtained by stacking $\mathfrak{T}$ with Arf TFT.

## III. SYMMETRY TFT

In this section, we first briefly review the symmetry TFT construction of 1+1d bosonic systems with general fusion category symmetry following [2, 7, 92], see also [8, 12, 15, 193, 194] for earlier discussions and [140] for a mathematical formulation. The symmetry TFT is also called a symmetry topological order [34, 71, 107, 162, 195, 196] or topological holography [48, 197, 198] in the condensed matter literature.[12]

We then proceed to discuss SymTFTs for fermionic symmetries. As we will see, the symmetry TFTs for fermionic symmetries remain bosonic, while their symmetry boundaries are taken to be fermionic. Since the symmetry TFTs are bosonic, one can readily apply the methods developed in [3–6] to the study of fermionic gapped and gapless phases with non-invertible symmetries, which will be the subject of later sections.

### A. SymTFTs for Bosonic Symmetries

Let's first review the case of bosonic symmetries. Let $\mathcal{S}$ be a fusion category specifying a bosonic symmetry, i.e. a symmetry of bosonic theories. The symmetry TFT associated to $\mathcal{S}$ is a 3d bosonic TFT $\mathfrak{Z}(\mathcal{S})$ that admits a bosonic topological boundary condition $\mathfrak{B}_{\mathcal{S}}^{\mathrm{sym}}$, such that the topological line operators living on $\mathfrak{B}_{\mathcal{S}}^{\mathrm{sym}}$ form the fusion category $\mathcal{S}$. This TFT can be obtained by performing the Turaev-Viro-Barrett-Westbury [199, 200] construction with $\mathcal{S}$ as the input category and has the property that the topological line defects of the 3d TFT, also known as anyons, form the MTC $\mathcal{Z}(\mathcal{S})$, known as the Drinfeld center of $\mathcal{S}$.

By a bosonic topological boundary condition we mean a topological boundary condition which does not carry a fermion parity symmetry, or in other words the fermion parity symmetry acts trivially and is generated by $(-1)^F = 1$. In particular this implies that any topological local operator arising at an end of an anyon $a \in \mathcal{Z}(\mathcal{S})$

---

[12] The symmetry TFT was originally called a categorical symmetry in [195, 196].

along the boundary (and not attached to any topological line living on the boundary) is a boson, i.e. $2\pi$ rotations centered on such a local operator do not change correlation functions. This implies that any anyon $a$ that can end along the boundary must have trivial topological spin. Such an anyon is also referred to as a boson.

Note that there may be multiple different bosonic topological boundary conditions of $\mathfrak{Z}(\mathcal{S})$ carrying the fusion category $\mathcal{S}$. In what follows, we fix one such boundary condition to be $\mathfrak{B}_{\mathcal{S}}^{\mathrm{sym}}$ and refer to it as the symmetry boundary.

The connection of the 3d SymTFT $\mathfrak{Z}(\mathcal{S})$ to 1+1d theories with $\mathcal{S}$ symmetry is as follows. Any $\mathcal{S}$-symmetric 1+1d bosonic theory $\mathfrak{T}$ can be constructed as an interval compactification of $\mathfrak{Z}(\mathcal{S})$, with one of the ends of the interval occupied by the symmetry boundary $\mathfrak{B}_{\mathcal{S}}^{\mathrm{sym}}$, while the other end occupied by a not necessarily topological boundary condition $\mathfrak{B}_{\mathfrak{T}}^{\mathrm{phys}}$, that captures the dynamical information of the system $\mathfrak{T}$. We call $\mathfrak{B}_{\mathfrak{T}}^{\mathrm{phys}}$ as the physical boundary. The topological line operators of $\mathfrak{T}$ implementing the $\mathcal{S}$ symmetry arise as images of topological line operators living on $\mathfrak{B}_{\mathcal{S}}^{\mathrm{sym}}$ under this interval compactification.

$\mathfrak{T}$ is gapped if $\mathfrak{B}_{\mathfrak{T}}^{\mathrm{phys}}$ is gapped, while it is gapless if $\mathfrak{B}_{\mathfrak{T}}^{\mathrm{phys}}$ is gapless. In particular, 1+1d bosonic gapped phases with symmetry $\mathcal{S}$ are obtained by choosing the physical boundary to be topological [3, 4, 12], and are more precisely identified with deformation classes of topological boundary conditions of the SymTFT $\mathfrak{Z}(\mathcal{S})$. The construction described above is known as the symmetry TFT construction or sandwich construction in the literature.

Consider now an arbitrary topological boundary condition $\mathfrak{B}^{\mathrm{top}}$ of the SymTFT $\mathfrak{Z}(\mathcal{S})$. The topological lines of $\mathfrak{B}^{\mathrm{top}}$ form a fusion category $\mathcal{S}'$ that is Morita equivalent to $\mathcal{S}$ [201]. The boundary $\mathfrak{B}^{\mathrm{top}}$ can be produced from $\mathfrak{B}_{\mathcal{S}}^{\mathrm{sym}}$ by performing a gauging of the $\mathcal{S}$ symmetry of $\mathfrak{B}_{\mathcal{S}}^{\mathrm{sym}}$ corresponding to an indecomposable module category $\mathcal{M}$ over $\mathcal{S}$, which is the category formed by topological lines lying at an interface between $\mathfrak{B}_{\mathcal{S}}^{\mathrm{sym}}$ and $\mathfrak{B}^{\mathrm{top}}$. We express this gauging as

$$\mathfrak{B}^{\mathrm{top}} = \mathfrak{B}_{\mathcal{S}}^{\mathrm{sym}}/\mathcal{M}. \qquad \text{(III.1)}$$

The symmetry $\mathcal{S}'$ of $\mathfrak{B}^{\mathrm{top}}$ is a combination of the residual symmetry left from the gauging procedure and the dual symmetry obtained after gauging, which can be expressed as [10]

$$\mathcal{S}' = \mathcal{S}_{\mathcal{M}}^{*}, \qquad \text{(III.2)}$$

where $\mathcal{S}_{\mathcal{M}}^{*}$ is the fusion category formed by $\mathcal{S}$-module endofunctors of $\mathcal{M}$ [161]. Performing the sandwich construction by replacing the symmetry boundary $\mathfrak{B}_{\mathcal{S}}^{\mathrm{sym}}$ with $\mathfrak{B}^{\mathrm{top}}$ while keeping the physical boundary $\mathfrak{B}_{\mathfrak{T}}^{\mathrm{phys}}$ fixed leads to the (1+1)d theory

$$\mathfrak{T}' = \mathfrak{T}/\mathcal{M} \qquad \text{(III.3)}$$

obtained by gauging the symmetry $\mathcal{S}$ of $\mathfrak{T}$ according the module category $\mathcal{M}$.

## B. SymTFTs for Fermionic Symmetries

Whatever we have discussed so far extends straightforwardly to fermionic symmetries. Let $\mathcal{S}_f$ be a fermionic fusion supercategory specifying a fermionic symmetry, i.e. a symmetry of fermionic theories. The symmetry TFT associated to $\mathcal{S}_f$ is again a 3d bosonic TFT $\mathfrak{Z}(\mathcal{S}_f)$ that admits a *fermionic* topological boundary condition $\mathfrak{B}_{\mathcal{S}_f}^{\mathrm{sym}}$, such that the topological line operators living on $\mathfrak{B}_{\mathcal{S}_f}^{\mathrm{sym}}$ form the fermionic fusion supercategory $\mathcal{S}_f$. The anyons of $\mathfrak{Z}(\mathcal{S}_f)$ form an MTC that we label as $\mathcal{Z}(\mathcal{S}_f)$, which can be understood as the Drinfeld center of $\mathcal{S}_f$.

By a fermionic topological boundary condition we mean a topological boundary condition on which the fermion parity symmetry $(-1)^F$ may act non-trivially. This includes bosonic topological boundary conditions as sub-cases where the action of $(-1)^F$ is trivial. Moreover, each bosonic topological boundary $\mathfrak{B}$ gives rise to a fermionic topological boundary

$$\widetilde{\mathfrak{B}}_f = \mathfrak{B} \boxtimes \mathrm{Arf} \qquad \text{(III.4)}$$

which is obtained by stacking $\mathfrak{B}$ with the 2d Arf theory, on which $(-1)^F$ acts non-trivially as $(-1)^F = \pi$. We say that $\widetilde{\mathfrak{B}}_f$ is a non-bosonic fermionic topological boundary obtained by adding an Arf term to the bosonic topological boundary $\mathfrak{B}$.

We may also have fermionic topological boundary conditions for which $(-1)^F \notin \{1, \pi\}$. Such topological boundaries may be referred to as *genuinely fermionic*. On a genuinely fermionic topological boundary we necessarily have at least one topological local operator arising at the end of some non-trivial anyon $a$ of the bulk 3d TFT (and not attached to any topological line living on the boundary) which is a fermion i.e. a local $2\pi$ rotation centered on the local operator changes a correlation function by a non-trivial sign. This implies that the anyon $a$ carries topological spin $-1$. Such an anyon is referred to as a fermion.

More generally, a fermionic topological boundary condition satisfies the following property: any topological local operator arising at an end of an anyon $a \in \mathcal{Z}(\mathcal{S}_f)$ along the boundary (and not attached to any topological line living on the boundary) is either a boson or a fermion.

Again, there may be multiple different fermionic topological boundary conditions of $\mathfrak{Z}(\mathcal{S}_f)$ carrying the same fermionic fusion supercategory $\mathcal{S}_f$. In what follows, we fix one such boundary condition to be $\mathfrak{B}_{\mathcal{S}_f}^{\mathrm{sym}}$ and refer to it as the symmetry boundary.

The connection of the 3d SymTFT $\mathfrak{Z}(\mathcal{S}_f)$ to 1+1d theories with $\mathcal{S}_f$ symmetry is analogous to the bosonic case. Any $\mathcal{S}_f$-symmetric 1+1d fermionic theory $\mathfrak{T}_f$ can be constructed as an interval compactification of $\mathfrak{Z}(\mathcal{S}_f)$, with one of the ends of the interval occupied by the symmetry boundary $\mathfrak{B}_{\mathcal{S}_f}^{\mathrm{sym}}$, while the other end is occupied by a not necessarily topological *bosonic* boundary condition $\mathfrak{B}_{\mathfrak{T}_f}^{\mathrm{phys}}$, that captures the dynamical information of the

system $\mathfrak{T}_f$. We call $\mathfrak{B}^{\mathrm{phys}}_{\mathfrak{T}_f}$ as the physical boundary. The topological line operators of $\mathfrak{T}_f$ implementing the $\mathcal{S}_f$ symmetry arise as images of topological line operators living on $\mathfrak{B}^{\mathrm{sym}}_{\mathcal{S}_f}$ under this interval compactification. This construction of 1+1d fermionic systems was originally proposed in [8] with a particular focus on the case of invertible symmetries.

$\mathfrak{T}_f$ is gapped if $\mathfrak{B}^{\mathrm{phys}}_{\mathfrak{T}_f}$ is gapped, while it is gapless if $\mathfrak{B}^{\mathrm{phys}}_{\mathfrak{T}_f}$ is gapless. In particular, 1+1d fermionic gapped phases with symmetry $\mathcal{S}_f$ are obtained by choosing the physical boundary to be a bosonic topological boundary, and are more precisely identified with deformation classes of bosonic topological boundary conditions of the SymTFT $\mathfrak{Z}(\mathcal{S}_f)$. This construction described above may be referred to as the fermionic symmetry TFT construction or fermionic sandwich construction.

Consider now an arbitrary fermionic topological boundary condition $\mathfrak{B}^{\mathrm{top}}_f$ of the SymTFT $\mathfrak{Z}(\mathcal{S}_f)$, which may actually be bosonic, i.e. the fermionic parity may act trivially on it. Such a boundary can be obtained from $\mathfrak{B}^{\mathrm{sym}}_{\mathcal{S}_f}$ by performing a fermionic gauging of the $\mathcal{S}_f$ symmetry of $\mathfrak{B}^{\mathrm{sym}}_{\mathcal{S}_f}$. By a fermionic gauging we include all possible combinations of gaugings, fermionizations and bosonizations. Such a fermionic gauging corresponds to an indecomposable supermodule category $\mathcal{M}_f$ of $\mathcal{S}_f$, which is the category formed by topological lines lying at an interface between $\mathfrak{B}^{\mathrm{sym}}_{\mathcal{S}_f}$ and $\mathfrak{B}^{\mathrm{top}}_f$. We express this gauging as

$$\mathfrak{B}^{\mathrm{top}}_f = \mathfrak{B}^{\mathrm{sym}}_{\mathcal{S}_f}/\mathcal{M}_f. \tag{III.5}$$

The symmetry formed by topological lines of $\mathfrak{B}^{\mathrm{top}}_f$, which is a combination of the residual symmetry left from the gauging procedure and the dual symmetry obtained after the gauging procedure, is $(\mathcal{S}_f)^*_{\mathcal{M}_f}$, which is the fermionic fusion supercategory formed by $\mathcal{S}_f$-supermodule endofunctors of $\mathcal{M}_f$. Performing the sandwich construction by replacing the symmetry boundary $\mathfrak{B}^{\mathrm{sym}}_{\mathcal{S}_f}$ with $\mathfrak{B}^{\mathrm{top}}_f$ while keeping the physical boundary $\mathfrak{B}^{\mathrm{phys}}_{\mathfrak{T}}$ fixed leads to the (1+1)d theory

$$\mathfrak{T}'_f = \mathfrak{T}_f/\mathcal{M}_f \tag{III.6}$$

obtained by a fermionic gauging of the symmetry $\mathcal{S}_f$ of $\mathfrak{T}_f$ according the module category $\mathcal{M}_f$.

Note that there always exists a bosonic topological boundary condition of $\mathfrak{Z}(\mathcal{S}_f)$, obtained simply by bosonizing $\mathfrak{B}^{\mathrm{sym}}_{\mathcal{S}_f}$, which is possible since we assume that $\mathbb{Z}^f_2$ subsymmetry of $\mathcal{S}_f$ is non-anomalous. When $\mathbb{Z}^f_2$ is anomalous, this is no longer possible and the bosonic symmetry TFT for such a fermionic symmetry is no longer non-chiral, that is does not admit a bosonic topological boundary condition. More precisely, the symmetry TFT for an anomalous $\mathbb{Z}^f_2$ symmetry with an anomaly $n \in \mathbb{Z}_{16}$ is the $\mathrm{Spin}(n)_1$ Chern-Simons theory [8, 202], which is a 3d chiral bosonic TFT.

## C. Bosonic Topological Boundaries

In light of the above discussion, let us discuss how one can characterize bosonic and fermionic topological boundaries of a 3d TFT $\mathfrak{Z}$ whose anyons form MTC $\mathcal{Z}$. For bosonic boundaries, it is well known that they are characterized by Lagrangian algebras in $\mathcal{Z}$ [201, 203–205]. A Lagrangian algebra $\mathcal{A}$ is an object in $\mathcal{Z}$ of the form

$$\mathcal{A} = \bigoplus_{a \in \mathcal{Z}} N_a a, \qquad N_a \in \mathbb{Z}_{\geq 0}, \ N_1 = 1, \tag{III.7}$$

where the summation is taken over all simple anyons $a$ in $\mathcal{Z}$. This object is additionally equipped with a commutative multiplication (which is a morphism $\mathcal{A} \otimes \mathcal{A} \to \mathcal{A}$ in $\mathcal{Z}$) and satisfies

$$\dim(\mathcal{A})^2 = \sum_a \dim(a)^2. \tag{III.8}$$

Practically, in the examples that we will discuss in later sections, Lagrangian algebras are uniquely determined by their underlying objects (III.7) and thus we will not specify their algebra morphisms explicitly.

Each Lagrangian algebra $\mathcal{A}$ characterizes a one real parameter family of (unitary) topological bosonic boundary conditions. Let $\mathfrak{B}$ be one of these boundary conditions. Then the Lagrangian algebra $\mathcal{A}$ describes how the anyons of $\mathfrak{Z}$ can end along $\mathfrak{B}$. More precisely, $N_a$ describes the dimension of the vector space formed by topological ends of the anyon $a$ on the boundary $\mathfrak{B}$ (with no residual boundary line attached to any of the ends). Since we are considering bosonic topological boundary conditions, all anyons $a$ for which $N_a \neq 0$ in $\mathcal{A}$ are bosons.

The whole family of topological boundary conditions associated to $\mathcal{A}$ can be generated from $\mathfrak{B}$ by stacking an Euler term $\mathfrak{T}_\lambda$ giving rise to boundary conditions

$$\mathfrak{B}_\lambda = \mathfrak{B} \boxtimes \mathfrak{T}_\lambda. \tag{III.9}$$

An Euler term is an invertible bosonic 2d TFT whose partition function of on a 2d closed manifold with genus $g$ is $e^{-\lambda(2-2g)}$. The fusion category formed by topological lines living on $\mathfrak{B}$, that we call $\mathcal{S}$, is determined by the Lagrangian algebra $\mathcal{A}$ as

$$\mathcal{S} = \mathcal{Z}_{\mathcal{A}}, \tag{III.10}$$

where $\mathcal{Z}_{\mathcal{A}}$ is the category of right $\mathcal{A}$-modules in $\mathcal{Z}$ [205]. There is a bulk to boundary map

$$F: \ \mathcal{Z} \to \mathcal{S} = \mathcal{Z}_{\mathcal{A}}, \tag{III.11}$$

taking bulk lines in $\mathcal{Z}$ to boundary lines in $\mathcal{S}$ (simply by stacking them onto the boundary), which is simply

$$F(a) = a \otimes \mathcal{A}, \tag{III.12}$$

viewed as a right $\mathcal{A}$-module.

If we choose $\mathcal{S}$ to be our starting bosonic symmetry, then we can choose $\mathfrak{B}$ to be the symmetry boundary

$$\mathfrak{B}_{\mathcal{S}}^{\mathrm{sym}} = \mathfrak{B}\,, \qquad \text{(III.13)}$$

in which case $\mathcal{A}$ is referred to as the symmetry Lagrangian algebra and denoted as $\mathcal{A}^{\mathrm{sym}}$

$$\mathcal{A}^{\mathrm{sym}} = \mathcal{A}\,. \qquad \text{(III.14)}$$

### D. Fermionic Topological Boundaries

Now we extend the above discussion to fermionic topological boundaries. Such boundaries are characterized by Lagrangian superalgebras in $\mathcal{Z}$ [206], which is an object $\mathcal{A}_f$ in $\mathcal{Z}$

$$\mathcal{A}_f = \bigoplus_{a \in \mathcal{Z}} N_a a, \qquad N_a \in \mathbb{Z}_{\geq 0},\ N_1 = 1, \qquad \text{(III.15)}$$

equipped with a super-commutative multiplication and satisfying the condition

$$\dim(\mathcal{A}_f)^2 = \sum_a \dim(a)^2\,. \qquad \text{(III.16)}$$

Each Lagrangian superalgebra $\mathcal{A}_f$ characterizes a $\mathbb{Z}_2 \times \mathbb{R}$ set of (unitary) topological fermionic boundary conditions of the 3d TFT $\mathfrak{Z}$. Let $\mathfrak{B}_f$ be one of these boundary conditions. The Lagrangian superalgebra $\mathcal{A}_f$ describes how the anyons of $\mathfrak{Z}$ can end along $\mathfrak{B}_f$ with $N_a$ describing the dimension of the vector space formed by topological ends of the anyon $a$ on the boundary $\mathfrak{B}_f$ (with no residual boundary line attached to any of the ends). Since we are considering fermionic topological boundary conditions, all anyons $a$ for which $N_a \neq 0$ in $\mathcal{A}$ are either bosons or fermions, and we can express $\mathcal{A}_f$ as

$$\mathcal{A}_f = \bigoplus_{b \in \mathcal{Z}} N_b b \oplus \bigoplus_{f \in \mathcal{Z}} N_f f\,, \qquad \text{(III.17)}$$

where $b$ and $f$ label bosonic and fermionic anyons respectively. The $N_b$ number of ends of $b$ are all bosonic, and the $N_f$ number of ends of $f$ are all fermionic.

We can alternatively express $\mathcal{A}_f$ as a Lagrangian algebra in the modular tensor supercategory[13] $\mathsf{sVec} \boxtimes \mathcal{Z}$, where the $\mathsf{sVec}$ factor adds the $\pi$ line to the list of bulk topological lines. Note that the $\pi$ line is itself a fermion. Then $\mathcal{A}_f$ can be expressed as

$$\mathcal{A}_f = \bigoplus_{b \in \mathcal{Z}} N_b b \oplus \bigoplus_{f \in \mathcal{Z}} N_f (\pi f)\,, \qquad \text{(III.18)}$$

---

[13] Lagrangian algebras in modular tensor supercategories are known to characterize topological boundary conditions of 3d fermionic TFTs [207–209]. Here we are regarding the bosonic 3d TFT $\mathfrak{Z}$ as a special case of a fermionic 3d TFT whose topological lines are given by modular tensor supercategory $\mathsf{sVec} \boxtimes \mathcal{Z}$ and the fermion parity is trivial $(-1)^F = 1$.

where the terms involving fermions in (III.17) are converted into terms involving bosons in (III.18) by multiplying with the $\pi$ line.

Physically, passing from the expression (III.17) to the expression (III.18) involves converting the fermionic ends of $f$ along $\mathfrak{B}_f$ into bosonic ends of $\pi f$ along $\mathfrak{B}_f$ by stacking each fermionic end of $f$ with the canonical fermionic end $\mathcal{O}_\pi$ of the $\pi$ line

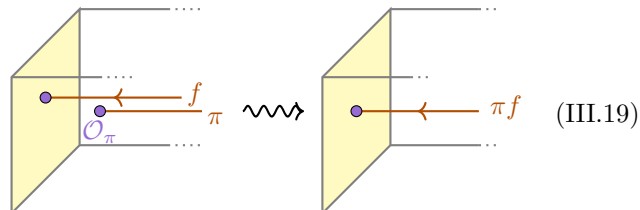

$$\text{(III.19)}$$

Thus the expression (III.18) captures the information of all bosonic topological ends of bulk topological lines along $\mathfrak{B}_f$. In the examples appearing in this paper, we will use the expression (III.18) for $\mathcal{A}_f$, and refer to such an $\mathcal{A}_f$ as a **fermionic Lagrangian algebra** in $\mathcal{Z}$.

The other topological boundary conditions described by $\mathcal{A}_f$ can be generated from $\mathfrak{B}_f$ by adding Euler and Arf terms and can be expressed as

$$\mathfrak{B}_{f,\lambda} = \mathfrak{B}_f \boxtimes \mathfrak{T}_\lambda, \qquad \mathfrak{B}_{f,\lambda} \boxtimes \mathrm{Arf}\,. \qquad \text{(III.20)}$$

The fusion $\pi$-supercategory formed by topological lines living on $\mathfrak{B}_f$, that we call $\mathcal{S}_f$, is determined by the fermionic Lagrangian algebra $\mathcal{A}_f$ as

$$\mathcal{S}_f = (\mathsf{sVec} \boxtimes \mathcal{Z})_{\mathcal{A}_f}\,, \qquad \text{(III.21)}$$

where $(\mathsf{sVec} \boxtimes \mathcal{Z})_{\mathcal{A}_f}$ is the category of right $\mathcal{A}_f$-modules in $\mathsf{sVec} \boxtimes \mathcal{Z}$. There is a bulk to boundary map

$$F :\ \mathsf{sVec} \boxtimes \mathcal{Z} \to \mathcal{S}_f\,, \qquad \text{(III.22)}$$

taking bulk lines in $\mathsf{sVec} \boxtimes \mathcal{Z}$ to boundary lines in $\mathcal{S}_f$ (simply by stacking them onto the boundary) that is simply

$$F(a) = a \otimes \mathcal{A}_f\,, \qquad \text{(III.23)}$$

viewed as a right $\mathcal{A}_f$-module.

We expect that any $\mathcal{S}_f$ obtained in this way has exactly two special simple objects (upto bosonic isomorphisms) $P_f$ and $\pi P_f$, whose corresponding boundary lines on $\mathfrak{B}_f$ act trivially on the topological ends of bosonic anyons $b$ in $\mathcal{A}_f$, and by a non-trivial sign on the topological ends of fermionic anyons $f$ in $\mathcal{A}_f$. This action is encoded in the fact that the half-braiding of any such $b$ with $P_f$ and $\pi P_f$ is $+1$ and the half-braiding of any such $f$ with $P_f$ and $\pi P_f$ is $-1$. The precise choice $\mathfrak{B}_f$ of fermionic topological boundary condition corresponds to choosing either $P_f$ or $\pi P_f$ as the fermionic parity symmetry $(-1)^F$. Without loss of generality, let's say $(-1)^F = P_f$ is the choice corresponding to $\mathfrak{B}_f$. Then, the other choice $(-1)^F = \pi P_f$ corresponds to the fermionic topological boundary $\mathfrak{B}_f \boxtimes \mathrm{Arf}$.

If we choose $\mathcal{S}_f$ with $(-1)^F = P_f$ to be our starting fermionic symmetry, then we can choose $\mathfrak{B}_f$ to be the symmetry boundary

$$\mathfrak{B}_{\mathcal{S}_f}^{\text{sym}} = \mathfrak{B}_f, \tag{III.24}$$

in which case $\mathcal{A}_f$ is referred to as the symmetry fermionic Lagrangian algebra and denoted as $\mathcal{A}_f^{\text{sym}}$

$$\mathcal{A}_f^{\text{sym}} = \mathcal{A}_f. \tag{III.25}$$

### E. Examples

#### 1. $\mathbb{Z}_2^f$ Symmetry

The symmetry TFT for the $\mathbb{Z}_2^f$ symmetry is the $\mathbb{Z}_2$ (untwisted) Dijkgraaf-Witten theory [210], which describes the low energy limit of the Toric Code Hamiltonian [211]. The anyon content of this TFT is described by the Drinfeld center of $\mathsf{Vec}_{\mathbb{Z}_2}$, which has four simple objects

$$\mathcal{Z}(\mathsf{Vec}_{\mathbb{Z}_2}) = \{1, e, m, f\}. \tag{III.26}$$

Here, $f = em$ is a fermion, while the other three anyons are bosons.

The bosonic Lagrangian algebras are

$$\mathcal{A}_e = 1 \oplus e, \qquad \mathcal{A}_m = 1 \oplus m, \tag{III.27}$$

both of which describe topological boundary conditions carrying non-anomalous bosonic $\mathbb{Z}_2$ symmetry.

There is a single non-bosonic fermionic Lagrangian algebra[14] in $\mathcal{Z}(\mathsf{Vec}_{\mathbb{Z}_2})$ given by

$$\mathcal{A}_f = 1 \oplus \pi f. \tag{III.28}$$

Let $\mathfrak{B}_f$ be a topological boundary associated to $\mathcal{A}_f$. Then we have the bulk to boundary map given by[15]

$$\begin{aligned} F(1) &= 1, & F(e) &= \pi P, \\ F(m) &= P, & F(f) &= \pi, \end{aligned} \tag{III.29}$$

where $P$ generates a $\mathbb{Z}_2$ symmetry on the boundary, i.e, $P^2 = 1$, because $e$ obeys the $\mathbb{Z}_2$ group-like fusion rule and $F$ preserves the tensor product structure.

This implies that the fusion $\pi$-supercategory formed by lines living on $\mathfrak{B}_f$ is

$$\mathcal{S}_f = \mathsf{sVec}_{\mathbb{Z}_2} \tag{III.30}$$

---

[14] From now on, when discussing examples, by a fermionic Lagrangian algebra we will mean a non-bosonic fermionic Lagrangian algebra.

[15] Here and in what follows, bosonic isomorphisms are written as equalities.

which as discussed earlier describes the fermion parity symmetry. There are two possible choices for the fermion parity line:

$$(-1)^F = P, \quad \text{or} \quad (-1)^F = \pi P. \tag{III.31}$$

Let us choose $\mathfrak{B}_f$ such that the choice

$$(-1)^F = P \tag{III.32}$$

is realized. Then, such a $\mathfrak{B}_f$ is obtained as fermionization of a topological boundary $\mathfrak{B}_e$ associated to $\mathcal{A}_e$. To see this note that an end of $m$ on $\mathfrak{B}_f$ is a bosonic topological local operator attached to the boundary line $P = (-1)^F$, or in other words the end of $m$ is a boson in R-sector. Recall from (II.29) that the bosonization of such an operator is a twisted sector operator for the $\mathbb{Z}_2$ symmetry which is uncharged under the $\mathbb{Z}_2$ symmetry [167, 212]. Indeed, an end of $m$ along $\mathfrak{B}_e$ has precisely these properties. One could derive this conclusion also by considering instead an end of $m$ on $\mathfrak{B}_f$, which is a bosonic topological local operator attached to the boundary line $\pi(-1)^F$, or equivalently a fermionic topological local operator attached to the boundary line $(-1)^F$. In other words, the end of $e$ is a fermion in R-sector. The bosonization of such an operator is an untwisted sector operator which is charged under the $\mathbb{Z}_2$ symmetry [167, 212]. Indeed the $e$ line ends on $\mathfrak{B}_e$ without being attached to a boundary line and such an end is charged under the $\mathbb{Z}_2$ symmetry living on $\mathfrak{B}_e$.

On the other hand, the boundary

$$\widetilde{\mathfrak{B}}_f := \mathfrak{B}_f \boxtimes \text{Arf} \tag{III.33}$$

corresponds to the choice

$$(-1)^F = \pi P. \tag{III.34}$$

By the same arguments as above, we see that $\widetilde{\mathfrak{B}}_f$ is obtained as fermionization of a topological boundary $\mathfrak{B}_m$ associated to $\mathcal{A}_m$.

#### 2. $\mathbb{Z}_4^f$ and $\mathbb{Z}_4^{\pi f}$ Symmetries

The symmetry TFT for $\mathbb{Z}_4^f$ and $\mathbb{Z}_4^{\pi f}$ symmetries is the $\mathbb{Z}_4$ (untwisted) Dijkgraaf-Witten theory, whose anyon content is described by the Drinfeld center of $\mathsf{Vec}_{\mathbb{Z}_4}$:

$$\mathcal{Z}(\mathsf{Vec}_{\mathbb{Z}_4}) = \{e^a m^b \mid a, b = 0, 1, 2, 3\}. \tag{III.35}$$

Here, both $e$ and $m$ are bosons and they have non-trivial mutual braiding $i$. The spin of $e^a m^b$ is given by $\theta_{e^a m^b} = i^{ab}$.

There are three bosonic Lagrangian algebras

$$\begin{aligned} \mathcal{A}_e &= 1 \oplus e \oplus e^2 \oplus e^3, \\ \mathcal{A}_m &= 1 \oplus m \oplus m^2 \oplus m^3, \\ \mathcal{A}_{e^2, m^2} &= 1 \oplus e^2 \oplus m^2 \oplus e^2 m^2, \end{aligned} \tag{III.36}$$

and two fermionic Lagrangian algebras

$$\mathcal{A}_{em^2} = 1 \oplus \pi em^2 \oplus e^2 \oplus \pi e^3 m^2,$$
$$\mathcal{A}_{e^2 m} = 1 \oplus \pi e^2 m \oplus m^2 \oplus \pi e^2 m^3. \qquad \text{(III.37)}$$

The bosonic symmetries corresponding to $\mathcal{A}_e$ and $\mathcal{A}_m$ are both non-anomalous $\mathbb{Z}_4$, and the one corresponding to $\mathcal{A}_{e^2,m^2}$ is $\mathbb{Z}_2 \times \mathbb{Z}_2$ with mixed 't Hooft anomaly. On the other hand, the fusion $\pi$-supercategory corresponding to both $\mathcal{A}_{em^2}$ and $\mathcal{A}_{e^2 m}$ is $\mathsf{sVec}_{\mathbb{Z}_4}$. The bulk-to-boundary functor for $\mathcal{A}_{em^2}$ is given by

$$F(e) = \pi P^2, \quad F(m) = P, \qquad \text{(III.38)}$$

where $P$ generates a $\mathbb{Z}_4$ symmetry on the boundary, i.e., $P^4 = 1$, which is determined using $F(e^2) = 1$ and $F(em^2) = \pi$ following from the expression for $\mathcal{A}_{em^2}$. The bulk-to-boundary functor for the other fermionic Lagrangian algebra $\mathcal{A}_{e^2 m}$ is obtained by exchanging $e$ and $m$.

As in the previous example, there are two choices for the fermion parity line, namely,

$$(-1)^F = P^2, \quad \text{or} \quad (-1)^F = \pi P^2. \qquad \text{(III.39)}$$

The former choice leads to $\mathbb{Z}_4^f$ symmetry, while the latter choice leads to $\mathbb{Z}_4^{\pi f}$ symmetry.

A fermionic boundary $\left(\mathcal{A}_{em^2}, (-1)^F = P^2\right)$ carrying $\mathbb{Z}_4^f$ symmetry is obtained as fermionization of bosonic boundary $\mathcal{A}_e$, while $\left(\mathcal{A}_{em^2}, (-1)^F = \pi P^2\right)$ carrying $\mathbb{Z}_4^{\pi f}$ symmetry is obtained as fermionization using one of the $\mathbb{Z}_2$s on the bosonic boundary $\mathcal{A}_{e^2,m^2}$. These two fermionic boundaries are related by stacking with the Arf TFT. Similarly, a fermionic boundary $\left(\mathcal{A}_{e^2 m}, (-1)^F = P^2\right)$ carrying $\mathbb{Z}_4^f$ symmetry is obtained as fermionization of bosonic boundary $\mathcal{A}_m$, while $\left(\mathcal{A}_{e^2 m}, (-1)^F = \pi P^2\right)$ carrying $\mathbb{Z}_4^{\pi f}$ symmetry is obtained as fermionization using the other $\mathbb{Z}_2$ on the bosonic boundary $\mathcal{A}_{e^2,m^2}$. Again, these two fermionic boundaries are related by stacking with the Arf TFT.

### 3. $\mathsf{Rep}(S_3)^f$ and $\mathsf{Rep}(S_3)^{\pi f}$ Symmetries

The symmetry TFT for $\mathsf{Rep}(S_3)^f$ and $\mathsf{Rep}(S_3)^{\pi f}$ symmetries is the $S_3$ (untwisted) Dijkgraaf-Witten theory. The anyons of this TFT are labeled by pairs $([g], R)$, where $[g]$ is a conjugacy class in $S_3 = \{a^i b^j \mid i = 0, 1, 2 \quad j = 0, 1, \quad ba = a^2 b\}$ and $R$ is an irreducible representation of the centralizer subgroup $C(g) = \{h \in S_3 \mid gh = hg\}$ of an element $g \in [g]$. Concretely, the anyon content of the $S_3$ Dijkgraaf-Witten theory is given by

$$\begin{array}{lll} (1,1), & (1,P), & (1,E), \\ (a,1), & (a,\omega), & (a,\omega^2), \\ (b,+), & (b,-), \end{array} \qquad \text{(III.40)}$$

where $P$ and $E$ are non-trivial one-dimensional and two-dimensional irreducible representations of $S_3$ respectively, $\omega$ denotes a non-trivial one-dimensional representation of $C(a) \cong \mathbb{Z}_3$, and $-$ denotes the sign representation of $C(b) \cong \mathbb{Z}_2$. In the above equation, the conjugacy class of $g$ is simply written as $g$ by abuse of notation.

There are four bosonic Lagrangian algebras

$$\begin{aligned} \mathcal{A}_{P,E} &= (1,1) \oplus (1,P) \oplus 2(1,E), \\ \mathcal{A}_{P,a} &= (1,1) \oplus (1,P) \oplus 2(a,1), \\ \mathcal{A}_{E,b} &= (1,1) \oplus (1,E) \oplus (b,+), \\ \mathcal{A}_{a,b} &= (1,1) \oplus (a,1) \oplus (b,+), \end{aligned} \qquad \text{(III.41)}$$

and two fermionic Lagrangian algebras

$$\begin{aligned} \mathcal{A}_{b-,E} &= (1,1) \oplus \pi(b,-) \oplus (1,E), \\ \mathcal{A}_{b-,a} &= (1,1) \oplus \pi(b,-) \oplus (a,1). \end{aligned} \qquad \text{(III.42)}$$

$\mathcal{A}_{P,E}$ and $\mathcal{A}_{P,a}$ describe $S_3$ symmetry, while $\mathcal{A}_{E,b}$ and $\mathcal{A}_{a,b}$ describe $\mathsf{Rep}(S_3)$ symmetry. On the other hand, $\mathcal{A}_{b-,E}$ describes $\mathsf{Rep}(S_3)^f$ and $\mathsf{Rep}(S_3)^{\pi f}$ symmetries depending on the choice of $(-1)^F$, and $\mathcal{A}_{b-,a}$ also describes $\mathsf{Rep}(S_3)^f$ and $\mathsf{Rep}(S_3)^{\pi f}$ symmetries.

The bulk-to-boundary functor for $\mathcal{A}_{b-,E}$ is given by

$$\begin{aligned} F(1,1) &= 1, & F(1,P) &= P, \\ F(1,E) &= 1 \oplus P, & F(a,\omega^p) &= E, \\ F(b,+) &= \pi P \oplus \pi E, & F(b,-) &= \pi \oplus \pi E, \end{aligned} \qquad \text{(III.43)}$$

where $p \in \{0, 1, 2\}$, and the objects $P$ and $E$ on the RHS are the simple objects of the fusion $\pi$-supercategory $\mathsf{sVec} \boxtimes \mathsf{Rep}(S_3)$ formed by topological lines living on a topological boundary associated to $\mathcal{A}_{b-,E}$. These boundary lines have the fusion rules described in (II.42). The above equations (III.43) and the fusion rules can be derived from the fact that (1) $F$ preserves the tensor product structure and (2) the condensed anyons are those in $\mathcal{A}_{b-,E}$. The bulk-to-boundary functor for the other fermionic Lagrangian algebra $\mathcal{A}_{b-,a}$ is obtained by exchanging $(1,E)$ and $(a,1)$. Possible choices for the fermion parity line are

$$(-1)^F = P, \quad \text{or} \quad (-1)^F := \pi P. \qquad \text{(III.44)}$$

The former choice leads to $\mathsf{Rep}(S_3)^f$ symmetry, while the latter choice leads to $\mathsf{Rep}(S_3)^{\pi f}$ symmetry [9].

The bosonizations of these genuinely fermionic boundaries are as follows:

$$\begin{aligned} \left(\mathcal{A}_{b-,E}, (-1)^F = P\right) &\longrightarrow \mathcal{A}_{E,b}, \\ \left(\mathcal{A}_{b-,a}, (-1)^F = P\right) &\longrightarrow \mathcal{A}_{a,b}, \\ \left(\mathcal{A}_{b-,E}, (-1)^F = \pi P\right) &\longrightarrow \mathcal{A}_{P,E}, \\ \left(\mathcal{A}_{b-,a}, (-1)^F = \pi P\right) &\longrightarrow \mathcal{A}_{P,a}. \end{aligned} \qquad \text{(III.45)}$$

This is quickly seen for $\mathcal{A}_{b-,E}$ by analyzing $F(1,E) = 1 \oplus P$. In one case, this becomes $F(1,E) = 1 \oplus (-1)^F$, meaning that there are two topological ends of $(1,E)$

along the boundary: one of them is an NS-sector boson, while the other is an R-sector boson. Bosonizing them we obtain an untwisted sector operator uncharged under dual $\mathbb{Z}_2$ and a $\mathbb{Z}_2$-twisted sector operator uncharged under dual $\mathbb{Z}_2$. These are precisely the properties of the ends of $(1, E)$ along $\mathcal{A}_{E,b}$ [3]. In the other case, this becomes $F(1, E) = 1 \oplus \pi (-1)^F$, meaning that there are two topological ends of $(1, E)$ along the boundary: one of them is an NS-sector boson, while the other is an R-sector fermion. Bosonizing them we obtain an untwisted sector operator uncharged under dual $\mathbb{Z}_2$ and an untwisted sector operator charged under dual $\mathbb{Z}_2$. These are precisely the properties of the ends of $(1, E)$ along $\mathcal{A}_{P,E}$. For determining bosonizations of $\mathcal{A}_{b-,a}$, we simply interchange $(1, E)$ and $(a, 1)$ in the above argument.

#### 4. $\mathbb{Z}_2 \times \mathbb{Z}_2^f$ Symmetry with a Gu-Wen Anomaly

The symmetry TFT for $\mathbb{Z}_2 \times \mathbb{Z}_2^f$ symmetry with a Gu-Wen anomaly $\nu = 2 \bmod 4$ is the $\mathbb{Z}_4$ Dijkgraaf-Witten theory with a twist $\omega = 2 \in H^3(\mathbb{Z}_4, U(1)) \cong \mathbb{Z}_4$. The anyon content of this TFT is described by the Drinfeld center $\mathcal{Z}(\mathsf{Vec}_{\mathbb{Z}_4}^\omega)$ of $\mathsf{Vec}_{Z_4}^\omega$:

$$\mathcal{Z}(\mathsf{Vec}_{\mathbb{Z}_4}^\omega) = \{e^a m^b \mid a, b = 0, 1, 2, 3\}. \quad \text{(III.46)}$$

Here, $e$ is a boson, while $m$ is an abelian anyon with spin $\theta_m = e^{i\pi/4}$. The mutual statistics between $e$ and $m$ is $i$. Thus, the spin of anyon $e^a m^b$ is

$$\theta_{e^a m^b} = e^{i\pi b(2a+b)/4}. \quad \text{(III.47)}$$

There are two bosonic Lagrangian algebras

$$\begin{aligned} \mathcal{A}_e &= 1 \oplus e \oplus e^2 \oplus e^3, \\ \mathcal{A}_{em^2} &= 1 \oplus em^2 \oplus e^2 \oplus e^3 m^2. \end{aligned} \quad \text{(III.48)}$$

The symmetry categories on the corresponding bosonic boundaries are both $\mathbb{Z}_4$ with an anomaly $\omega = 2 \in H^3(\mathbb{Z}_4, U(1)) = \mathbb{Z}_4$.

There is also a unique fermionic Lagrangian algebra

$$\mathcal{A}_{m^2, e^2} = 1 \oplus \pi m^2 \oplus e^2 \oplus \pi e^2 m^2. \quad \text{(III.49)}$$

The bulk-to-boundary functor for the fermionic Lagrangian algebra $\mathcal{A}_{m^2, e^2}$ is

$$F(e) = P, \quad F(m) = \eta, \quad \text{(III.50)}$$

with the fusion rules

$$P^2 = 1, \quad \eta^2 = \pi. \quad \text{(III.51)}$$

The above fusion rules imply that the symmetry category on the fermionic boundary is $\mathbb{Z}_2 \times \mathbb{Z}_2^f$ with a Gu-Wen anomaly, cf. eq. (II.46). There are two choices for the fermion parity line, i.e.,

$$(-1)^F = P, \quad \text{or} \quad (-1)^F = \pi P, \quad \text{(III.52)}$$

both of which lead to the same fermionic symmetry because a $\mathbb{Z}_2 \times \mathbb{Z}_2^f$ symmetry with a Gu-Wen anomaly is invariant under stacking the Arf TFT as we discussed in Section II E 4. Fermionic boundary $(\mathcal{A}_{e^2, m^2}, (-1)^F = P)$ is the fermionization of bosonic boundary $\mathcal{A}_{em^2}$, while the other fermionic boundary $(\mathcal{A}_{e^2, m^2}, (-1)^F = \pi P)$ is the fermionization of bosonic boundary $\mathcal{A}_e$. Thus, from the point of view of SymTFT the two fermionic boundaries related by stacking of Arf TFT are also related by the action of a 0-form symmetry of the 3d TFT acting on anyons as

$$e \to em^2, \quad m \to m \quad \text{(III.53)}$$

which also explains why stacking with Arf TFT does not change the fermionic symmetry.

#### 5. $\mathbb{Z}_2 \times \mathbb{Z}_2^f$ Symmetry with a beyond Gu-Wen Anomaly

The symmetry TFT for $\mathbb{Z}_2 \times \mathbb{Z}_2^f$ symmetry with a beyond Gu-Wen anomaly $\nu = 1, 7 \bmod 8$ is the doubled Ising TFT, whose anyon content is described by

$$\mathcal{Z}(\mathsf{Ising}_+) = \mathsf{Ising} \boxtimes \overline{\mathsf{Ising}}, \quad \text{(III.54)}$$

where $\mathsf{Ising} = \{1, \psi, \sigma\}$ denotes the modular tensor category describing the anyons of the Ising TFT, and $\overline{\mathsf{Ising}} = \{\overline{1}, \overline{\psi}, \overline{\sigma}\}$ is the Ising category with the opposite braiding statistics. The fusion rules of these anyons are given by

$$\sigma^2 = 1 \oplus \psi, \quad \psi\sigma = \sigma\psi = \sigma. \quad \text{(III.55)}$$

This TFT has a unique bosonic Lagrangian algebra

$$\mathcal{A}_{\sigma\overline{\sigma}} = 1\overline{1} \oplus \psi\overline{\psi} \oplus \sigma\overline{\sigma}. \quad \text{(III.56)}$$

The symmetry category on the corresponding bosonic boundary is $\mathsf{Ising}_+$.

The above TFT also has a unique fermionic Lagrangian algebra given by

$$\mathcal{A}_{\psi, \overline{\psi}}^f = 1 \oplus \pi\psi \oplus \pi\overline{\psi} \oplus \psi\overline{\psi}. \quad \text{(III.57)}$$

The bulk-to-boundary functor for this fermionic Lagrangian algebra $\mathcal{A}_{\psi, \overline{\psi}}^f$ maps the bulk anyons to

$$\begin{aligned} F(\psi) &= F(\overline{\psi}) = \pi, \\ F(\sigma) &= q, \quad F(\overline{\sigma}) = \overline{q}, \end{aligned} \quad \text{(III.58)}$$

which obey the following fusion rules:

$$q^2 = \overline{q}^2 = 1 \oplus \pi, \quad \pi q = q\pi = q, \quad \pi\overline{q} = \overline{q}\pi = \overline{q}. \quad \text{(III.59)}$$

The above fusion rules imply that both $q$ and $\overline{q}$ are q-type objects whose quantum dimensions are $\dim(q) = \dim(\overline{q}) = \sqrt{2}$.

To see the relation between $q$ and $\bar{q}$, we consider their fusion $q\bar{q}$. First of all, $q\bar{q}$ cannot be a simple object because $\mathrm{Hom}(q\bar{q}, q\bar{q})$ is neither $\mathbb{C}^{1|0}$ nor $\mathbb{C}^{1|1}$:

$$\mathrm{Hom}(q\bar{q}, q\bar{q}) \supset \mathrm{Hom}(q,q) \otimes \mathrm{Hom}(\bar{q},\bar{q}) = \mathbb{C}^{1|1} \otimes \mathbb{C}^{1|1} \cong \mathbb{C}^{2|2}. \tag{III.60}$$

Furthermore, since the quantum dimension of $q\bar{q}$ is $\dim(q\bar{q}) = \dim(q)\dim(\bar{q}) = 2$, it has to be a direct sum of two m-type objects of quantum dimension 1.[16] In addition, since $q\bar{q}$ has a fermionic automorphism, we find

$$q\bar{q} = P \oplus \pi P, \tag{III.61}$$

where $P$ is an m-type object with $\dim(P) = 1$. By multiplying $q$ on both sides of the above equation, we obtain

$$\bar{q} = qP = Pq. \tag{III.62}$$

This equation together with the fusion rules (III.59) implies that $P^2$ is either 1 or $\pi$. Since the symmetry category should contain a non-anomalous fermion parity symmetry, we conclude

$$P = (-1)^F, \quad \text{or} \quad P = \pi(-1)^F, \tag{III.63}$$

which is consistent only with $P^2 = 1$.

Summarizing, we find that the symmetry category on a boundary associated with the fermionic Lagrangian algebra (III.57) consists of topological lines $\{1, \pi, P, \pi P = P\pi, q, Pq = qP\}$ that obey the following fusion rules:

$$P^2 = 1, \quad q^2 = 1 \oplus \pi. \tag{III.64}$$

This shows that the fermionic boundary of the doubled Ising TFT indeed realizes $\mathbb{Z}_2 \times \mathbb{Z}_2^f$ symmetry with a beyond Gu-Wen anomaly. The underlying category of the subcategory $\mathcal{S}_q$ consisting only of $\{1, \pi, q\}$ is equivalent to the Ising fusion category $\mathsf{Ising}_+$ because $q$ originates from the Ising anyon $\sigma$ in the bulk. Similarly, the underlying category of the subcategory $\mathcal{S}_{\bar{q}}$ consisting only of $\{1, \pi, \bar{q} = Pq\}$ is also equivalent to $\mathsf{Ising}_+$ as a fusion category. Therefore, the anomaly of this $\mathbb{Z}_2 \times \mathbb{Z}_2^f$ symmetry is $\nu = 1, 7 \mod 8$, cf. Section II E 5.

We note that there are two choices for the fermion parity line as shown in eq. (III.63). These two choices lead to the same fermionic boundary. In other words, the fermionic boundary associated with fermionic Lagrangian algebra $\mathcal{A}^f_{\psi,\bar{\psi}}$ is invariant under stacking with the Arf TFT. This invariance follows from the fact that its bosonization $\mathcal{A}_{\sigma\bar{\sigma}}$ is invariant under gauging the $\mathbb{Z}_2$ subgroup of $\mathsf{Ising}_+$ symmetry on the boundary.

When the anomaly of $\mathbb{Z}_2 \times \mathbb{Z}_2^f$ symmetry is $\nu = 3, 5 \mod 8$, the symmetry TFT is the Drinfeld center of $\mathsf{Ising}_-$

$$\begin{aligned} \mathcal{Z}(\mathsf{Ising}_-) &\cong \mathcal{Z}(\mathsf{Rep}(\mathrm{SU}(2)_2)) \\ &\cong \mathsf{Rep}(\mathrm{SU}(2)_2) \boxtimes \overline{\mathsf{Rep}(\mathrm{SU}(2)_2)} \end{aligned} \tag{III.65}$$

rather than the doubled Ising TFT $\mathcal{Z}(\mathsf{Ising}_+)$. This TFT has a unique bosonic Lagrangian algebra (III.56), which describes a bosonic topological boundary with symmetry category $\mathsf{Ising}_-$. Similarly, the above TFT also has a unique fermionic Lagrangian algebra (III.57). The symmetry category on the corresponding fermionic boundary is determined in the same way as above. In particular, the topological lines on the fermionic boundary have the same fusion rules as eq. (III.64). However, the underlying categories of the subcategories $\mathcal{S}_q$ and $\mathcal{S}_{\bar{q}}$ are equivalent to $\mathsf{Ising}_- \cong \mathsf{Rep}(\mathrm{SU}(2)_2)$ rather than $\mathsf{Ising}_+$. Thus, the symmetry category on the fermionic boundary of $\mathcal{Z}(\mathsf{Ising}_-)$ is indeed $\mathbb{Z}_2 \times \mathbb{Z}_2^f$ with a beyond Gu-Wen anomaly $\nu = 3, 5 \mod 8$.

## IV. GENERALIZED CHARGES

### A. General Setup

The charges of a symmetry are encoded by bulk anyons of its associated SymTFT. This applies to both bosonic and fermionic symmetries. Since the bosonic case is a special subcase of the fermionic case, we treat everything fermionically in what follows, extending the detailed presentation of the bosonic case appearing in [2].

Let $\mathcal{S}_f$ be a fermionic symmetry. A simple anyon $a$ of the SymTFT $\mathfrak{Z}(\mathcal{S}_f)$ describes an irreducible multiplet of local operators charged under $\mathcal{S}_f$. Here irreducibility of the multiplet means that the existence of any of the local operators lying in such a multiplet guarantees the existence of all other local operators participating in the multiplet.

The different local operators in the multiplet correspond to topological ends $\mathcal{E}_a^Y$ of the anyon $a$ along the symmetry boundary $\mathfrak{B}_{\mathcal{S}_f}^{\mathrm{sym}}$ attached to a simple boundary topological line $Y \in \mathcal{S}_f$. All the possible local operators in the multiplet are obtained by spanning over all simple $Y$ and all possible $\mathcal{E}_a^Y$.

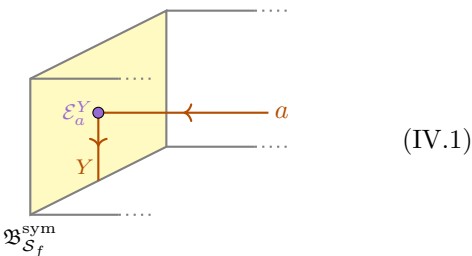

$$\tag{IV.1}$$

Let $\mathfrak{T}_f$ be an $\mathcal{S}_f$-symmetric 2d theory and $\mathcal{M}$ a multiplet[17] of local operators in $\mathfrak{T}_f$ carrying charge $a$. The multiplet $\mathcal{M}$ corresponds to a local operator $\mathcal{E}_\mathcal{M}$ lying at the end of the anyon $a$ along the physical boundary

---

[16] We recall that the quantum dimension of a q-type object cannot be less than $\sqrt{2}$.

[17] This should not be confused with a $\mathcal{S}$-module category, which was also denoted by $\mathcal{M}$ above.

$\mathfrak{B}^{\mathrm{phys}}_{\mathfrak{T}_f}$. The local operators $\mathcal{O}^Y_a$ of $\mathfrak{T}_f$ in the multiplet $\mathcal{M}$ are constructed using the sandwich construction where the anyon $a$ stretches between the symmetry and physical boundaries, ending on the symmetry boundary in $\mathcal{E}^Y_a$ and on the physical boundary in $\mathcal{E}_{\mathcal{M}}$ (See IV.2). An operator $\mathcal{O}^Y_a$ arising from an end $\mathcal{E}^Y_a$ lives in $Y$-twisted sector, i.e. it is attached to the topological line implementing the symmetry $Y$ of $\mathfrak{T}_f$. The operators $\mathcal{O}^Y_a$ are all topological or non-topological if $\mathcal{E}_{\mathcal{M}}$ is topological or non-topological respectively.

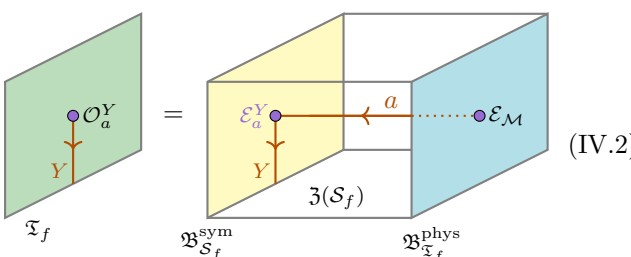

$$(\mathrm{IV.2})$$

The symmetry $\mathcal{S}_f$ acts on the local operators $\mathcal{O}^Y_a$ by linking. A linking action of a simple line $X \in \mathcal{S}_f$ on $Y$-twisted sector operator $\mathcal{O}^Y_a$ can land in $Y'$-twisted sector. Such an action is specified by a choice of a topological local operator in $\mathcal{S}_f$ lying at a junction where the line $X$ crosses $Y$ converting it into $Y'$, as in

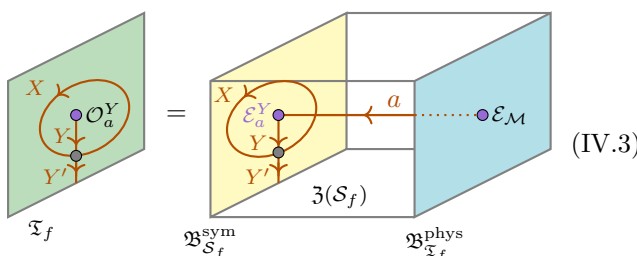

$$(\mathrm{IV.3})$$

Squeezing the $X$-loop on top of the operator $\mathcal{O}^Y_a$ results in a linear combination of local operators $\mathcal{O}^{Y'}_a$, which is the result of such a linking action on $\mathcal{O}^Y_a$.

Such a linking action is entirely captured by the SymTFT as it lifts to a linking action of topological line $X$ living on symmetry boundary $\mathfrak{B}^{\mathrm{sym}}_{\mathcal{S}_f}$ on the topological end $\mathcal{E}^Y_a$ of $a$ along $\mathfrak{B}^{\mathrm{sym}}_{\mathcal{S}_f}$. In particular, the action does not depend on the choice of physical boundary. As discussed in detail in [2], such a linking action can be computed in terms of the half-braiding of the anyon $a$ with the boundary line $X$, which is encoded in the realization of $a$ as an object of the Drinfeld center $\mathcal{Z}(\mathcal{S}_f)$ of $\mathcal{S}_f$.

In this way, the charges of local operators under a symmetry $\mathcal{S}_f$ are captured by anyons of the SymTFT $\mathfrak{Z}(\mathcal{S}_f)$. The fusions of the anyons describe the possible charges that can be carried by operators arising in the operator product expansion (OPE) of two charged operators. On the other hand, the braiding of anyons describes the mutual non-locality between two charged local operators, i.e. the changes in correlation functions induced by moving a charged operator around another charged operator in a circle. Thus the whole structure of the modular tensor category $\mathcal{Z}(\mathcal{S}_f)$ formed by the anyons is crucial for the description of generalized charges of the symmetry $\mathcal{S}_f$.

### B. Examples

#### 1. $\mathbb{Z}^f_2$ Symmetry

The SymTFT and symmetry boundaries are discussed in section III E 1. Taking $\mathfrak{B}_f$ to be the symmetry boundary, the generalized charge interpretation of the SymTFT anyons is:

- $1$ describes a bosonic local operator in the untwisted sector (NS sector).

- $e$ describes a fermionic local operator in the $(-1)^F$-twisted sector (R sector).

- $m$ describes a bosonic local operator in the $(-1)^F$-twisted sector (R sector).

- $f$ describes a fermionic local operator in the untwisted sector (NS sector).

This is straightforwardly obtained by applying the fermionization map (II.29) to the generalized charge interpretation of the anyons for a boundary corresponding to the Lagrangian algebra $\mathcal{A}_e$. Here, we recall that the choice of $(-1)^F$ on $\mathfrak{B}_f$ is fixed as in eq. (III.32). If we take $\mathfrak{B}_f \boxtimes$ Arf to be the physical boundary instead, the roles of $e$ and $m$ are exchanged, i.e., $e$ describes a bosonic local operator in the $(-1)^F$-twisted secor and $m$ describes a fermionic local operator in the $(-1)^F$-twisted sector.

#### 2. $\mathbb{Z}^f_4$ and $\mathbb{Z}^{\pi f}_4$ Symmetries

We choose symmetry boundaries for these symmetries to be given by the fermionic Lagrangian algebra $\mathcal{A}_{em^2}$, cf. section III E 2. The anyons of the $\mathbb{Z}_4$ Dijkgraaf-Witten gauge theory, which is the SymTFT, act as the following generalized charges for both $\mathbb{Z}^f_4$ and $\mathbb{Z}^{\pi f}_4$ symmetries:

- $e$ describes a $P^2$-twisted sector (R sector) local operator with charge 1 (mod 4) under $P$. In particular, such an operator is charged under $P^2$ and $\pi P^2$,[18] and hence is a fermion for both cases $\mathbb{Z}^f_4$ and $\mathbb{Z}^{\pi f}_4$.

- $m$ describes a $P$-twisted sector local operator which is uncharged under $P$. In particular, such an operator is a boson.

---

[18] Adding a $\pi$ line does not change the charge.

- $e^i m^j$ describes a $P^{2i+j}$-twisted sector local operator with charge $i$ (mod 4) under $P$. Such an operator is a boson if $i$ is even and a fermion if $i$ is odd.

Note that from the point of view of $P$, the above generalized charges have a uniform description irrespective of whether we are working with $\mathbb{Z}_4^f$ symmetry or $\mathbb{Z}_4^{\pi f}$ symmetry. However, the charges differ from the point of view of $(-1)^F$ subsymmetry of these two symmetries. For example, consider the $e$ charge. For $\mathbb{Z}_4^f$ symmetry, it describes a *fermionic* local operator in $(-1)^F$-twisted sector. On the other hand, for $\mathbb{Z}_4^{\pi f}$ symmetry, it describes a fermionic local operator in $\pi(-1)^F$-twisted sector, which is equivalent to a *bosonic* local operator in $(-1)^F$-twisted sector.

These results can be easily derived from fermionization. For example, consider a fermionic boundary associated to $\mathcal{A}_{em^2}$ which carries $\mathbb{Z}_4^f$ symmetry. As we discussed, it is obtained by fermionizing a bosonic boundary associated to $\mathcal{A}_e$ carrying bosonic $\mathbb{Z}_4$ symmetry. From the point of view of this $\mathbb{Z}_4$ symmetry, the $e$ anyon describes an untwisted sector local operator carrying charge 1 (mod 4) under the $\mathbb{Z}_4$ symmetry generator $P$. The charge remains preserved through fermionization, but the operator after fermionization has to be an R-sector operator, cf (II.29). On the other hand, the $m$ anyon describes a $P$-twisted sector local operator carrying charge 0 (mod 4) under the $\mathbb{Z}_4$ symmetry generator $P$, which are properties that remain preserved through fermionization.

### 3. $\mathsf{Rep}(S_3)^f$ Symmetry

We take the symmetry boundary to be the one corresponding to fermionic Lagrangian algebra $\mathcal{A}_{b-,a}$, cf section III E 3. The anyons of the $S_3$ Dijkgraaf-Witten gauge theory, which is the SymTFT, act as the following generalized charges for $\mathsf{Rep}(S_3)^f$ symmetry.

- $(1, P)$ describes a bosonic $(-1)^F$-twisted sector (R sector) local operator which is uncharged under $E$.

- $(1, E)$ describes a multiplet of two operators.[19] One of them is an $E$-twisted sector bosonic local operator, and the other is a bosonic local operator converting $E$ line operator into $(-1)^F$ line operator. Both of these are uncharged under $E$.

---

[19] Naively one might think that there is a single operator in the multiplet as the bulk to boundary functor takes bulk line $(1, E)$ to boundary $E$ line without any multiplicity. If we view the symmetry boundary as coming from fermionization of $S_3$ boundary (up to stacking of Arf), then it is preferable to consider two operators in the $(1, E)$ multiplet, because the $S_3$ boundary carries two operators in this multiplet. From the point of view of $\mathsf{Rep}(S_3)^f$ boundary, the two operators are related by a trivalent junction comprising of two $E$ and one $P$ line in the $\mathsf{Rep}(S_3)^f$ symmetry category.

- $(a, 1)$ describes a multiplet of two operators. One of them is an untwisted sector (NS sector) bosonic local operator, and the other is a $(-1)^F$-twisted sector (R sector) bosonic local operator. The two operators are mixed by the action of $E$ in the same way as discussed in equation (5.16) of [3] (with $P$ replaced by $(-1)^F$).

- $(a, \omega)$ and $(a, \omega^2)$ both describe multiplets of two operators. Both of them are bosonic operators in the $E$-twisted sector. The two operators are mixed by the action of $E$ in a way quite similar to equation (5.16) of [3]. The coefficients involve in this mixing differentiate between the $(a, \omega)$ and $(a, \omega^2)$ multiplets. We leave the determination of these coefficients to an interested reader, following the methods described in [3].

- $(b, +)$ describes a multiplet of three operators. One of them is a fermionic $(-1)^F$-twisted sector (R sector) local operator. Another is an $E$-twisted sector fermionic local operator, and the last one is a fermionic local operator converting $E$ line operator into $(-1)^F$ line operator. The action of $E$ mixes the three operators.

- $(b, -)$ describes a multiplet of three operators. One of them is a fermionic untwisted sector (NS sector) local operator. Another is an $E$-twisted sector fermionic local operator, and the last one is a fermionic local operator converting $E$ line operator into $(-1)^F$ line operator. The action of $E$ mixes the three operators.

These can be derived by fermionizing the $\mathcal{A}_{a,b}$ boundary which carries bosonic $\mathsf{Rep}(S_3)$ symmetry. From the point of view of this symmetry, the various generalized charges, which are discussed in [3], are indeed bosonizations of the above ones. Let us discuss a few examples: $(1, P)$ describes an uncharged $P$-twisted sector operator for $\mathsf{Rep}(S_3)$ symmetry. $(1, E)$ describes an uncharged $E$-twisted sector operator and an uncharged operator converting $E$ line to $P$ line. $(a, 1)$ describes an untwisted sector and a $P$-twisted sector operator, both uncharged under $P$ and mixed by the action of $E$ as in equation (5.16) of [3].

### 4. $\mathbb{Z}_2 \times \mathbb{Z}_2^f$ with a Gu-Wen Anomaly

As described in Sec. III E, systems with a fermionic symmetry $\mathbb{Z}_2 \times \mathbb{Z}_2^f$ with a Gu-Wen anomaly $\nu = 2, 6$ mod 8 can be obtained via fermionization of bosonic systems with an anomalous $\mathbb{Z}_4$ symmetry $\mathsf{Vec}_{\mathbb{Z}_4}^\omega$, where $\omega = 2 \in H^3(\mathbb{Z}_4, U(1)) = \mathbb{Z}_4$. An anomaly $\omega = 2$ trivializes upon restricting to $\mathbb{Z}_2 \subset \mathbb{Z}_4$, implying that this $\mathbb{Z}_2$ subgroup is non-anomalous and hence can be fermionized. Consequently, the SymTFT for the $\mathbb{Z}_2 \times \mathbb{Z}_2^f$ Gu-Wen

anomalous symmetry is the $\mathbb{Z}_4$ Dijkgraaf-Witten TFT with the topological action $\omega$.

The anyon content of this SymTFT is given by the Drinfeld center

$$\mathcal{Z}(\mathsf{Vec}_{\mathbb{Z}_4}^\omega) = \{e^a m^b \mid a, b = 0, 1, 2, 3\}. \qquad \text{(IV.4)}$$

The topological spin and $S$-matrix are

$$
\begin{aligned}
\theta_{e^a m^b} &= \exp\left\{\frac{i\pi b(2a+b)}{4}\right\}, \\
S_{e^a m^b, e^{a'} m^{b'}} &= \exp\left\{\frac{2\pi i(ab' + a'b + bb')}{4}\right\}.
\end{aligned}
\qquad \text{(IV.5)}
$$

There are two bosonic gapped boundaries corresponding to the following Lagrangian algebras

$$
\begin{aligned}
\mathcal{A}_e &= 1 \oplus e \oplus e^2 \oplus e^3, \\
\mathcal{A}_{em^2} &= 1 \oplus em^2 \oplus e^2 \oplus e^3 m^2,
\end{aligned}
\qquad \text{(IV.6)}
$$

and a single fermionic topological boundary (up to Arf term) corresponding to the algebra

$$\mathcal{A}_{m^2, e^2} = 1 \oplus \pi m^2 \oplus e^2 \oplus \pi e^2 m^2. \qquad \text{(IV.7)}$$

The fermionic symmetry is generated by $\eta$ and $P$ which are obtained via the bulk-to-boundary functor for the fermionic Lagrangian algebra $\mathcal{A}_{m^2, e^2}$ as

$$F(e) = P, \qquad F(m) = \eta. \qquad \text{(IV.8)}$$

These lines have the fusion rules

$$P^2 = 1, \qquad \eta^2 = \pi. \qquad \text{(IV.9)}$$

We may choose $(-1)^F = P$ or $(-1)^F = \pi P$. We choose $(-1)^F = P$. The other choice can be worked out analogously.

First, we describe the generalized charges for the fermionic symmetry which are labeled by objects in $\mathcal{Z}(\mathsf{Vec}_{\mathbb{Z}_4}^\omega)$. Each line $a \in \mathcal{Z}(\mathsf{Vec}_{\mathbb{Z}_4}^\omega)$ in the SymTFT corresponds to a point operator denoted as $\mathcal{O}_a$ that can be characterized by (i) the twisted sector it belongs to and (ii) its charge under $P$ and $\eta$ which we denote as the tuple $(q_P, q_\eta)$.

- 1 is a local bosonic operator in the untwisted (NS) sector with a symmetry charge $(1, 1)$.

- $e^2$ is a local bosonic operator in the untwisted (NS) sector with symmetry charge $(1, -1)$.

- $m^2$ is a local fermionic operator in the untwisted (NS) sector with a symmetry charge $(-1, -1)$. Here $q_P = -1$ follows from the fact that $\pi m^2$ is condensed on the fermionic symmetry boundary and therefore $m^2$ has a bosonic interface to the $\pi$ line or equivalently a fermionic interface to the identity line. This is consistent with the fact that the braiding of $e$ with $m^2$ is $-1$. The charge $q_\eta = -1$ follows from the mutual-braiding of $m$-lines in the SymTFT.

- $e$ corresponds to a bosonic $P$-twisted (i.e., R sector) operator which carries a symmetry charge $(1, i)$.

- $m$ corresponds to an $\eta$-twisted operator which carries a charge $(i, i)$.

All the remaining charges can be obtained by taking the product of these charges. We note that an $\eta$ line acts on the $(-1)^F$-twisted sector operators as $\pm i$. Similarly, a fermion parity line $(-1)^F$ acts on the $\eta$-twisted sector operators as $\pm i$. These actions are related to each other via the exchange of space and time, i.e., a modular $S$-transformation.

It is illustrative to instead deduce these generalized charges by fermionizing the symmetry on the bosonic boundary $\mathcal{A}_{em^2}$. The category of lines on this boundary is $\mathsf{Vec}_{\mathbb{Z}_4}^\omega$ generated by $X$. The bulk to boundary map denoted as $F_{em^2}$ acts as

$$F_{em^2}(m) = X, \qquad F_{em^2}(m^2) = F_{em^2}(e) = X^2. \qquad \text{(IV.10)}$$

To obtain the fermionic symmetry with $(-1)^F = P$ described above, we fermionize the non-anomalous $\mathbb{Z}_2$ symmetry generated by $X^2$. Doing so, we naturally obtain a dual fermionic symmetry generated by $(-1)^F$ and the residual $\mathbb{Z}_4/\mathbb{Z}_2 \cong \mathbb{Z}_2$ generated by $X$ (which we call $\eta$ as a fermionic symmetry). Recall the map of sectors from the bosonic theory to the fermionic theory summarized in II.29 Using this fermionization map and (IV.10), we can now describe the bosonic generalized charges and what they map to under the fermionization.

- 1 is a local uncharged operator which maps to an untwisted (NS) sector boson uncharged under the fermionic symmetry group.

- $e^2$, being in $\mathcal{A}_{em^2}$, becomes a local operator. This carries a charge $-1$ under $X$ due to the bulk braiding of $-1$ between $e^2$ and $m$. This operator is uncharged under $X^2$ and therefore in the uncharged untwisted sector with respect to the symmetry being gauged. Under the fermionization we obtain an NS sector local boson with charge $(1, -1)$ as previously found.

- $m^2$ is an $X^2$ twisted sector operator. Its charge under $X$ is $-1$ due to the braiding between $m^2$ and $m$. Meanwhile its charge under $X^2$ is also -1. This follows from the braiding between the bulk lines $e$ and $m^2$. Note that in general there is a relation $q_{X^2} = -q_X^2$ on the $X^2$ twisted sector owing to the anomaly in the bosonic symmetry category.

where we use the explicit choice of cocycle

$$\omega(X^a, X^b, X^c) = e^{\frac{\pi i}{4} a(b+c-[b+c]_4)}, \qquad \text{(IV.11)}$$

with $[b+c]_4 = b + c \bmod 4$. Being in the twisted charged sector, this operator maps to the untwisted (NS) sector fermion with symmetry charges $(-1, -1)$ as we found earlier.

- $e$ corresponds to an $X^2$ twisted sector operator. Its charge $q_X = i$ which follows from the bulk braiding between $m$ and $e$. Meanwhile its $X^2$ charge is $-q_X^2 = +1$. This is in agreement with the trivial braiding of $e$ with itself. Being in the twisted uncharged sector, this operator maps to a twisted (R) sector boson with symmetry charge $(1, i)$, as expected.

- $m$ is an $X$-twisted sector operator with $q_X = q_{X^2} = i$. This follows from the self-braiding of $m$ and the mutual braiding of $e$ with $m$. Under the fermionization map, this maps to an $\eta$-twisted sector with symmetry charges $(i, i)$.

We thus recover the expected generalized charges from fermionization.

5. *$\mathbb{Z}_2 \times \mathbb{Z}_2^f$ Symmetry with a beyond Gu-Wen Anomaly*

As discussed in Section III E 5, the symmetry TFT for $\mathbb{Z}_2 \times \mathbb{Z}_2^f$ symmetry with a beyond Gu-Wen anomaly $\nu = 1, 7 \bmod 8$ is the doubled Ising TFT $\mathcal{Z}(\mathsf{Ising}_+)$. We choose the symmetry boundary of this TFT to be the one associated with the unique fermionic Lagrangian algebra $\mathcal{A}^f_{\psi, \overline{\psi}}$, see eq. (III.57). This fermionic boundary is invariant under stacking the Arf TFT, meaning that two choices $(-1)^F = P$ and $(-1)^F = \pi P$ of a fermion parity line give rise to the same fermionic boundary.

From the expression of $\mathcal{A}^f_{\psi, \overline{\psi}}$, we find that the anyons of the doubled Ising TFT $\mathcal{Z}(\mathsf{Ising}_+)$ act as the following generalized charges for $\mathbb{Z}_2 \times \mathbb{Z}_2^f$ symmetry with an anomaly $\nu = 1, 7 \bmod 8$:

- $1\overline{1}$ describes a bosonic operator in the untwisted sector (i.e., the NS sector) with charge $+\sqrt{2}$ under the action of $q$.

- $\psi\overline{\psi}$ describes a bosonic operator in the untwisted sector (i.e., the NS sector) with charge $-\sqrt{2}$ under the action of $q$.

- $1\overline{\psi}$ describes a fermionic operator in the untwisted sector (i.e., the NS sector) with the charge $+\sqrt{2}$ under the action of $q$.

- $\psi\overline{1}$ describes a fermionic operator in the untwisted sector (i.e., the NS sector) with charge $-\sqrt{2}$ under the action of $q$.

- $1\overline{\sigma}$ describes a multiplet of two operators in the $q(-1)^F$-twisted sector, one of which is bosonic and the other is fermionic. These operators are exchanged by the action of a $q$ line decorated by a fermionic point-like defect.

- $\sigma\overline{1}$ describes a multiplet of two operators in the $q$-twisted sector, one of which is bosonic and the other is fermionic. These operators are exchanged by the action of a $q$ line decorated by a fermionic point-like defect.

- $\psi\overline{\sigma}$ describes a multiplet of two operators in the $q(-1)^F$-twisted sector, one of which is bosonic and the other is fermionic. These operators are obtained by the fusion of operators with generalized charges $\psi\overline{1}$ and $1\overline{\sigma}$.

- $\sigma\overline{\psi}$ describes a multiplet of two operators in the $q$-twisted sector, one of which is bosonic and the other is fermionic. These operators are obtained by the fusion of operators with generalized charges $\sigma\overline{1}$ and $1\overline{\psi}$.

- $\sigma\overline{\sigma}$ describes a multiplet of two operators in the $(-1)^F$-twisted sector (i.e., the R sector), one of which is bosonic and the other is fermionic. These operators are exchanged by the action of a $q$ line decorated by a fermionic point-like defect.

## V. PHASES AND TRANSITIONS

### A. General Theory

The SymTFT $\mathfrak{Z}(\mathcal{S}_f)$ associated to a fermionic symmetry $\mathcal{S}_f$ can be used to understand the structure of possible gapped and gapless phases with symmetry $\mathcal{S}_f$, along with some of the transitions between the gapped phases. This is an extension of the bosonic case discussed recently in [3–6].

A gapped or gapless phase for $\mathcal{S}_f$, referred to simply as a *phase* for $\mathcal{S}_f$ in what follows, is by definition a set of possible IR physical phenomena compatible with the existence of $\mathcal{S}_f$ symmetry. More precisely, in (1+1)d a phase can be characterized by the **confinement** of a set $Q_C$ of charges, which means that a system lying in such a phase has no states in the IR and no local operators mapping between the IR states (referred to simply as IR local operators) that carry any of the charges lying in the set $Q_C$. Due to the lack of all possible charges in the IR, the symmetry $\mathcal{S}_f$ does not quite act faithfully on the IR theory, and phases may thus be equivalently characterized by possible choices of $\mathcal{S}_f$ symmetry with varying degrees of faithfulness.

**Condensed Charges.** Let us begin with the canonical gapless phase, which exists for every fermionic symmetry $\mathcal{S}_f$. This corresponds to having no confined charges, meaning that all the generalized charges for $\mathcal{S}_f$ appear

in the IR, and the $\mathcal{S}_f$ symmetry acts fully faithfully on the IR. Any other phase is obtained by condensing a set $Q$ of charges. This means that we give non-zero vacuum expectation values (vevs) to some local operators carrying charges lying in the set $Q$. This results in the IR carrying topological local operators that form multiplets transforming in charges lying in the set $Q$. The set $Q$ is quite constrained:

- First of all, any charge $a \in Q$ must only describe bosonic local operators $\mathcal{O}_a^Y$. This may be understood as imposing the requirement of having a Lorentz invariant vacuum in the IR.[20]

- Any charge $a$ that is mutually non-local with some other charge $b \in Q$ has to be confined. This is a generalization of Meissner effect. Consequently $a$ cannot be condensed, i.e. $a \notin Q$. Thus, all charges in $Q$ have to be mutually local.

Recalling that charges are identified with anyons of the SymTFT, we consider the above two conditions in the context of SymTFT. These conditions essentially describe what is known as a bosonic condensable algebra $\mathcal{A}$ in the MTC $\mathcal{Z}(\mathcal{S}_f)$ formed by SymTFT anyons. This is a (not necessarily simple) object of $\mathcal{Z}(\mathcal{S}_f)$ equipped with a commutative multiplication $\mathcal{A} \otimes \mathcal{A} \to \mathcal{A}$ that does not necessarily satisfy the condition (III.8). We can express the object underlying $\mathcal{A}$ as

$$\mathcal{A} = \bigoplus_{a \in \mathcal{Z}(\mathcal{S}_f)} N_a a, \qquad N_a \in \mathbb{Z}_{\geq 0}, \ N_1 = 1, \qquad \text{(V.1)}$$

Then the set $Q$ of condensed charges is the set of anyons having $N_a \neq 0$.

**Confined and Deconfined Charges.** Given the set $Q$, we can compute the set $Q_C$ of confined charges as the set of charges that are mutually non-local with at least one charge in $Q$. In terms of SymTFT, this is the set of anyons braiding non-trivially with at least one anyon in $\mathcal{A}$.

On the other hand, there may be a set $Q_D$ of non-condensed charges that remain deconfined in the phase. These are charges that are mutually local with all the charges in $Q$. In terms of SymTFT, this is the set of anyons braiding trivially (or local) with all anyons in $\mathcal{A}$ but are not themselves in $\mathcal{A}$. Such charges have to be carried by non-topological operators and gapless states in the IR theory. Thus, if $Q_D$ is non-empty then the phase under discussion is a gapless phase. However, if it is empty then the phase is a gapped phase in the sense that it can be realized by gapped systems with $\mathcal{S}_f$ symmetry.[21]

**Reduced Topological Order formed by Deconfined Charges.** As discussed above, the deconfined charges correspond to anyons local with the condensable algebra $\mathcal{A}$. Mathematically, we can characterize deconfined charges in terms of local modules for the algebra $\mathcal{A}$ in the MTC $\mathcal{Z}(\mathcal{S}_f)$. As discussed in [205], such modules form a smaller MTC $\mathcal{Z}'$ denoted as

$$\mathcal{Z}' = \mathcal{Z}(\mathcal{S}_f)_{\mathcal{A}}^{\text{loc}} \qquad \text{(V.2)}$$

and referred to as reduced topological order (TO).

Let us call the 3d TFT associated to $\mathcal{Z}'$ as $\mathfrak{Z}'$. Then the condensable algebra characterizes a topological interface $\mathcal{I}$ from $\mathfrak{Z}(\mathcal{S}_f)$ to $\mathfrak{Z}'$. The interface $\mathcal{I}$ is specified only up to stacking by invertible topological codimension-1 defects of $\mathfrak{Z}'$, or in other words up to the action of 0-form symmetries of $\mathfrak{Z}'$. The anyons appearing in $\mathcal{A}$ can topologically end on $\mathcal{I}$ without being attached to any other topological line, but no anyon of $\mathfrak{Z}'$ can end topologically on $\mathcal{I}$ in this fashion. See figure 2. The coefficient $N_a$ of an anyon $a$ in $\mathcal{A}$ is the dimension of the vector space formed by such topological ends of $a$.

By reflecting $\mathfrak{Z}'$ across the interface, $\mathcal{I}$ can be viewed as a bosonic topological boundary of the 3d TFT

$$\mathfrak{Z}(\mathcal{S}_f) \boxtimes \overline{\mathfrak{Z}'} \qquad \text{(V.3)}$$

where $\overline{\mathfrak{Z}'}$ is the orientation reversal of $\mathfrak{Z}'$. Thus $\mathcal{I}$ is associated to a bosonic Lagrangian algebra

$$\mathcal{L} \in \mathcal{Z}(\mathcal{S}_f) \boxtimes \overline{\mathcal{Z}'} \qquad \text{(V.4)}$$

where $\overline{\mathcal{Z}'}$ is the MTC formed by anyons of $\overline{\mathfrak{Z}'}$. The condensable algebra $\mathcal{A}$ can be recovered as the subalgebra of $\mathcal{L}$ involving only elements from the subfactor $\mathcal{Z}(\mathcal{S}_f)$ of $\mathcal{Z}(\mathcal{S}_f) \boxtimes \overline{\mathcal{Z}'}$. We call $\mathcal{L}$ as a Lagrangian algebra completion of the condensable algebra $\mathcal{A}$.

As described in [5], the information of $\mathcal{L}$ can be used to construct a functor

$$\mathcal{Z}_{\mathcal{L}}: \ \mathcal{Z}' \to \mathcal{Z}(\mathcal{S}_f) \qquad \text{(V.5)}$$

which describes the anyons of $\mathfrak{Z}(\mathcal{S}_f)$ that can be produced by passing the anyons of $\mathfrak{Z}'$ through the interface $\mathcal{I}$. See figure 2. The image of $\mathcal{Z}_{\mathcal{L}}$ captures precisely the set $Q_D$ of deconfined charges.

**Gapped vs Gapless Phases.** As discussed above, a gapped phase is characterized by an empty set $Q_D$ of deconfined anyons, meaning that the reduced topological order $\mathfrak{Z}'$ is the trivial 3d TFT. In other words, $\mathcal{I}$ is a topological boundary condition of the SymTFT $\mathfrak{Z}(\mathcal{S}_f)$. This means that the condensable algebra $\mathcal{A}$ is actually a

---

[20] A special case of this may be familiar from the study of relativistic QFTs, where only scalar fields are allowed to acquire non-zero vevs.

[21] It should be noted that a phase with empty $Q_D$ may also be realized by gapless systems, where all the gapless excitations and IR non-topological local operators do carry trivial charges under $\mathcal{S}_f$. Such a gapless phase can be deformed to a gapped phase while preserving the $\mathcal{S}_f$ symmetry.

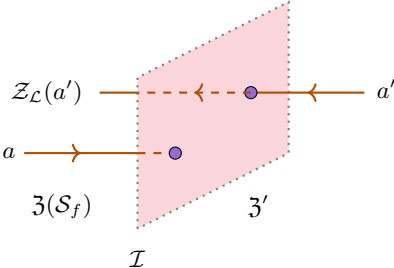

Figure 2. The interface $\mathcal{I}$ implements a map of lines from $\mathcal{Z}'$ to $\mathcal{Z}(\mathcal{S}_f)$. A line $a \in \mathcal{A}$ can end end on the interface from the left. Meanwhile a simple line $a' \in \mathcal{Z}'$ maps to a possibly non-simple line $\mathcal{Z}_{\mathcal{L}}(a') \in \mathcal{Z}(\mathcal{S}_f)$.

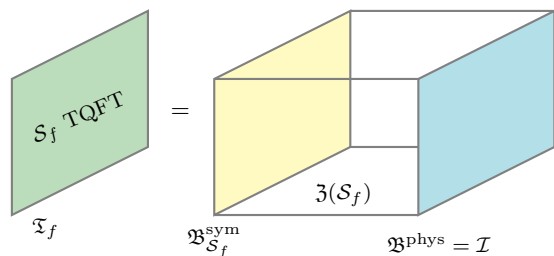

Figure 3. A 2d $\mathcal{S}_f$ symmetric fermionic TFT $\mathfrak{T}_f$ constructed as the interval compactification of the SymTFT $\mathfrak{Z}(\mathcal{S}_f)$ with symmetry boundary $\mathfrak{B}^{\mathrm{sym}}_{\mathcal{S}_f}$ and the bosonic topological boundary $\mathfrak{B}^{\mathrm{phys}} = \mathcal{I}$ on the other end.

bosonic Lagrangian algebra in $\mathcal{Z}(\mathcal{S}_f)$ satisfying the condition (III.8). A Lagrangian algebra is a maximal condensable algebra and describes a maximal set $Q$ of condensed charges.

Thus, gapped phases for fermionic symmetry $\mathcal{S}_f$ are characterized by bosonic Lagrangian algebras formed by anyons of the SymTFT $\mathfrak{Z}(\mathcal{S}_f)$, while the gapless phases for $\mathcal{S}_f$ are characterized by bosonic non-Lagrangian condensable algebras formed by anyons of the SymTFT $\mathfrak{Z}(\mathcal{S}_f)$.

**IR Theories for Gapped Phases.** The IR theory describing an $\mathcal{S}_f$ symmetric fermionic system in a gapped phase associated to a Lagrangian algebra $\mathcal{A}$ is a 2d $\mathcal{S}_f$ symmetric fermionic TFT $\mathfrak{T}_f$ which is constructed as the interval compactification of SymTFT $\mathfrak{Z}(\mathcal{S}_f)$ with symmetry boundary $\mathfrak{B}^{\mathrm{sym}}_{\mathcal{S}_f}$ on one end, and the bosonic topological boundary $\mathcal{I}$ on the other end, which is also referred to as a topological physical boundary and denoted as $\mathfrak{B}^{\mathrm{phys}}$ (see Fig. 3)

$$\mathfrak{B}^{\mathrm{phys}} = \mathcal{I}. \tag{V.6}$$

Let us describe the general structure of $\mathfrak{T}_f$ and realization of the symmetry $\mathcal{S}_f$ on it. Let $n$ be the number of vacua of $\mathfrak{T}_f$. This manifests as $\mathfrak{T}_f$ carrying an $n$-dimensional vector space of topological NS sector local operators or states on a circle. Such local operators of $\mathfrak{T}_f$ are constructed by taking simple anyons $a$ in Lagrangian algebras $\mathcal{A}$ and $\mathcal{A}_{\mathrm{sym}}$ (describing $\mathfrak{B}^{\mathrm{sym}}_{\mathcal{S}_f}$) and letting them completely end topologically (without being attached to other topological lines) on the topological boundaries $\mathfrak{B}^{\mathrm{sym}}_{\mathcal{S}_f}$ and $\mathfrak{B}^{\mathrm{phys}}$. The fusions of these operators can then be determined by using various consistency conditions with the action of topological lines living on the boundaries $\mathfrak{B}^{\mathrm{sym}}_{\mathcal{S}_f}$ and $\mathfrak{B}^{\mathrm{phys}}$. The vacua can be identified with idempotents of the algebra formed by these local operators under fusion. Let us label the vacua by an index $i$. The TFT $\mathfrak{T}_f$ in vacuum $i$ may reduce either to $\mathrm{Triv} \boxtimes \mathfrak{T}_{\lambda_i}$ or to $\mathrm{Arf} \boxtimes \mathfrak{T}_{\lambda_i}$, where $\mathfrak{T}_\lambda$ is an Euler term. The relative Euler terms $\lambda_{ij} = \lambda_j - \lambda_i$ are protected by the action of symmetry $\mathcal{S}_f$ if $\mathfrak{T}_f$ is an irreducible $\mathcal{S}_f$-symmetric 2d TFT, but it is possible to shift

all the Euler terms by the same constant $\lambda_i \to \lambda_i + r$ without breaking the $\mathcal{S}_f$ symmetry. Thus, the information of the overall Euler term does not enter an $\mathcal{S}_f$-symmetric $(1+1)$d gapped phase, but the information of relative Euler terms does.

The full set of topological line operators of the 2d TFT $\mathfrak{T}_f$ forms a fermionic multi-fusion $\pi$-supercategory $\mathcal{S}_f(\mathfrak{T}_f)$. First of all, we have indecomposable lines living in each vacuum $i$, which are simply $1_{ii}$ and $\pi_{ii}$. If there is no relative Arf term between two vacua $i$ and $j$, then the indecomposable lines from vacuum $i$ to vacuum $j$ are $1_{ij}, \pi_{ij}$, which are exchanged by the action of $\pi_{ii}, \pi_{jj}$

$$\begin{aligned} \pi_{ii} \otimes 1_{ij} = 1_{ij} \otimes \pi_{jj} = \pi_{ij} \\ \pi_{ii} \otimes \pi_{ij} = \pi_{ij} \otimes \pi_{jj} = 1_{ij} \end{aligned} \tag{V.7}$$

The linking action of $1_{ij}, \pi_{ij}$ on vacuum $v_i$ results in a multiple of vacuum $v_j$ determined by the relative Euler term between $i$ and $j$

$$1_{ij}, \pi_{ij} : \ v_i \to e^{-(\lambda_j - \lambda_i)} v_j \tag{V.8}$$

See section 2.3 of [3] for an explanation. This linking action may also be referred to as the quantum dimension of the lines $1_{ij}, \pi_{ij}$. On the other hand, if there is a relative Arf term between vacua $i$ and $j$, then there is only one indecomposable $q$-type line $S_{ij}$ from vacuum $i$ to vacuum $j$ satisfying

$$\begin{aligned} 1_{ii} \otimes S_{ij} = S_{ij} \otimes 1_{jj} = S_{ij} \\ \pi_{ii} \otimes S_{ij} = S_{ij} \otimes \pi_{jj} = S_{ij} \end{aligned} \tag{V.9}$$

The line $S_{ij}$ describes the end of Kitaev chain, and its quantum dimension is

$$S_{ij} : \ v_i \to e^{-(\lambda_j - \lambda_i)} \sqrt{2} v_j \tag{V.10}$$

Other non-zero fusions of these lines are

$$\begin{aligned} 1_{ij} \otimes 1_{jk} = \pi_{ij} \otimes \pi_{jk} = 1_{ik} \\ \pi_{ij} \otimes 1_{jk} = 1_{ij} \otimes \pi_{jk} = \pi_{ik} \\ 1_{ij} \otimes S_{jk} = \pi_{ij} \otimes S_{jk} = S_{ik} \\ S_{ij} \otimes 1_{jk} = S_{ij} \otimes \pi_{jk} = S_{ik} \\ S_{ij} \otimes S_{jk} = 1_{ik} \oplus \pi_{ik} \end{aligned} \tag{V.11}$$

The fermionic parity symmetry of $\mathcal{S}_f(\mathfrak{T}_f)$ is

$$(-1)_{\mathfrak{T}_f}^F = \bigoplus_i \Pi_{ii} \tag{V.12}$$

where $\Pi_{ii} = 1_{ii}$ if the vacuum $i$ comprises of $\text{Triv} \boxtimes \mathfrak{T}_{\lambda_i}$, and $\Pi_{ii} = \pi_{ii}$ if the vacuum $i$ comprises of $\text{Arf} \boxtimes \mathfrak{T}_{\lambda_i}$.

The fermionic symmetry $\mathcal{S}_f$ is described by a subset of the topological lines of $\mathfrak{T}_f$, which is described by a supertensor functor [173, 174]

$$\sigma: \ \mathcal{S}_f \to \mathcal{S}_f(\mathfrak{T}_f) \tag{V.13}$$

constrained to satisfy

$$\sigma\big((-1)^F\big) = (-1)_{\mathfrak{T}_f}^F \tag{V.14}$$

The information of $\sigma$ is determined as follows. We consider the NS sector local operators arising from interval compactifications of anyons. These are acted upon by $X \in \mathcal{S}_f$ line living on $\mathfrak{B}_{\mathcal{S}_f}^{\text{sym}}$. This provides us with the action of $X$ on the vacua. Then $\sigma(X) \in \mathcal{S}_f(\mathfrak{T}_f)$ is the line that acts on vacua in the same way.

A symmetry $X \in \mathcal{S}_f$ is said to be spontaneously broken in a vacuum $i$ if $\sigma(X)$ contains topological line operators taking the vacuum $i$ to some other vacuum $j$.

**IR Theories for Gapless Phases.** As mentioned at the starting of this section, a gapless phase by definition only describes a set of symmetry related IR properties of gapless $\mathcal{S}_f$ symmetric systems. The dynamical properties are constrained by these symmetry properties, but are not completely fixed by them. Consequently, for a particular gapless phase, there can be various different $\mathcal{S}_f$-symmetric fermionic 2d CFTs describing the IR behaviors of various (1+1)d fermionic systems said to be lying in this gapless phase. This should be contrasted with the situation for gapped phases where the IR theory is uniquely fixed to be an $\mathcal{S}_f$-symmetric fermionic 2d TFT.

Let us now describe the SymTFT construction for any $\mathcal{S}_f$-symmetric fermionic CFT $\mathfrak{T}_f$ arising in the IR of a gapless phase associated to a condensable algebra $\mathcal{A}$. This is obtained by inputting a conformal boundary condition $\mathfrak{B}^{\text{phys}}$ of $\mathfrak{Z}'$ into the SymTFT setup and compactifying the whole interval

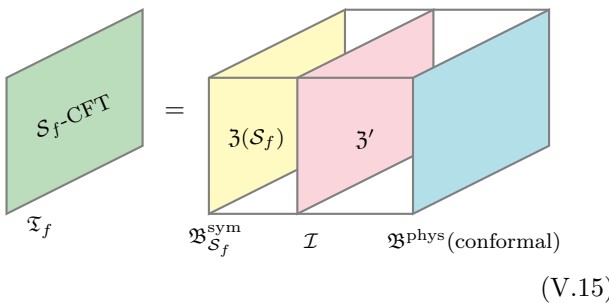

$$\tag{V.15}$$

Note that $\mathfrak{B}^{\text{phys}}$ is required to satisfy the following conditions:

- None of the anyons of $\mathfrak{Z}'$ can topologically end on $\mathfrak{B}^{\text{phys}}$ without being attached to any other topological line operator. This ensures that the introduction of $\mathfrak{B}^{\text{phys}}$ does not induce the condensation of charges outside the set $Q$.

- There is at least one (non-topological) end of every anyon of $\mathfrak{Z}'$ along $\mathfrak{B}^{\text{phys}}$ which is unattached to any other topological line operator. This ensures that the whole set $Q_D$ of deconfined charges is realized in the IR theory.

- We further assume that there are no topological local operators (other than multiples of identity operator) living on $\mathfrak{B}^{\text{phys}}$ which are unattached to any bulk or boundary topological line operator. This ensures that $\mathfrak{T}_f$ is an irreducible $\mathcal{S}_f$-symmetric CFT.

Let us describe the general structure of $\mathfrak{T}_f$ and realization of symmetry $\mathcal{S}_f$ on it. For this purpose, it is useful to first perform a club-quiche compactification where we only compactify the interval occupied by $\mathfrak{Z}(\mathcal{S}_f)$ but not by $\mathfrak{Z}'$. This constructs a fermionic topological boundary condition $\mathfrak{B}'_f$ of $\mathfrak{Z}'$ which is $\mathcal{S}_f$-symmetric. The boundary $\mathfrak{B}'_f$ is irreducible as an $\mathcal{S}_f$-symmetric boundary, but may be reducible when the $\mathcal{S}_f$ symmetry is forgotten.

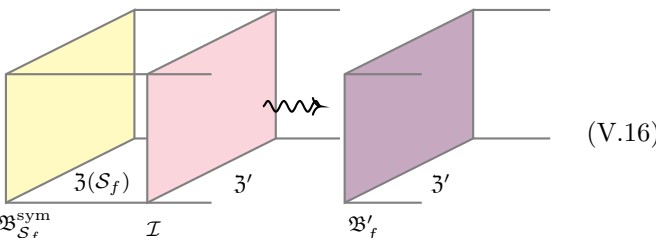

$$\tag{V.16}$$

Let $n$ be the number of irreducible boundary conditions involved in $\mathfrak{B}'_f$. This manifests as $\mathfrak{B}'_f$ carrying an $n$-dimensional vector space of topological NS sector local operators. Such local operators of $\mathfrak{B}'_f$ are constructed by taking simple anyons $a$ in condensable algebra $\mathcal{A}$ and Lagrangian algebra $\mathcal{A}_{\text{sym}}$ (describing $\mathfrak{B}_{\mathcal{S}_f}^{\text{sym}}$) and letting them completely end topologically (without being attached to other topological lines) on the topological boundary $\mathfrak{B}_{\mathcal{S}_f}^{\text{sym}}$ and topological interface $\mathcal{I}$. The irreducible boundaries in $\mathfrak{B}'_f$ can be identified with idempotents of the algebra formed by these local operators under fusion. Let us label the irreducible boundaries in $\mathfrak{B}'_f$ as $\mathfrak{B}'_{f,i}$

$$\mathfrak{B}'_f = \bigoplus_i \mathfrak{B}'_{f,i} \tag{V.17}$$

The full set of topological line operators of $\mathfrak{B}'_f$ forms a fermionic multi-fusion $\pi$-supercategory $\mathcal{S}_f(\mathfrak{B}'_f)$ with fermion parity

$$(-1)_{\mathfrak{B}'_f}^F = \bigoplus_i (-1)_{\mathfrak{B}'_{f,i}}^F \tag{V.18}$$

The category $\mathcal{S}_f(\mathfrak{B}'_f)$ includes not only topological line operators living on $\mathfrak{B}'_{f,i}$, but also topological line operators living between two boundaries $\mathfrak{B}'_{f,i}$ and $\mathfrak{B}'_{f,j}$. The quantum dimensions of lines going between two different boundaries $\mathfrak{B}'_{f,i}$ and $\mathfrak{B}'_{f,j}$ depends on the relative Euler terms between these boundaries. The fermionic symmetry $\mathcal{S}_f$ is realized by a supertensor functor

$$\sigma: \ \mathcal{S}_f \to \mathcal{S}_f(\mathfrak{B}'_f) \qquad (\text{V.19})$$

constrained to satisfy

$$\sigma\big((-1)^F\big) = (-1)^F_{\mathfrak{B}'_f} \qquad (\text{V.20})$$

The information of $\sigma$ is determined as follows. We consider all the possible topological ends of anyons of $\mathfrak{Z}'$ along $\mathfrak{B}'$ that are not attached to any boundary topological lines. Under the club quiche compactification, these can all be constructed by passing anyons of $\mathfrak{Z}'$ transversely through $\mathcal{I}$ which converts them into anyons of $\mathfrak{Z}(\mathcal{S}_f)$ on the other side in terms of the functor $\mathcal{Z}_\mathcal{L}$ and ending the $\mathfrak{Z}(\mathcal{S}_f)$ anyons on $\mathfrak{B}^{\text{sym}}_{\mathcal{S}_f}$. Thus these ends are acted upon by $X \in \mathcal{S}_f$ line living on $\mathfrak{B}^{\text{sym}}_{\mathcal{S}_f}$. This provides us with the action of $X$ on the ends of anyons of $\mathfrak{Z}'$ along $\mathfrak{B}'_f$. Then $\sigma(X) \in \mathcal{S}_f(\mathfrak{B}'_f)$ is the line operator of $\mathfrak{B}'_f$ that acts on these ends in the same way.

The CFT $\mathfrak{T}_f$ is constructed by performing the club sandwich compactification in which we subsequently compactify the interval occupied by $\mathfrak{Z}'$. This reduces into a direct sum of compactifications $(\mathfrak{B}'_{f,i}, \mathfrak{B}^{\text{phys}})$, each of them giving rise to a CFT $\mathfrak{T}_{f,i}$ with a single topological NS sector local operator. The full CFT $\mathfrak{T}_f$ thus comprises of $n$ universes and can be expressed as

$$\mathfrak{T}_f = \bigoplus_i \mathfrak{T}_{f,i} \qquad (\text{V.21})$$

From this compactification we can read the action of fermionic multi-fusion category $\mathcal{S}_f(\mathfrak{B}'_f)$ on $\mathfrak{T}_f$. Using the functor (V.19), we obtain an action $\mathcal{S}_f$ on $\mathfrak{T}_f$ converting it into an $\mathcal{S}_f$-symmetric CFT.

**Generalized Fermionic KT Transformations.** We have constructed above an $\mathcal{S}_f$-symmetric CFT $\mathfrak{T}_f$ describing the IR of a system lying in a gapless phase associated to a condensable algebra $\mathcal{A}$, provided that we are given a conformal boundary condition $\mathfrak{B}^{\text{phys}}$ of $\mathfrak{Z}'$ satisfying certain properties.

The question now arises how one can construct such boundary conditions. For this purpose, we use the sandwich construction in reverse. Let $\mathcal{S}'$ be a bosonic or fermionic symmetry whose SymTFT is $\mathfrak{Z}'$

$$\mathfrak{Z}(\mathcal{S}') = \mathfrak{Z}' \qquad (\text{V.22})$$

Assume that we know a 2d CFT $\mathfrak{T}'$ that describes the IR of the canonical gapless phase of $\mathcal{S}'$. That is, all the generalized charges of $\mathcal{S}'$ arise in $\mathfrak{T}'$ and are carried by non-topological local operators in $\mathfrak{T}'$.

Then, $\mathfrak{B}^{\text{phys}}$ can be taken to the physical boundary arising in the SymTFT construction of $\mathfrak{T}'$ with some choice of symmetry boundary $\mathfrak{B}^{\text{sym}}_{\mathcal{S}'}$.

The map

$$\mathfrak{T}' \longrightarrow \mathfrak{T}_f \qquad (\text{V.23})$$

may be referred to as a fermionic generalized Kennedy-Tasaki (KT) transformation, extending the generalized KT transformations discussed in the bosonic case by [5].

In fact, each $\mathfrak{T}_{f,i}$ is obtained by some fermionic gauging of the $\mathcal{S}'$ symmetry of $\mathfrak{T}'$, since $\mathfrak{T}_{f,i}$ is obtained by modifying the symmetry boundary from $\mathfrak{B}^{\text{sym}}_{\mathcal{S}'}$ to $\mathfrak{B}'_{f,i}$.

**Deformations of Phases.** Let us consider that we are deep inside a phase specified by a set $Q$ of condensed charges, i.e. all the condensed local operators have non-small vevs of order one. Now, a small deformation of such a system cannot uncondense any of the charges, but can condense some other charges by providing small non-zero vevs to some local operators carrying deconfined charges not in the set $Q$.

After such a deformation, we move to a phase characterized by a set $Q'$ of condensed charges, which is bigger than $Q$

$$Q \subseteq Q' \qquad (\text{V.24})$$

If $\mathcal{A}$ and $\mathcal{A}'$ are the corresponding condensable algebras, then we have the condition that $\mathcal{A}$ be a subalgebra of $\mathcal{A}'$. The set of confined charges also increases under such a deformation

$$Q_C \subseteq Q'_C \qquad (\text{V.25})$$

as condensing new charges confines some of the previously deconfined charges. On the other hand, the set of non-condensed deconfined charges shrinks

$$Q'_D \subseteq Q_D \qquad (\text{V.26})$$

which is consistent with the physical expectation that such deformation should increases the amount of gapped excitations while decreasing the amount of gapless ones.

Thus we have a partial order on the set of $\mathcal{S}_f$-symmetric phases given by inclusions of condensable algebras, which captures possible deformations between the phases. This allows us to arrange the phases into a Hasse diagram in which the canonical gapless phase sits at the top and the gapped phases sit at the bottom, with various other gapless phases sitting in the middle [6].

**Classification of Phases.** The $\mathcal{S}_f$-symmetric fermionic phases discussed here can be classified into various classes. A gapped phase is called a spontaneous symmetry breaking (SSB) phase if it carries more than one vacuum, and a symmetry preserving topological (SPT) phase if it carries a single vacuum. On the other hand, a gapless phase is called a gapless SSB (gSSB) phase if it carries more than one universe, and a gapless SPT (gSPT) phase if it carries a single universe.

Furthermore a gSSB or gSPT phase $\mathcal{P}$ is called an intrisic gSSB or gSPT phase (igSPT or igSSB) if any gapped phase obtained after an allowed deformation of $\mathcal{P}$ has strictly more number of vacua than the number of universes in $\mathcal{P}$.

An igSSB or igSPT phase exhibits **symmetry protected criticality**: any deformation of the phase preserves gapless criticality unless we are willing to increase the amount of order in the system and (spontaneously) break some of the symmetry in the process.

**Transitions between Gapped Phases.** We can obtain transitions between $\mathcal{S}_f$-symmetric gapped phases by applying generalized fermionic KT transformations on known transitions between gapped phases of smaller symmetries.

Consider the KT transformation associated to a condensable algebra $\mathcal{A}$ and let $\mathcal{S}'$ be a (bosonic or fermionic) symmetry such that $\mathfrak{Z}(\mathcal{S}') = \mathfrak{Z}'$ as above. The KT transformation then maps $\mathcal{S}'$ symmetric systems to $\mathcal{S}_f$ symmetric systems.

Now assume that we know an $\mathcal{S}'$-symmetric CFT $\mathfrak{T}'$ that admits a relevant deformation $\epsilon$, which is uncharged under $\mathcal{S}'$, such that deforming $\mathfrak{T}'$ by two signs of $\epsilon$ gives rise to two $\mathcal{S}'$ symmetric gapped phases $\mathfrak{T}'_+$ and $\mathfrak{T}'_-$ in the IR

$$
\begin{aligned}
\mathfrak{T}' + \epsilon &\to \mathfrak{T}'_+ \\
\mathfrak{T}' - \epsilon &\to \mathfrak{T}'_-
\end{aligned}
\tag{V.27}
$$

We can now apply the KT transformation to obtain an $\mathcal{S}_f$-symmetric CFT $KT(\mathfrak{T}')$ which admits a relevant deformation $KT(\epsilon)$ such that deforming $KT(\mathfrak{T}')$ by two signs of $KT(\epsilon)$ gives rise to two $\mathcal{S}_f$ symmetric gapped phases $KT(\mathfrak{T}'_+)$ and $KT(\mathfrak{T}'_-)$ in the IR

$$
\begin{aligned}
KT(\mathfrak{T}') + KT(\epsilon) &\to KT(\mathfrak{T}'_+) \\
KT(\mathfrak{T}') - KT(\epsilon) &\to KT(\mathfrak{T}'_-)
\end{aligned}
\tag{V.28}
$$

The gapped phases $KT(\mathfrak{T}'_\pm)$ can be easily determined. Let $\mathcal{A}_\pm$ be the Lagrangian algebras in $\mathcal{Z}'$ associated to the physical topological boundaries associated to $\mathcal{S}'$-symmetric gapped phases $\mathfrak{T}'_\pm$. Then, the Lagrangian algebras $KT(\mathcal{A}_\pm) \in \mathcal{Z}(\mathcal{S}_f)$ associated to $\mathcal{S}_f$-symmetric gapped phases $KT(\mathfrak{T}'_\pm)$ can be computed by applying the functor $\mathcal{Z}_\mathcal{L}$

$$
KT(\mathcal{A}_\pm) = \mathcal{Z}_\mathcal{L}(\mathcal{A}_\pm)
\tag{V.29}
$$

**Order Parameters.** The operators condensed in a phase are often referred to as order parameters for that phase. This terminology is typically used for gapped phases, but we may extend it to gapless phases.

The charges of order parameters associated to a phase are of particular physical relevance. These are obtained simply as the set $Q$ of condensed charges associated to the phase, which is specified by the anyons of the SymTFT $\mathfrak{Z}(\mathcal{S}_f)$ appearing in the associated condensable algebra $\mathcal{A}$.

The charges of order parameters for transitions between two gapped phases with condensed charges $Q_\pm$ are given by the set

$$
(Q_+ \cup Q_-) - (Q_+ \cap Q_-)
\tag{V.30}
$$

which are the charges that distinguish the two gapped phases.

## B. $\mathbb{Z}_2^f$ Symmetry

The SymTFT $\mathfrak{Z}(\mathbb{Z}_2^f)$ for the simplest fermionic symmetry group $\mathbb{Z}_2^f$ is the Toric code or $\mathbb{Z}_2$ Dijkgraaf-Witten gauge theory, which was described in III E 1. The anyon content is

$$
\mathcal{Z}(\mathsf{Vec}_{\mathbb{Z}_2}) = \{1, e, m, f = em\},
\tag{V.31}
$$

where $e$ and $m$ are bosons and $f$ is a fermion. We take symmetry boundary to be $\mathfrak{B}_f$ which is the unique fermionic boundary. The choice of $(-1)^F$ on $\mathfrak{B}_f$ is again fixed as in eq. (III.32) which implies the generalized charges charges associated to the objects in $\mathcal{Z}(\mathbb{Z}_2^f)$ as discussed in IV B 1. The symmetry fermionic Lagrangian algebra is $\mathcal{A}_{\text{sym}} = \mathcal{A}_f = 1 \oplus \pi f$.

The gapped phases correspond to Lagrangian algebras in $\mathcal{Z}(\mathsf{Vec}_{\mathbb{Z}_2})$, for which there are two possibilities.

$$
\mathcal{A}_e = 1 \oplus e, \qquad \mathcal{A}_m = 1 \oplus m,
\tag{V.32}
$$

With the following rather simple Hasse diagram of inclusion of condensable algebras

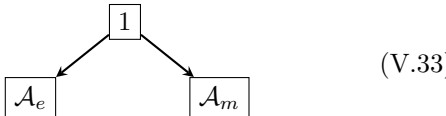

$$\tag{V.33}$$

Choosing the physical Lagrangian algebra (i.e. Lagrangian algebra for physical boundary) to be

$$
\mathcal{A}_{\text{phys}} = \mathcal{A}_m,
\tag{V.34}
$$

we find an SPT phase, since we have

$$
\mathcal{A}_{\text{sym}} \cap \mathcal{A}_{\text{phys}} = \mathcal{A}_f \cap \mathcal{A}_m = 1,
\tag{V.35}
$$

implying a single (untwisted sector) local operator and therefore a unique ground state. From the form of $\mathcal{A}_m$, we see that this gapped phase contains a topological point-like operator carrying generalized charge $m$, or in other words a topological point-like operator which is a boson and lives at the end of $(-1)^F$. This implies that $(-1)^F$ is bosonically isomorphic to the identity line in this gapped phase:

$$
(-1)^F = 1.
\tag{V.36}
$$

This phase is the trivial SPT phase for $\mathbb{Z}_2^f$ symmetry and is denoted as Triv in this paper. The confined charges

for this phase are $e$ and $f$. The order parameter for the phase carries charge $m$, i.e., it is an R sector boson.

Choosing the physical Lagrangian algebra to be

$$\mathcal{A}_{\text{phys}} = \mathcal{A}_e, \qquad (\text{V.37})$$

we again find an SPT phase since

$$\mathcal{A}_{\text{sym}} \cap \mathcal{A}_{\text{phys}} = \mathcal{A}_f \cap \mathcal{A}_e = 1. \qquad (\text{V.38})$$

From the form of $\mathcal{A}_e$, we see that this gapped phase contains a topological point-like operator carrying generalized charge $e$, or in other words a topological point-like operator which is a fermion and lives at the end of $(-1)^F$. Combining it with the fermion living at the end of $\pi$ line, we obtain a boson living at the end of $\pi(-1)^F$, implying that this line is bosonically isomorphic to the identity line $\pi(-1)^F = 1$, or in other words

$$(-1)^F = \pi. \qquad (\text{V.39})$$

This phase is the non-trivial SPT phase for $\mathbb{Z}_2^f$ symmetry and is denoted as Arf in this paper. The confined charges for this phase are $m$ and $f$. The order parameter for the phase carries charge $e$, i.e. it is an R sector fermion.

We can also obtain these phases via the fermionization of $\mathbb{Z}_2$ symmetric Bosonic phases. The trivial or symmetry preserving $\mathbb{Z}_2$ phase has a single untwisted sector and a single twisted sector ground state, both of which are uncharged under $\mathbb{Z}_2$. Under fermionization, these map to the NS sector and R sector ground states respectively that are both fermion parity even. Hence we recover the Triv phase. Instead, starting from the $\mathbb{Z}_2$ SSB bosonic phase, the IR theory has two untwisted sector ground states which are swapped under the $\mathbb{Z}_2$ action. There are no twisted sector ground states. The even combination of ground states is uncharged and untwisted under $\mathbb{Z}_2$, and under fermionization maps to a fermion parity even NS sector ground state. The odd combination, being charged under $\mathbb{Z}_2$ maps to the R sector state which is fermion parity odd. Hence we find that the fermionization of the $\mathbb{Z}_2$ SSB gives the Arf phase.

Naturally, the fermionization of the transition between the $\mathbb{Z}_2$ trivial and $\mathbb{Z}_2$ SSB phases delivers a transition between the two fermionic SPTs Triv and Arf. On the bosonic side, the minimal such transition lies in the Ising universality class, whose fermionization is the Majorana CFT labeled by Maj. Therefore to summarize, the Hasse diagram of phases in (V.33) is realized for $\mathbb{Z}_2^f$ symmetric phases as

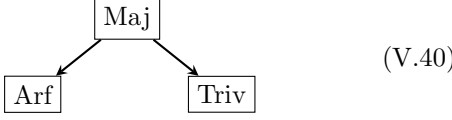

$$(\text{V.40})$$

which is the fermionization of $\mathbb{Z}_2$ symmetric bosonic phases which realize the Hasse diagram

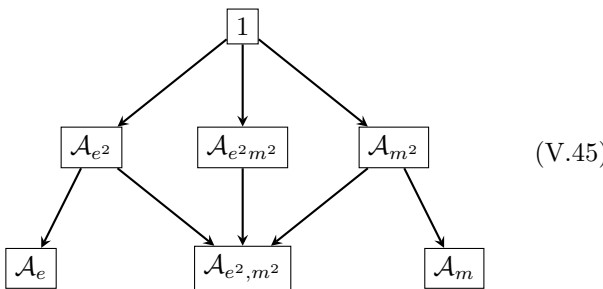

$$(\text{V.41})$$

## C.   $\mathbb{Z}_4^f$ and $\mathbb{Z}_4^{\pi f}$ Symmetries

The SymTFT for the $\mathbb{Z}_4^f$ and $\mathbb{Z}_4^{\pi f}$ symmetries is the $\mathbb{Z}_4$ Dijkgraaf-Witten (DW) gauge theory as the SymTFT which was discussed in Sec. III E 2. Recall that to define the symmetry boundary we chose the following fermionic Lagrangian algebra in $\mathcal{Z}(\mathbb{Z}_4)$

$$\mathcal{A}_{em^2} = 1 \oplus \pi e m^2 \oplus e^2 \oplus \pi e^3 m^2. \qquad (\text{V.42})$$

This algebra can give rise to two different boundary conditions (related by stacking of Arf theory) which carry two different symmetries. These are

$$\begin{aligned} \mathbb{Z}_4^f &= \{1, P, P^2 = (-1)^F, P^3\}, \\ \mathbb{Z}_4^{\pi f} &= \{1, P, P^2 = \pi(-1)^F, P^3\}, \end{aligned} \qquad (\text{V.43})$$

differentiated by whether $P^2$ is identified with $(-1)^F$ or with $\pi(-1)^F$. The relation between the bulk anyons and boundary topological lines was discussed in III E 2, while the relation between the SymTFT anyons and generalized charges of $\mathbb{Z}_4^f / \mathbb{Z}_4^{\pi f}$ are described in IV B 2.

The possible gapped and gapless phases as well as transitions can be studied in the SymTFT via the nontrivial bosonic condensable algebras in $\mathbb{Z}_4$ Dijkgraaf-Witten gauge theory. These are

$$\begin{aligned} \mathcal{A}_{e^2} &= 1 \oplus e^2 \\ \mathcal{A}_{m^2} &= 1 \oplus m^2 \\ \mathcal{A}_{e^2 m^2} &= 1 \oplus e^2 m^2 \\ \mathcal{A}_e &= 1 \oplus e \oplus e^2 \oplus e^3 \\ \mathcal{A}_m &= 1 \oplus m \oplus m^2 \oplus m^3 \\ \mathcal{A}_{e^2, m^2} &= 1 \oplus e^2 \oplus m^2 \oplus e^2 m^2 \end{aligned} \qquad (\text{V.44})$$

Out of these $\mathcal{A}_e$, $\mathcal{A}_m$ and $\mathcal{A}_{e^2,m^2}$ are Lagrangian, which give rise to fermionic gapped phases while the remaining correspond to transitions between these phases. We have the following Hasse diagram of inclusion of condensable algebras

$$(\text{V.45})$$

We now describe the gapped and gapless phases obtained for different choices of condensable algebras following the general construction described in Sec. V A.

Below is a Hasse diagram of gapped and gapless phases for the $\mathbb{Z}_4^f$ symmetry that we find in what follows. The

Hasse diagram for $\mathbb{Z}_4^{\pi f}$ symmetry can be simply obtained by stacking each node with the Arf theory

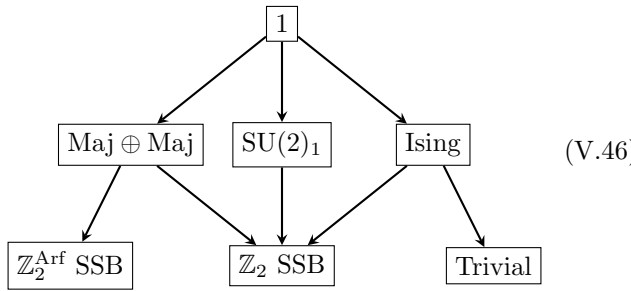

This Hasse diagram can be obtained as a fermionization of the following Bosonic Hasse diagram with the choice $\mathfrak{B}^{\text{sym}} = \mathcal{A}_e$

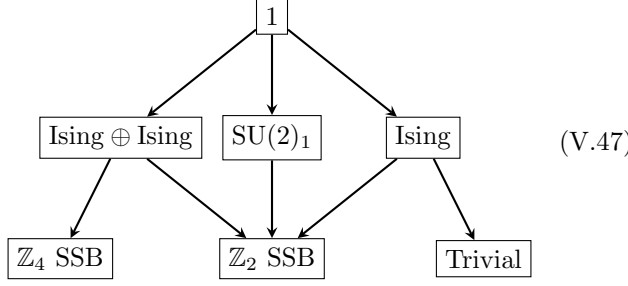

### 1. $\mathbb{Z}_2^{Arf}$ SSB phase

Consider first the gapped phase with physical Lagrangian algebra

$$\mathcal{A}_{\text{phys}} = \mathcal{A}_e. \tag{V.48}$$

This phase has two vacua because not only 1 but also $e^2$ can end on both boundaries. Additionally from $\mathcal{A}_e$ we learn that the phase contains topological point-like operators $\mathcal{O}_{e^i}$ carrying charges $e^i$, which are the IR images of the order parameters associated to the gapped phase. These operators have product

$$\mathcal{O}_{e^i}\mathcal{O}_{e^j} = \mathcal{O}_{e^{i+j}}. \tag{V.49}$$

The untwisted sector local operators are generated by $\mathcal{O}_{e^0} = 1$ and $\mathcal{O}_{e^2}$, from which we identify the two vacua to be

$$v_0 = \frac{1 + \mathcal{O}_{e^2}}{2},$$
$$v_1 = \frac{1 - \mathcal{O}_{e^2}}{2}, \tag{V.50}$$

which satisfy the condition

$$v_i v_j = \delta_{ij} v_i. \tag{V.51}$$

The operators $\mathcal{O}_e$ and $\mathcal{O}_{e^3}$ are both fermions in $P^2$-twisted sector, so we can construct linear combinations

$$\mathcal{O}_{e,0} = \frac{1}{2}\left(\mathcal{O}_e + \mathcal{O}_{e^3}\right),$$
$$\mathcal{O}_{e,1} = \frac{i}{2}(\mathcal{O}_e - \mathcal{O}_{e^3}), \tag{V.52}$$

which satisfy

$$\mathcal{O}_{e,i}v_j = \delta_{ij}\mathcal{O}_{e,i}, \qquad \mathcal{O}_{e,i}\mathcal{O}_{e,j} = \delta_{ij}v_i \tag{V.53}$$

meaning that $\mathcal{O}_{e,i}$ is a fermionic $P^2$-twisted sector local operator in vacuum $v_i$. This identifies the operator implementing $P^2$ symmetry as

$$P^2 = \pi_{00} \oplus \pi_{11}, \tag{V.54}$$

as the end of $\pi$ line is fermionic. The above equation can also be derived from the fact that the $e$ line in the bulk becomes $\pi P^2$ on the fermionic symmetry boundary, while it becomes the trivial line on the bosonic physical boundary, and hence we have $\pi P^2 = 1$ in this fermionic gapped phase. For $\mathbb{Z}_4^f$ symmetry, eq. (V.54) implies that the fermionic parity operator is

$$(-1)^F = \pi_{00} \oplus \pi_{11}, \tag{V.55}$$

or in other words both vacua are described by the Arf theory, and the underlying IR TQFT describing the gapped phase can be represented as

$$\mathfrak{T}^{\text{IR}} = \text{Arf}_0 \oplus \text{Arf}_1 \tag{V.56}$$

On the other hand, for $\mathbb{Z}_4^{\pi f}$ symmetry, eq. (V.54) implies that the fermionic parity operator is

$$(-1)^F = 1_{00} \oplus 1_{11}, \tag{V.57}$$

or in other words both vacua are described by the Triv theory, and the underlying IR TQFT describing the gapped phase can be represented as

$$\mathfrak{T}^{\text{IR}} = \text{Triv}_0 \oplus \text{Triv}_1. \tag{V.58}$$

The underlying IR TQFTs for $\mathbb{Z}_4^f$ and $\mathbb{Z}_4^{\pi f}$ symmetries differ by an overall Arf factor, a fact that is true for all systems with this symmetry, which stems from the fact that the corresponding symmetry boundaries in the SymTFT differ by stacking of Arf theory. Consequently, we will only discuss $\mathbb{Z}_4^f$ symmetric phases from this point on; the $\mathbb{Z}_4^{\pi f}$ symmetric phases are simply obtained by stacking an overall Arf factor.

The action of the line operator $P$ is

$$P: \ \mathcal{O}_{e^k} \to i^k \mathcal{O}_{e^k}, \tag{V.59}$$

which implies the actions

$$\begin{aligned} &v_0 \leftrightarrow v_1, \\ P: \ &\mathcal{O}_{e,0} \to \mathcal{O}_{e,1}, \\ &\mathcal{O}_{e,1} \to -\mathcal{O}_{e,0}. \end{aligned} \tag{V.60}$$

This equation allows us to determine the $P$ line operator to be

$$P = 1_{01} \oplus \pi_{10}, \tag{V.61}$$

where

$$\pi_{10} = \pi_{11} \otimes 1_{10} = (-1)^F_{11} \otimes 1_{10} \qquad (V.62)$$

is responsible for the sign in the action $\mathcal{O}_{e,1} \to -\mathcal{O}_{e,0}$ utilizing the fact that $\mathcal{O}_{e,1}$ is a fermion. Here, the operators $1_{ij}$ for $i, j = 0, 1$ are defined by the following actions on the untwisted and twisted sector operators:

$$1_{ij} : v_i \to v_j, \quad \mathcal{O}_{e,i} \to \mathcal{O}_{e,j}. \qquad (V.63)$$

We note that eq. (V.61) is a unique solution for eq. (V.54) up to the exchange of labels 0 and 1. We refer to this phase as the $\mathbb{Z}_2^{\mathrm{Arf}}$ SSB phase of the $\mathbb{Z}_4^f$ symmetry. This phase is the fermionization of a bosonic gapped phase that spontaneously breaks a non-anomalous $\mathbb{Z}_4$ symmetry down to the trivial group.

We note that there is an ambiguity in the definition of the twisted sector operators $\mathcal{O}_{e,i}$ for $i = 0, 1$. Specifically, we can redefine them as follows:

$$\mathcal{O}'_{e,0} := \mathcal{O}_{e,0}, \quad \mathcal{O}'_{e,1} := -\mathcal{O}_{e,1}, \qquad (V.64)$$

This redefinition preserves the operator algebra (V.53). On the other hand, there is no ambiguity in the definition of the untwisted sector operators $v_i$ due to the operator algebra (V.51). Accordingly, we can also redefine the line operators $1_{ij}$ and $\pi_{ij}$ by

$$\begin{aligned} 1'_{ij} &: v_i \to v_j, \quad \mathcal{O}'_{e,i} \to \mathcal{O}'_{e,j}, \\ \pi'_{ij} &: v_i \to v_j, \quad \mathcal{O}'_{e,i} = -\mathcal{O}'_{e,j}. \end{aligned} \qquad (V.65)$$

In particular, we have $1'_{ij} = \pi_{ij}$ and $\pi'_{ij} = 1_{ij}$ for $i \neq j$. In terms of these redefined operators, the $P$ line in eq. (V.61) is expressed as

$$P = 1_{01} \oplus \pi_{10} = \pi'_{01} \oplus 1'_{10}, \qquad (V.66)$$

which shows that the expression for $P$ depends on a convention. Nevertheless, we emphasize that the operator $P$ itself is defined unambiguously regardless of a convention. Similar comments also apply to the other examples that we will discuss in this and later sections.

This phase is a fermionization of the $\mathbb{Z}_4$ SSB phase carrying 4 vacua. From the point of view of $\mathbb{Z}_2$ subgroup of $\mathbb{Z}_4$, the $\mathbb{Z}_4$ SSB phase splits into a direct sum of two $\mathbb{Z}_2$ SSB phases. The fermionization of each $\mathbb{Z}_2$ SSB phase is Arf, and hence the fermionization indeed results in the $\mathbb{Z}_2^{\mathrm{Arf}}$ SSB phase whose underlying TFT is Arf $\oplus$ Arf.

### 2. $\mathbb{Z}_2$ SSB phase

Take the physical Lagrangian algebra to now be

$$\mathcal{A}_{\mathrm{phys}} = \mathcal{A}_{e^2, m^2}. \qquad (V.67)$$

Again there are two vacua because of the same reason as above. The IR images of the order parameters are topological point-like operators $\mathcal{O}_{e^2}$, $\mathcal{O}_{m^2}$ and $\mathcal{O}_{e^2 m^2}$ carrying

the generalized charges described by the subscript. The untwisted sector operators are 1 and $\mathcal{O}_{e^2}$, in terms of which the vacua can again be expressed as in (V.50). The other two operators $\mathcal{O}_{m^2}$ and $\mathcal{O}_{e^2 m^2}$ are in $P^2 = (-1)^F$-twisted sector and both of them are bosons. The operators

$$\begin{aligned} \mathcal{O}_{m^2,0} &= \mathcal{O}_{m^2} + \mathcal{O}_{e^2 m^2}, \\ \mathcal{O}_{m^2,1} &= \mathcal{O}_{m^2} - \mathcal{O}_{e^2 m^2} \end{aligned} \qquad (V.68)$$

describe bosonic ends of $(-1)^F$ in the two vacua, implying that $(-1)^F$ is realized trivially in both vacua, and the underlying IR TQFT describing the gapped phase can be represented as

$$\mathfrak{T}^{\mathrm{IR}} = \mathrm{Triv}_0 \oplus \mathrm{Triv}_1. \qquad (V.69)$$

The symmetry operator $P$ acts by a non-trivial sign on $\mathcal{O}_{e^2}$ and $\mathcal{O}_{e^2 m^2}$, which means that it exchanges the two vacua. Thus, we can express it as

$$P = 1_{01} \oplus 1_{10}. \qquad (V.70)$$

We refer to this phase as the $\mathbb{Z}_2$ SSB phase of the $\mathbb{Z}_4^f$ symmetry.

This phase is the fermionization of a bosonic gapped phase that spontaneously breaks a non-anomalous $\mathbb{Z}_4$ symmetry down to $\mathbb{Z}_2$. From the point of view of the $\mathbb{Z}_2$ subgroup of $\mathbb{Z}_4$, this phase is bosonic Triv $\oplus$ Triv. The fermionization of each bosonic Triv factor is a fermionic Triv factor, thus reproducing the above result.

### 3. Trivial phase

Finally, take the physical Lagrangian algebra to be

$$\mathcal{A}_{\mathrm{phys}} = \mathcal{A}_m. \qquad (V.71)$$

The corresponding fermionic gapped phase has a single vacuum as no non-trivial anyon can end on both $\mathfrak{B}^{\mathrm{sym}}$ and $\mathfrak{B}^{\mathrm{phys}}$. The IR images of order parameters are topological point-like operators $\mathcal{O}_{m^i}$ carrying charges described by their subscript. In particular, $\mathcal{O}_{m^2}$ provides a bosonic topological end of $(-1)^F$ and hence the IR TQFT is a single copy of the trivial theory

$$\mathfrak{T}^{\mathrm{IR}} = \mathrm{Triv}. \qquad (V.72)$$

Similarly, $\mathcal{O}_m$ provides a bosonic topological end of $P$, implying that $P$ acts completely trivially and can be identified with the identity line operator

$$P = 1. \qquad (V.73)$$

We refer to this phase as the trivial phase of the $\mathbb{Z}_4^f$ symmetry. This phase is the fermionization of the trivial bosonic gapped phase with a non-anomalous $\mathbb{Z}_4$ symmetry.

#### 4. $e^2$-condensed phase

Now let us discuss gapless phases with $\mathbb{Z}_4^f$ symmetry. First, consider the physical condensable algebra to be

$$\mathcal{A}_{\mathrm{phys}} = \mathcal{A}_{e^2}. \qquad (\mathrm{V.74})$$

This defines a bosonic topological interface $\mathcal{I}_{e^2}$ from the $\mathbb{Z}_4$ DW theory to the toric code. Thus, the club quiche compactification produces a topological boundary condition $\mathfrak{B}'$ of the toric code. As $e^2$ also appears in $\mathcal{A}_{\mathrm{sym}}$, we learn that the boundary $\mathfrak{B}'$ has two topological local operators on it, which implies that $\mathfrak{B}'$ is comprised of two irreducible topological boundary conditions

$$\mathfrak{B}' = \mathfrak{B}'_0 \oplus \mathfrak{B}'_1. \qquad (\mathrm{V.75})$$

As we pass the interface $\mathcal{I}_{e^2}$, the anyons of toric code are converted into the anyons of $\mathbb{Z}_4$ DW theory, according to the map

$$\begin{aligned}
1 &\mapsto 1 \oplus e^2, \\
e' &\mapsto e \oplus e^3, \\
m' &\mapsto m^2 \oplus e^2 m^2, \\
f' &\mapsto em^2 \oplus e^3 m^2,
\end{aligned} \qquad (\mathrm{V.76})$$

where we have denoted the anyons of the toric code by an additional prime to avoid confusing them with the anyons of the $\mathbb{Z}_4$ DW theory. From this and the Lagrangian algebra $\mathcal{A}_{\mathrm{sym}} = \mathcal{A}_{em^2}$, we learn that there are two fermionic topological ends of $f'$ along $\mathfrak{B}'$, one coming from the end of $em^2$ along $\mathfrak{B}^{\mathrm{sym}}_{\mathbb{Z}_4^f}$ and the other coming from the end of $e^3 m^2$ along $\mathfrak{B}^{\mathrm{sym}}_{\mathbb{Z}_4^f}$. We label these ends respectively as $\mathcal{E}_{em^2}$ and $\mathcal{E}_{e^3 m^2}$. Furthermore there is a topological local operator $\mathcal{O}_{e^2}$ along $\mathfrak{B}'$ coming from compactifying the $e^2$ in between $\mathfrak{B}^{\mathrm{sym}}_{\mathbb{Z}_4^f}$ and $\mathcal{I}_{e^2}$. The products of these operators obey the fusion of their subscripts:

$$\begin{aligned}
\mathcal{E}_{e^i m^2} \mathcal{E}_{e^j m^2} &= \mathcal{O}_{e^{i+j}}, \\
\mathcal{E}_{e^i m^2} \mathcal{O}_{e^j} &= \mathcal{E}_{e^{i+j} m^2}, \\
\mathcal{O}_{e^i} \mathcal{O}_{e^j} &= \mathcal{O}_{e^{i+j}},
\end{aligned} \qquad (\mathrm{V.77})$$

where $\mathcal{O}_{e^0} = 1$. The identity operators along the two boundaries $\mathfrak{B}'_0$ and $\mathfrak{B}'_1$ are

$$\begin{aligned}
v_0 &= \frac{1 + \mathcal{O}_{e^2}}{2}, \\
v_1 &= \frac{1 - \mathcal{O}_{e^2}}{2},
\end{aligned} \qquad (\mathrm{V.78})$$

satisfying

$$v_i v_j = \delta_{ij} v_i. \qquad (\mathrm{V.79})$$

We have fermionic topological ends

$$\begin{aligned}
\mathcal{E}_0 &= \frac{1}{2}(\mathcal{E}_{em^2} + \mathcal{E}_{e^3 m^2}), \\
\mathcal{E}_1 &= \frac{i}{2}(\mathcal{E}_{em^2} - \mathcal{E}_{e^3 m^2})
\end{aligned} \qquad (\mathrm{V.80})$$

of $f'$ along $\mathfrak{B}'_0$ and $\mathfrak{B}'_1$ respectively, which satisfy

$$\mathcal{E}_i v_j = \delta_{ij} \mathcal{E}_i, \qquad \mathcal{E}_i \mathcal{E}_j = \delta_{ij} v_i. \qquad (\mathrm{V.81})$$

This means that both $\mathfrak{B}'_0$ and $\mathfrak{B}'_1$ are irreducible fermionic topological boundaries of toric code associated to the fermionic Lagrangian algebra $1 \oplus \pi f'$.

Note that this does not determine yet whether there is a relative Euler term or Arf term between $\mathfrak{B}'_0$ and $\mathfrak{B}'_1$. This can be determined by studying the $\mathbb{Z}_4^f$ symmetry action. We know that the symmetry generator $P$ has linking action

$$P : \begin{array}{l} \mathcal{E}_{e^j m^2} \to i^j \, \mathcal{E}_{e^j m^2}, \\ \mathcal{O}_{e^j} \to i^j \, \mathcal{O}_{e^j}, \end{array} \qquad (\mathrm{V.82})$$

which implies the linking action

$$P : \begin{array}{l} \mathcal{E}_0 \to \mathcal{E}_1, \\ \mathcal{E}_1 \to -\mathcal{E}_0, \\ v_0 \leftrightarrow v_1. \end{array} \qquad (\mathrm{V.83})$$

This means that $P$ is implemented by line operators lying between $\mathfrak{B}'_0$ and $\mathfrak{B}'_1$. The fact that these line operators have to be invertible implies that there is no relative Arf term between $\mathfrak{B}'_0$ and $\mathfrak{B}'_1$. This is because the interface between $\mathfrak{B}'_0$ and $\mathfrak{B}'_1$ would become non-invertible if there were a relative Arf term, cf. the discussion around (V.11). Moreover, the linking action of $P$ on $v_i$ implies that the quantum dimension of these boundary changing line operators must be 1, and hence there is no relative Euler term between $\mathfrak{B}'_0$ and $\mathfrak{B}'_1$.

Due to eq. (V.83), we can express the $\mathbb{Z}_4^f$ symmetry generator $P$ as

$$P = 1_{01} \oplus (-1)^F_{10}, \qquad (\mathrm{V.84})$$

where $(-1)^F_{10} := 1_{10} \otimes (-1)^F_{00} = (-1)^F_{11} \otimes 1_{10}$ and we have denoted the fermionic symmetry generating line operator along the fermionic boundary $\mathfrak{B}'_i$ as $(-1)^F_{ii}$. The full fermionic symmetry generator indeed involves the fermionic symmetry generator along both $\mathfrak{B}'_0$ and $\mathfrak{B}'_1$

$$(-1)^F = P^2 = (-1)^F_{00} \oplus (-1)^F_{11}. \qquad (\mathrm{V.85})$$

We note that eq. (V.84) is the unique solution for the condition $P^2 = (-1)^F = (-1)^F_{01} \oplus (-1)^F_{10}$ up to the exchange of labels 0 and 1.

Providing a gapless physical boundary $\mathfrak{B}^{\mathrm{phys}}_{\mathfrak{T}^f}$, which is a gapless boundary of the toric code, completes the club sandwich and constructs the IR theory $\mathfrak{T}^{\mathrm{IR}}$ of a system in the corresponding gapless phase, which can be expressed as

$$\mathfrak{T}^{\mathrm{IR}} = \mathfrak{T}^f_0 \oplus \mathfrak{T}^f_1, \qquad (\mathrm{V.86})$$

where $\mathfrak{T}^f_i$ are two copies of a gapless theory $\mathfrak{T}^f$ with $\mathbb{Z}_2^f$ fermionic symmetry. The gapless phase thus comprises of two universes and hence a gapless SSB (gSSB) phase for

$\mathbb{Z}_4^f$ symmetry. The $\mathbb{Z}_4^f$ symmetry acts as in (V.84) and (V.85), where now $(-1)_{ii}^F$ is the generator of $\mathbb{Z}_2^f$ symmetry of $\mathfrak{T}_i^f$.

$\mathfrak{T}^f$ can be any $\mathbb{Z}_2^f$ symmetric fermionic CFT carrying all generalized charges for $\mathbb{Z}_2^f$. The simplest example is provided by the Majorana CFT, and hence one of the IR theories realizing this gapless phase is

$$\text{Maj}_0 \oplus \text{Maj}_1 \,, \qquad (V.87)$$

with $\mathbb{Z}_4^f$ being realized by (V.84).

Let $\epsilon$ be the relevant operator, uncharged under $\mathbb{Z}_2^f$, responsible for deforming the Maj CFT to Triv and Arf TFTs. Then it is clear from (V.84) that the operator $\epsilon_0 + \epsilon_1$ of $\text{Maj}_0 \oplus \text{Maj}_1$ is uncharged under $\mathbb{Z}_4^f$, as the generator $P$ simply exchanges $\epsilon_0$ and $\epsilon_1$. Using $\epsilon_0 + \epsilon_1$ as the deformation, we obtain a transition between the $\mathbb{Z}_2$ SSB and $\mathbb{Z}_2^{\text{Arf}}$ SSB phases of $\mathbb{Z}_4^f$ symmetry.

Note that the condensable algebras corresponding to these two gapped phases are $\mathcal{A}_e$ and $\mathcal{A}_{e^2,m^2}$, both of which contain the condensable algebra $\mathcal{A}_{e^2}$ for the gapless phase under consideration.

Let us now discuss the gapless phase from the point of view of fermionization. In terms of the bosonic $\mathbb{Z}_4$ symmetry, the condensable algebra $\mathcal{A}_{e^2}$ corresponds to a phase for which the IR CFT is

$$\mathfrak{T}_0 \oplus \mathfrak{T}_1 \qquad (V.88)$$

where $\mathfrak{T}_i$ is a copy of a $\mathbb{Z}_2$ symmetric CFT $\mathfrak{T}$. This $\mathbb{Z}_2$ symmetry is identified with the $\mathbb{Z}_2$ subgroup of $\mathbb{Z}_4$. Thus, the fermionization simply fermionizes each $\mathfrak{T}_i$ factor leading to (V.86), where $\mathfrak{T}^f$ is fermionization of $\mathfrak{T}$. Taking $\mathfrak{T}$ to be Ising CFT, we obtain the above choice $\mathfrak{T}^f = \text{Maj}$.

### 5. $m^2$-condensed phase

Now consider the physical condensable algebra to be

$$\mathcal{A}_{\text{phys}} = \mathcal{A}_{m^2}. \qquad (V.89)$$

This also defines a bosonic topological interface $\mathcal{I}_{m^2}$ from $\mathbb{Z}_4$ DW theory to the toric code. As no non-trivial anyons appear in both $\mathcal{A}_{\text{sym}}$ and $\mathcal{A}_{\text{phys}}$, we learn that the club quiche compactification of $\mathfrak{B}_{\mathbb{Z}_4^f}^{\text{sym}}$ and $\mathcal{I}_{m^2}$ produces an irreducible topological boundary condition $\mathfrak{B}'$ of $\mathbb{Z}_2$ DW theory. As we pass the interface $\mathcal{I}_{m^2}$, the anyons of $\mathbb{Z}_2$ DW theory are converted into the anyons of $\mathbb{Z}_4$ DW theory according to the map

$$\begin{aligned}
1 &\mapsto 1 \oplus m^2, \\
e' &\mapsto e^2 \oplus e^2 m^2, \\
m' &\mapsto m \oplus m^3, \\
f' &\mapsto e^2 m \oplus e^2 m^3.
\end{aligned} \qquad (V.90)$$

Thus $e'$ is the only non-trivial anyon that can end along $\mathfrak{B}'$ via the end of $e^2$ along $\mathfrak{B}_{\mathbb{Z}_4^f}^{\text{sym}}$, and we recognize $\mathfrak{B}'$

to be the irreducible boundary of $\mathbb{Z}_2$ DW theory associated to the Lagrangian algebra $1 \oplus e'$. The $\mathbb{Z}_4^f$ symmetry generator $P$ acts on the end of $e^2$ along $\mathfrak{B}_{\mathbb{Z}_4^f}^{\text{sym}}$ by a non-trivial sign, which means that the realization of $P$ along $\mathfrak{B}'$ acts by a non-trivial sign on the end of $e'$ along $\mathfrak{B}'$. That is, $P$ is realized by the $\mathbb{Z}_2$ symmetry generator $P'$ along $\mathfrak{B}'$

$$P' = P, \qquad (V.91)$$

which identifies

$$(-1)^F = P^2 = 1. \qquad (V.92)$$

Due to this reason, we can identify $\mathfrak{B}'$ to be the bosonic topological boundary condition $\mathfrak{B}_{e'}$ associated to $1 \oplus e'$ without the Arf term. If instead we were considering $\mathbb{Z}_4^{\pi f}$ symmetry, the fermion parity would be realized on $\mathfrak{B}'$ as $(-1)^F = \pi$, and we would have $\mathfrak{B}' = \mathfrak{B}_{e'} \boxtimes \text{Arf}$, namely the fermionic boundary associated to $1 \oplus e'$ that is obtained from the bosonic boundary $\mathfrak{B}_{e'}$ by stacking an additional Arf term.

Completing the club sandwich by a bosonic gapless physical boundary, we obtain a gSPT phase for $\mathbb{Z}_4^f$ symmetry, in which the IR theory $\mathfrak{T}^{\text{IR}}$ is a bosonic gapless theory (stacked with a trivial fermionic TFT) with a symmetry $\mathbb{Z}_2$ acting faithfully on it, and the $\mathbb{Z}_4^f$ symmetry is realized on $\mathfrak{T}^{\text{IR}}$ via (V.91) and (V.92).

We can choose the bosonic theory to be Ising CFT, which transition between the $\mathbb{Z}_2$ trivial and $\mathbb{Z}_2$ SSB phases for $\mathbb{Z}_4^f$ symmetry. These gapped phases correspond to $\mathcal{A}_{e^2,m^2}$ and $\mathcal{A}_m$, both of which carry $\mathcal{A}_{m^2}$ as a subalgebra.

From the point of view of non-anomalous bosonic $\mathbb{Z}_4$ symmetry, $\mathcal{A}_{m^2}$ corresponds to a gapless phase comprising of a $\mathbb{Z}_2$ symmetric CFT, with the $\mathbb{Z}_4$ being realized by the surjective homomorphism $\mathbb{Z}_4 \to \mathbb{Z}_2$. The $\mathbb{Z}_2$ subgroup of $\mathbb{Z}_4$ acts trivially, and hence fermionization does not change the theory.

### 6. $e^2 m^2$-condensed phase

Finally, consider the condensable algebra

$$\mathcal{A}_{\text{phys}} = \mathcal{A}_{e^2 m^2}. \qquad (V.93)$$

This defines a bosonic topological interface $\mathcal{I}_{e^2 m^2}$ from $\mathbb{Z}_4$ DW theory to the double semion model, or twisted $\mathbb{Z}_2$ DW theory. As no non-trivial anyons appear in common in both $\mathcal{A}_{\text{sym}}$ and $\mathcal{A}_{\text{phys}}$, we learn that the club quiche compactification of $\mathfrak{B}_{\mathbb{Z}_4^f}^{\text{sym}}$ and $\mathcal{I}_{e^2 m^2}$ produces an irreducible topological boundary condition $\mathfrak{B}'$ of twisted $\mathbb{Z}_2$ DW theory. As we pass the interface $\mathcal{I}_{e^2 m^2}$, the anyons of twisted $\mathbb{Z}_2$ DW theory are converted into the anyons

of $\mathbb{Z}_4$ DW theory, according to the map

$$
\begin{aligned}
1 &\mapsto 1 \oplus e^2 m^2, \\
s &\mapsto em \oplus e^3 m^3, \\
\bar{s} &\mapsto em^3 \oplus e^3 m, \\
s\bar{s} &\mapsto e^2 \oplus m^2,
\end{aligned}
\tag{V.94}
$$

where $s$ and $\bar{s}$ denote the semion and anti-semion respectively. We thus learn that there is a topological end of $s\bar{s}$ along $\mathfrak{B}'$ coming from an end of $e^2$ along $\mathfrak{B}_{\mathbb{Z}_4^f}^{\text{sym}}$. This deduces $\mathfrak{B}'$ to be a boundary associated to the Lagrangian algebra $1 \oplus s\bar{s}$ of twisted $\mathbb{Z}_2$ DW theory, which is a topological boundary condition carrying bosonic $\mathbb{Z}_2$ symmetry with non-trivial 't Hooft anomaly, or in short $\mathbb{Z}_2^\omega$ symmetry with $\omega \in H^3(\mathbb{Z}_2, U(1))$ being non-trivial. We let $P'$ be the topological line operator along $\mathfrak{B}'$ generating the $\mathbb{Z}_2^\omega$ symmetry.

The $\mathbb{Z}_4^f$ generator $P$ acts on the end of $e^2$ by a non-trivial sign, implying that $P$ is realized on $\mathfrak{B}'$ by a line operator under which the end of $s\bar{s}$ is charged, i.e.

$$
P = P'. \tag{V.95}
$$

This means that $(-1)^F$ is realized as

$$
(-1)^F = P^2 = 1. \tag{V.96}
$$

Due to this reason, we can identify $\mathfrak{B}'$ to be the bosonic topological boundary condition $\mathfrak{B}_{s\bar{s}}$ associated to $1 \oplus s\bar{s}$ without the Arf term. If instead we were considering $\mathbb{Z}_4^{\pi f}$ symmetry, the fermion parity would be realized on $\mathfrak{B}'$ as $(-1)^F = \pi$, and we would have $\mathfrak{B}' = \mathfrak{B}_{s\bar{s}} \boxtimes \text{Arf}$, namely the fermionic boundary associated to $1 \oplus s\bar{s}$ that is obtained from the bosonic boundary $\mathfrak{B}_{s\bar{s}}$ by stacking an additional Arf term.

In addition to these identification of line operators, there is a non-trivial identification of topological junction local operators between $\mathbb{Z}_4^f$ lines as topological junction local operators between $\mathbb{Z}_2^\omega$ lines which ensures consistency with the 't Hooft anomaly $\omega$. See [5] for this map of junction operators.

Completing the club sandwich by a bosonic gapless physical boundary, we obtain a gSPT phase for $\mathbb{Z}_4^f$ symmetry, in which the IR theory $\mathfrak{T}^{\text{IR}}$ is a bosonic gapless theory (stacked with a trivial fermionic TFT) with a symmetry $\mathbb{Z}_2^\omega$ acting faithfully on it, and the $\mathbb{Z}_4^f$ symmetry is realized on $\mathfrak{T}^{\text{IR}}$ via (V.95) and (V.96). In fact, this is an igSPT phase as any gapped deformation of a $\mathbb{Z}_2^\omega$ symmetric system produces more than one vacua. A candidate for such a critical theory is the $SU(2)_1$ WZW model, which also realizes the deformation from $\mathcal{A}_{e^2 m^2}$ gapless phase to $\mathcal{A}_{e^2, m^2}$ gapped phase where the $\mathbb{Z}_2^\omega$ symmetry is spontaneously broken.

For bosonic $\mathbb{Z}_4$ symmetry, $\mathcal{A}_{e^2 m^2}$ also describes a $\mathbb{Z}_2^\omega$ symmetric CFT on which $\mathbb{Z}_4$ acts according to surjective homomorphism $\mathbb{Z}_4 \to \mathbb{Z}_2$, and hence $\mathbb{Z}_2 \subset \mathbb{Z}_4$ acts trivially. Thus, the fermionization does not change the theory.

## D. $\text{Rep}(S_3)^f$ Symmetry

Now we consider the DW gauge theory for the smallest non-abelian group

$$
S_3 = \{1, a, a^2, b, ab, a^2 b\}, \quad a^3 = b^2 = 1, \quad ba = a^2 b \tag{V.97}
$$

as the SymTFT. Its anyon content is labeled as

$$
\begin{array}{lll}
\dim = 1: & (1,1), & (1,P) \\
\dim = 2: & (1,E), & (a, \omega^i) \\
\dim = 3: & (b,+), & (b,-)
\end{array}
\qquad
\begin{array}{l}
\omega = e^{2\pi i/3} \\[4pt]
i \in \{0,1,2\}
\end{array}
\tag{V.98}
$$

where we have also displayed the quantum dimensions of the anyons. The bosons are $(1,1)$, $(1,P)$, $(1,E)$, $(a,1)$ and $(b,+)$, and the only fermion is $(b,-)$. On the other hand, the spin of $(a, \omega^i)$ is $\omega^i$. There are two possible fermionic Lagrangian algebras [206, 213], that could be chosen to be the symmetry Lagrangian algebra

$$
\begin{aligned}
\mathcal{A}_{b-,E} &= (1,1) \oplus \pi(b,-) \oplus (1,E), \\
\mathcal{A}_{b-,a} &= (1,1) \oplus \pi(b,-) \oplus (a,1),
\end{aligned}
\tag{V.99}
$$

which are related by a 0-form symmetry of the $S_3$ DW theory that exchanges $(1,E)$ and $(a,1)$. As a consequence, the symmetry boundaries corresponding to the two Lagrangian algebras give equivalent results. Without loss of generality, we choose

$$
\mathcal{A}_{\text{sym}} = \mathcal{A}_{b-,a} \tag{V.100}
$$

to be the symmetry Lagrangian algebra. This algebra can give rise to two different boundary conditions (related by stacking of Arf theory) which carry two different symmetries. These are the $\text{Rep}(S_3)^f$ and $\text{Rep}(S_3)^{\pi f}$ symmetries discussed earlier. Moving forward, we fix the symmetry to be $\text{Rep}(S_3)^f$, as the results for $\text{Rep}(S_3)^{\pi f}$ are obtained simply by stacking an overall Arf term.

The non-trivial bosonic condensable algebras in $S_3$ Dijkgraaf-Witten gauge theory are

$$
\begin{aligned}
\mathcal{A}_P &= (1,1) \oplus (1,P) \\
\mathcal{A}_E &= (1,1) \oplus (1,E) \\
\mathcal{A}_a &= (1,1) \oplus (a,1) \\
\mathcal{A}_{P,E} &= (1,1) \oplus (1,P) \oplus 2(1,E) \\
\mathcal{A}_{P,a} &= (1,1) \oplus (1,P) \oplus 2(a,1) \\
\mathcal{A}_{E,b} &= (1,1) \oplus (1,E) \oplus (b,+) \\
\mathcal{A}_{a,b} &= (1,1) \oplus (a,1) \oplus (b,+)
\end{aligned}
\tag{V.101}
$$

Out of these $\mathcal{A}_{P,E}$, $\mathcal{A}_{P,a}$, $\mathcal{A}_{E,b}$ and $\mathcal{A}_{a,b}$ are Lagrangian. These condensable algebras form the following Hasse di-

agram

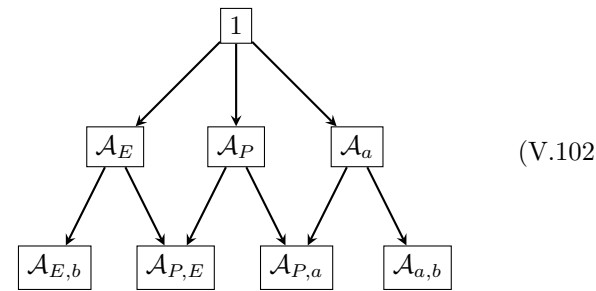

$$(V.102)$$

The bosonic Hasse diagram is

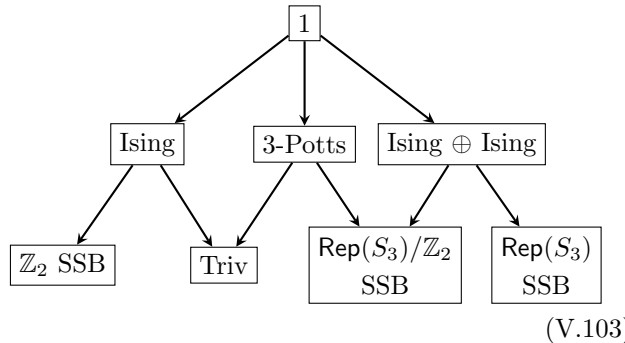

$$(V.103)$$

The fermionic Hasse diagram is

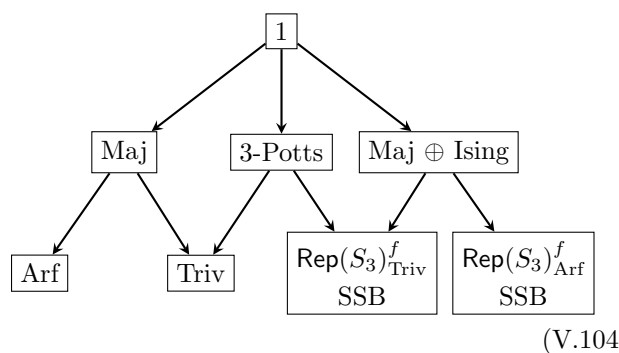

$$(V.104)$$

*1.  (1, P)-condensed phase*

First, consider the club quiche compactification with choice

$$\mathcal{A}_{\text{phys}} = \mathcal{A}_P. \qquad (V.105)$$

The reduced topological order is $\mathbb{Z}_3$ DW Gauge Theory, whose anyons are mapped to anyons of $S_3$ DW theory as

$$
\begin{aligned}
1 &\mapsto (1,1) \oplus (1,P), \\
e, e^2 &\mapsto (a,1), \\
m, m^2 &\mapsto (1,E), \\
em, e^2 m^2 &\mapsto (a,\omega), \\
em^2, e^2 m &\mapsto (a,\omega^2).
\end{aligned}
\qquad (V.106)
$$

We have the intersection $\mathcal{A}_P \cap \mathcal{A}_{\text{sym}} = 1$, meaning that the club quiche compactification results in an irreducible

topological boundary condition $\mathfrak{B}'$ of the $\mathbb{Z}_3$ DW Gauge Theory. Both $e$ and $e^2$ have a topological end along $\mathfrak{B}'$ as $(a,1)$ has a topological end along $\mathfrak{B}^{\text{sym}}_{\text{Rep}(S_3)^f}$. Thus $\mathfrak{B}'$ corresponds to Lagrangian algebra $\mathcal{A}_e = 1 \oplus e \oplus e^2$. From $\mathcal{A}_P$, we see that we only obtain a bosonic topological end of $(-1)^F$ along $\mathfrak{B}'$. This lets us recognize

$$\mathfrak{B}' = \mathfrak{B}_e, \qquad (V.107)$$

where $\mathfrak{B}_e$ is irreducible topological boundary condition corresponding to $\mathcal{A}_e$, without any additional Arf term. This in particular means that $(-1)^F$ is realized along $\mathfrak{B}'$ as

$$(-1)^F = 1. \qquad (V.108)$$

The boundary $\mathfrak{B}'$ realizes a $\mathbb{Z}_3$ symmetry

$$\mathcal{S}' = \text{Vec}_{\mathbb{Z}_3} = \{1, P, P^2\} \qquad (V.109)$$

The linking action of $E$ on the end of $(a,1)$ along $\mathfrak{B}^{\text{sym}}_{\text{Rep}(S_3)^f}$ can be computed as

$$\frac{S_{(1,E),(a,1)}|S_3|}{\dim(a,1)} = -1, \qquad (V.110)$$

where $S_{(1,E),(a,1)}$ on the left-hand side is the $((1,E),(a,1))$-component of the (unitary) modular $S$-matrix of the $S_3$ DW theory. This means that the realization of $E$ on $\mathfrak{B}'$ has $-1$ linking action on the ends of $e$ and $e^2$. Combining this with the fact that quantum dimension of $E$ is 2 implies that $E$ is realized by line operator

$$E = P \oplus P^2 \qquad (V.111)$$

along $\mathfrak{B}'$. Here, we used the fact that $P$ acts on the end of $e$ as $\omega = e^{2\pi i/3}$ and also used the equality $\omega + \omega^2 = -1$.

Completing the club sandwich by a bosonic gapless boundary, we obtain a gapless phase whose properties are obtained simply by translating the above club quiche analysis. In this gapless phase the charge $(1, P)$ has been condensed, which forces the charges

$$Q_C = \{(b, +), (b, -)\} \qquad (V.112)$$

to confine, since they are mutually non-local with $(1, P)$. The deconfined charges are captured by anyons of $\mathbb{Z}_3$ DW theory according to the map (V.106). Since the condensed charge $(1, P)$ is in the twisted sector, we obtain a gSPT phase for $\text{Rep}(S_3)^f$ symmetry. Given that we have untwisted sector operators in the charge $(a, 1)$, applying the map (V.106) we learn that we have untwisted sector operators in charges $e$ and $e^2$. This recognizes the symmetry acting faithfully in the IR of the gSPT phase to be $\mathbb{Z}_3$ as in (V.109). The way $\text{Rep}(S_3)^f$ acts in the IR via the $\mathbb{Z}_3$ symmetry is described in equations (V.108) and (V.111).

An actual CFT realizing the IR of this gapless phase is provided by the 3-state Potts model, which carries such

a $\mathbb{Z}_3$ symmetry. The $\mathsf{Rep}(S_3)^f$ symmetry is realized on it as described in equations (V.108) and (V.111).

From the point of view of the bosonized $\mathsf{Rep}(S_3)$ symmetry, $\mathcal{A}_P$ realizes the same IR physics with $P$ being regarded as a bosonic $\mathbb{Z}_2$ symmetry, which acts trivially in this gapless phase. The fermionization does not modify these results.

### 2. $(1, E)$-condensed phase

Now consider the club quiche compactification with choice

$$\mathcal{A}_{\text{phys}} = \mathcal{A}_E. \tag{V.113}$$

The reduced topological order is $\mathbb{Z}_2$ DW Gauge Theory, whose anyons are mapped to anyons of $S_3$ DW theory as

$$\begin{aligned} 1 &\mapsto (1, 1) \oplus (1, E), \\ m &\mapsto (1, P) \oplus (1, E), \\ e &\mapsto (b, +), \\ f &\mapsto (b, -). \end{aligned} \tag{V.114}$$

We have the intersection $\mathcal{A}_E \cap \mathcal{A}_{\text{sym}} = 1$, meaning that the club quiche compactification results in an irreducible topological boundary condition $\mathfrak{B}'$ of the $\mathbb{Z}_2$ DW Gauge Theory. The anyon $f$ has a topological end along $\mathfrak{B}'$ coming from the end of $(b, -)$ along $\mathfrak{B}^{\text{sym}}_{\mathsf{Rep}(S_3)^f}$. Thus $\mathfrak{B}'$ is a fermionic boundary condition corresponding to Lagrangian algebra $\mathcal{A}_f = 1 \oplus \pi f$. Since $(b, +)$ has a fermionic end along $\mathfrak{B}^{\text{sym}}_{\mathsf{Rep}(S_3)^f}$ in $(-1)^F$-twisted sector, $\mathfrak{B}'$ is completely fixed as the fermionic boundary along which $e$ has a fermionic end in $(-1)^F$-twisted sector. In other words, $e$ in the bulk is mapped to $\pi(-1)^F$ rather than $(-1)^F$ on the boundary $\mathfrak{B}'$. The boundary $\mathfrak{B}'$ realizes a $\mathbb{Z}_2^f$ symmetry

$$\mathcal{S}' = \mathsf{sVec}_{\mathbb{Z}_2} = \{1, \pi, (-1)^F, \pi(-1)^F\}. \tag{V.115}$$

The linking action of $E$ on the end of $(b, -)$ along $\mathfrak{B}^{\text{sym}}_{\mathsf{Rep}(S_3)^f}$ is computed as

$$\frac{S_{(1,E),(b,-)}|S_3|}{\dim(b, -)} = 0, \tag{V.116}$$

where $S_{(1,E),(b,-)}$ is again a component of the modular $S$-matrix. This means that the realization of $E$ on $\mathfrak{B}'$ has 0 linking action on the end of $f$. Combining this with the fact that quantum dimension of $E$ is 2 implies that $E$ is realized by line operator

$$E = 1 \oplus (-1)^F \tag{V.117}$$

along $\mathfrak{B}'$. We note the above equation is the unique solution for the fusion rule $E^2 = 1 \oplus (-1)^F \oplus E$, cf. eq. (II.42).

Completing the club sandwich by a bosonic gapless boundary, we obtain a gapless phase whose properties

are obtained simply by translating the above club quiche analysis. In this gapless phase the charge $(1, E)$ has been condensed, which forces the charges

$$Q_C = \{(a, 1), (a, \omega), (a, \omega^2)\} \tag{V.118}$$

to confine, since they are mutually non-local with $(1, E)$. The deconfined charges are captured by anyons of $\mathbb{Z}_2$ DW theory according to the map (V.114). Since the condensed charge $(1, E)$ only has operators in the twisted sector, we obtain a gSPT phase for $\mathsf{Rep}(S_3)^f$ symmetry. Given that we have untwisted sector fermionic operators transforming in the charge $(b, -)$, applying the map (V.114) we learn that we have untwisted sector fermionic operators in charge $f$. This recognizes the symmetry acting faithfully in the IR of the gSPT phase to be $\mathbb{Z}_2^f$ as in (V.115). The way $\mathsf{Rep}(S_3)^f$ acts in the IR via the $\mathbb{Z}_2^f$ symmetry is described in equation (V.117).

A concrete example of a CFT realizing the IR of such a gapless phase is the Maj CFT which carries $\mathcal{S}' = \mathbb{Z}_2^f$, and on which $\mathsf{Rep}(S_3)^f$ is realized according to (V.117).

From the point of view of bosonic $\mathsf{Rep}(S_3)$ symmetry, $\mathcal{A}_E$ realizes $\mathbb{Z}_2$ symmetric CFT with the $\mathbb{Z}_2$ subsymmetry of $\mathsf{Rep}(S_3)$ identified with this $\mathbb{Z}_2$. Fermionizing with respect to it, we indeed obtain $\mathbb{Z}_2^f$ symmetric CFT. Taking the $\mathbb{Z}_2$ symmetric CFT to be Ising, we recover the example of Maj CFT discussed above.

### 3. $(a, 1)$-condensed phase

Now consider the club quiche compactification with choice

$$\mathcal{A}_{\text{phys}} = \mathcal{A}_a. \tag{V.119}$$

The reduced topological order is $\mathbb{Z}_2$ DW Gauge Theory, whose anyons are mapped to anyons of $S_3$ DW theory as

$$\begin{aligned} 1 &\mapsto (1, 1) \oplus (a, 1), \\ e &\mapsto (1, P) \oplus (a, 1), \\ m &\mapsto (b, +), \\ f &\mapsto (b, -). \end{aligned} \tag{V.120}$$

The club quiche compactification results in a reducible topological boundary condition $\mathfrak{B}'$ of the $\mathbb{Z}_2$ DW Gauge Theory, which is a sum of two irreducible boundary conditions, as seen from the fact that $(a, 1)$ can end along both boundary $\mathfrak{B}^{\text{sym}}_{\mathsf{Rep}(S_3)^f}$ and interface $\mathcal{I}_{\text{phys}}$ associated to $\mathcal{A}_{\text{phys}}$. The anyons $f$ and $e$ both have a topological end along $\mathfrak{B}'$ coming from the ends of $(b, -)$ and $(a, 1)$ respectively along $\mathfrak{B}^{\text{sym}}_{\mathsf{Rep}(S_3)^f}$. Thus we can recognize $\mathfrak{B}'$ as

$$\mathfrak{B}' = \mathfrak{B}_e \oplus (\mathfrak{B}_f \boxtimes \text{Arf}) \tag{V.121}$$

Here $\mathfrak{B}_f$ is the topological boundary condition of $\mathbb{Z}_2$ DW theory associated to Lagrangian algebra $\mathcal{A}_f = 1 \oplus \pi f$

along which the $e$ line can end in a fermionic topological local operator in $(-1)^F$-twisted sector, but the fermionic boundary inside $\mathfrak{B}'$ is obtained by stacking an Arf term on $\mathfrak{B}_f$ since $e$ only has a bosonic R-sector topological end along $\mathfrak{B}'$ coming from such an end of $(a,1)$ along $\mathfrak{B}^{\text{sym}}_{\text{Rep}(S_3)^f}$. On the other hand, $\mathfrak{B}_e$ is a topological boundary condition of $\mathbb{Z}_2$ DW theory associated to Lagrangian algebra $\mathcal{A}_e = 1 \oplus e$. The boundary $\mathfrak{B}_e$ does not carry an Arf term, which can be seen by noting that the presence of $(a,1)$ in $\mathcal{A}_{\text{phys}}$ means that $\mathfrak{B}'$ carries a bosonic topological local operator in $(-1)^F$-twisted sector, which can only lie along $\mathfrak{B}_e$ and implies the absence of Arf term along $\mathfrak{B}_e$.

All of the topological line operators along $\mathfrak{B}'$ form a fermionic multi-fusion category $\mathcal{S}(\mathfrak{B}')$ with objects

$$\mathcal{S}(\mathfrak{B}') = \{1_{ee}, \pi_{ee}, P_{ee}, 1_{ff}, \pi_{ff}, (-1)^F_{ff}, S_{ef}, S_{fe}\}, \tag{V.122}$$

where $1_{ii}$ is the identity line along $\mathfrak{B}_i$, $\pi_{ii}$ is the $\pi$ line along $\mathfrak{B}_i$, $P_{ee}$ is the $\mathbb{Z}_2$ symmetry generator along $\mathfrak{B}_e$, $(-1)^F_{ff}$ is the $\mathbb{Z}_2^f$ symmetry generator along $\mathfrak{B}_f \boxtimes \text{Arf}$, $S_{ef}$ is a non-invertible line operator changing $\mathfrak{B}_e$ to $\mathfrak{B}_f \boxtimes \text{Arf}$, and $S_{fe}$ is a non-invertible line operator changing $\mathfrak{B}_f \boxtimes \text{Arf}$ to $\mathfrak{B}_e$. The fusion rules of $S_{ef}$ and $S_{fe}$ are

$$\begin{aligned} \pi_{ee} P_{ee} \otimes S_{ef} = S_{ef} \otimes (-1)^F_{ff} &= S_{ef}, \\ (-1)^F_{ff} \otimes S_{fe} = S_{fe} \otimes \pi_{ee} P_{ee} &= S_{fe}, \\ S_{ef} \otimes S_{fe} &= 1_{ee} \oplus \pi_{ee} P_{ee}, \\ S_{fe} \otimes S_{ef} &= 1_{ff} \oplus (-1)^F_{ff}. \end{aligned} \tag{V.123}$$

The above fusion rules are analogous to those of duality defects that implement the gauging of a non-anomalous $\mathbb{Z}_2$ symmetry. We recall that the fusion rules of the duality defects follow from the fact that gauging a $\mathbb{Z}_2$ symmetry is an operation to condense $1 \oplus P$ where $P$ is the generator of $\mathbb{Z}_2$. Similarly, the fusion rules (V.123) follow from the fact that $S_{fe}$ and $S_{ef}$ implement the GSO projection and its inverse, which are the operations to condense $1_{ff} \oplus (-1)^F_{ff}$ and $1_{ee} \oplus \pi_{ee} P_{ee}$ respectively.

Let $\mathcal{E}_e$ and $\mathcal{E}_f$ be the topological local operators lying at the ends of $e$ and $f$ along $\mathfrak{B}_e$ and $\mathfrak{B}_f \boxtimes \text{Arf}$ respectively, and let $v_e, v_f$ denote the identity operators along $\mathfrak{B}_e$ and $\mathfrak{B}_f \boxtimes \text{Arf}$ respectively. The linking action of $S_{ef}$ and $S_{fe}$ on these operators are

$$\begin{aligned} S_{ef} : \; &v_e \to \sqrt{2}\lambda v_f, \\ &\mathcal{E}_e \to 0, \\ S_{fe} : \; &v_f \to \sqrt{2}\lambda^{-1} v_e, \\ &\mathcal{E}_f \to 0, \end{aligned} \tag{V.124}$$

for some $\lambda \in \mathbb{R}^+$ which captures the relative Euler term between $\mathfrak{B}_e$ and $\mathfrak{B}_f \boxtimes \text{Arf}$. The action on $\mathcal{E}_e$ and $\mathcal{E}_f$ is zero because the lines $e$ and $f$ cannot end along $\mathfrak{B}_f \boxtimes \text{Arf}$ and $\mathfrak{B}_e$ respectively.

Performing a computation similar to the one performed in section IV.D.3 of [5], we find that the $\text{Rep}(S_3)^f$ sym-

metry of $\mathfrak{B}'$ is realized as

$$\begin{aligned} (-1)^F &= 1_{ee} \oplus (-1)^F_{ff}, \\ E &= S_{ef} \oplus S_{fe} \oplus \pi_{ee} P_{ee}. \end{aligned} \tag{V.125}$$

Given the quantum dimension of $E$, we know that the linking action of $E$ on $1 = v_e + v_f$ has to be

$$E : \; v_e + v_f \to 2(v_e + v_f), \tag{V.126}$$

which means that we have

$$\lambda = \sqrt{2}. \tag{V.127}$$

Completing the club sandwich by a bosonic gapless boundary, we obtain a gapless phase in which the charge $(a,1)$ has been condensed, which forces the charges

$$Q_C = \{(1,E), (a,\omega), (a,\omega^2)\} \tag{V.128}$$

to confine, since they are mutually non-local with $(a,1)$. The deconfined charges are captured by anyons of $\mathbb{Z}_2$ DW theory according to the map (V.120). Since the condensed charge $(a,1)$ has an operator in the untwisted sector, we obtain a gSSB phase for $\text{Rep}(S_3)^f$ symmetry with two universes. From the above club quiche analysis we learn that we can express the IR theory for a system lying in such a gapless phase as

$$\mathfrak{T}^{\text{IR}} = (\mathfrak{T}^{\text{IR}}_e \boxtimes \text{Triv}) \oplus (\mathfrak{T}^{\text{IR}}_f \boxtimes \text{Arf}) \tag{V.129}$$

where $\mathfrak{T}^{\text{IR}}_e$ is a theory with bosonic non-anomalous $\mathbb{Z}_2$ symmetry (whose generator is labeled by $P_{ee}$) and $\mathfrak{T}^{\text{IR}}_f$ is its fermionization with respect to this $\mathbb{Z}_2$ symmetry[22]

$$\mathfrak{T}^{\text{IR}}_f = \mathfrak{T}^{\text{IR}}_e \, /^f \, \mathbb{Z}_2. \tag{V.130}$$

We recall that $\mathfrak{T}^{\text{IR}}_f \boxtimes \text{Arf}$ and $\mathfrak{T}^{\text{IR}}_e$ are related via the GSO projection and its inverse, cf. Figure 1. The generator of fermionic $\mathbb{Z}_2^f$ symmetry of $\mathfrak{T}^{\text{IR}}_f \boxtimes \text{Arf}$ is labeled by $(-1)^F_{ff}$. The interfaces between the two theories implementing the GSO projection and its inverse are $S_{fe}$ and $S_{ef}$ respectively. Then the $\text{Rep}(S_3)^f$ is realized in the IR of this gapless phase as in equation (V.125).

Picking $\mathfrak{T}^{\text{IR}}_e$ to be Ising CFT implies that $\mathfrak{T}^{\text{IR}}_f = \text{Maj}$ and we obtain a concrete example for $\mathfrak{T}^{\text{IR}}$, which is

$$\mathfrak{T}^{\text{IR}} = \text{Ising} \oplus \text{Maj} \tag{V.131}$$

where we have used the fact that Maj CFT is invariant under stacking by Arf TFT.

From the point of view of bosonic $\text{Rep}(S_3)$ symmetry, $\mathcal{A}_a$ describes an IR system $\mathfrak{T} \oplus (\mathfrak{T}/\mathbb{Z}_2)$ where $\mathfrak{T}$ is a copy of a $\mathbb{Z}_2$ symmetric CFT, $\mathfrak{T}/\mathbb{Z}_2$ is the CFT obtained by gauging this $\mathbb{Z}_2$ symmetry, and the $\mathbb{Z}_2$ subsymmetry of $\text{Rep}(S_3)$ is realized as the $\mathbb{Z}_2$ symmetry of the $\mathfrak{T}$ factor in $\mathfrak{T} \oplus (\mathfrak{T}/\mathbb{Z}_2)$. Fermionizing with respect to this $\mathbb{Z}_2$ subsymmetry we obtain $(\mathfrak{T}/^f \mathbb{Z}_2) \oplus (\mathfrak{T}/\mathbb{Z}_2)$ which matches (V.129) with the identification $\mathfrak{T}^{\text{IR}}_e = \mathfrak{T}/\mathbb{Z}_2$. Picking $\mathfrak{T} = \text{Ising}$ leads to the example (V.131).

---

[22] This follows from the definition of $\mathfrak{B}_e$ and $\mathfrak{B}_f$.

### 4. SPT phase whose underlying TFT is Triv

Now let us consider gapped phases choosing first the physical Lagrangian algebra to be

$$\mathcal{A}_{\text{phys}} = \mathcal{A}_{P,E}, \qquad (\text{V.132})$$

which corresponds to condensing mutually local charges $(1,P)$ and $(1,E)$. The condensation of these charges forces the rest of the charges

$$Q_C = \{(a, \omega^p), (b, \pm)\} \qquad (\text{V.133})$$

to confine as they are mutually non-local with $(1,P)$ or $(1,E)$. Both the condensed charges contain only twisted sector operators, hence the resulting gapped phase has a unique ground state in the NS sector, i.e., it is an SPT phase for $\mathsf{Rep}(S_3)^f$ symmetry. The phase contains only a bosonic topological local operator in R sector coming from the condensation of $(1,P)$, which means that the underlying non-symmetric fermionic theory is the trivial theory. The $\mathsf{Rep}(S_3)^f$ symmetry is realized on it via

$$\begin{aligned} (-1)^F &= 1, \\ E &= 1 \oplus 1. \end{aligned} \qquad (\text{V.134})$$

The first line follows from the fact that the underlying fermionic TFT is trivial. The second line follows from the fusion rule $E^2 = 1 \oplus (-1)^F \oplus E$, cf. eq. (II.42).

This is fermionization of trivial phase of bosonic $\mathsf{Rep}(S_3)$ symmetry.

From the point of view of the club quiche associated to $\mathcal{A}_P$, this gapped phase is produced by choosing the physical boundary to be $\mathfrak{B}_m$ corresponding to Lagrangian algebra $\mathcal{A}_m = 1 \oplus m \oplus m^2$ of the $\mathbb{Z}_3$ DW theory, since ending $m$ and $m^2$ corresponds to ending $(1,E)$ according to equation (V.106). Recall that the topological boundary on the other side of the bulk $\mathbb{Z}_3$ DW theory is $\mathfrak{B}' = \mathfrak{B}_e$ as shown in eq. (V.107). It is known that the compactification of $\mathbb{Z}_3$ DW theory with $\mathfrak{B}_e$ and $\mathfrak{B}_m$ as two ends is indeed the trivial 2d TQFT on which the generator $P'$ of the $\mathbb{Z}_3$ symmetry is realized by $P' = 1$. Combining this with (V.108) and (V.111) we indeed obtain (V.134).[23]

We can also restate these results directly from the point of view of deformations of the gapless phase corresponding to $\mathcal{A}_P$ in which $(1,P)$ is already condensed and $(b, \pm)$ are already confined. The gapped phase corresponding to $\mathcal{A}_{P,E}$ is obtained by additionally condensing the charge $(1,E)$, leading to the confinement of remaining charges $(a, \omega^p)$. According to the map (V.106), we learn that the $(1,E)$ condensation corresponds to condensation of the charges $m$ and $m^2$ for the IR $\mathbb{Z}_3$ symmetry of the gapless phase. The $\mathbb{Z}_3$ symmetry after this condensation is realized as $P' = 1$ because the generator $P'$ originates from $m$ in the bulk, which is now condensed. Hence we obtain

_____________

[23] In eq. (V.111), the generator $P'$ of $\mathbb{Z}_3$ is written as $P$.

a gapped SPT phase for this $\mathbb{Z}_3$ symmetry on which the $\mathbb{Z}_3$ symmetry is realized as $P' = 1$. This again leads to (V.134).

Concretely this $\mathsf{Rep}(S_3)^f$ symmetric deformation is realized by the deformation of 3-Potts (which realizes the gapless phase $\mathcal{A}_P$) to the trivial gapped phase for $\mathbb{Z}_3$ symmetry.

On the other hand, from the point of view of the club quiche associated to $\mathcal{A}_E$, this gapped phase is produced by choosing the physical boundary to be $\mathfrak{B}_m$ corresponding to Lagrangian algebra $\mathcal{A}_m = 1 \oplus m$ of the $\mathbb{Z}_2$ DW theory, since ending $m$ corresponds to ending $(1,E)$ according to equation (V.114). Recall that the other boundary of $\mathbb{Z}_2$ DW theory is $\mathfrak{B}' = \mathfrak{B}_f$ as discussed in Section V D 2. It is known that the compactification of $\mathbb{Z}_2$ DW theory with $\mathfrak{B}_f$ and $\mathfrak{B}_m$ as two ends is indeed the trivial 2d TQFT on which the $\mathbb{Z}_2^f$ symmetry is realized by $(-1)^F = 1$. Combining this with (V.117) we indeed obtain (V.134).

We can also restate these results directly from the point of view of deformations of the gapless phase corresponding to $\mathcal{A}_E$ in which $(1,E)$ is already condensed and $(a, \omega^p)$ are already confined. The gapped phase corresponding to $\mathcal{A}_{P,E}$ is obtained by additionally condensing the charge $(1,P)$, leading to the confinement of remaining charges $(b, \pm)$. According to the map (V.114), we learn that the $(1,P)$ condensation corresponds to condensing the charge $m$ for the IR $\mathbb{Z}_2^f$ symmetry of the gapless phase, and hence we obtain a gapped SPT phase for this $\mathbb{Z}_2^f$ symmetry on which the $\mathbb{Z}_2^f$ symmetry is realized as $(-1)^F = 1$. This again leads to (V.134).

Concretely this $\mathsf{Rep}(S_3)^f$ symmetric deformation is realized by the deformation of Maj (which realizes the gapless phase $\mathcal{A}_E$) to the trivial phase for $\mathbb{Z}_2^f$ symmetry.

### 5. SPT phase whose underlying TFT is Arf

Now choose the physical Lagrangian algebra to be

$$\mathcal{A}_{\text{phys}} = \mathcal{A}_{E,b}, \qquad (\text{V.135})$$

which corresponds to condensing mutually local charges $(b, +)$ and $(1,E)$. This condensation forces the rest of the charges

$$Q_C = \{(1,P), (a, \omega^p)\} \qquad (\text{V.136})$$

to confine as they are mutually non-local with $(b, +)$ or $(1,E)$. Both the condensed charges contain only twisted sector operators, hence the resulting gapped phase is an SPT phase for $\mathsf{Rep}(S_3)^f$ symmetry. The phase contains only a fermionic topological local operator in R sector coming from the condensation of $(b, +)$, which means that the underlying non-symmetric fermionic theory is the Arf theory. The $\mathsf{Rep}(S_3)^f$ symmetry is realized on it via

$$\begin{aligned} (-1)^F &= \pi, \\ E &= 1 \oplus \pi. \end{aligned} \qquad (\text{V.137})$$

The first line follows from the fact that the underlying TFT of this SPT phase is the Arf TFT. The second line follows from the fusion rule $E^2 = 1 \oplus (-1)^F \oplus E$.

This is fermionization of $\mathbb{Z}_2$ SSB phase of bosonic $\mathsf{Rep}(S_3)$ symmetry in which $P \in \mathsf{Rep}(S_3)$ acts by exchanging the two vacua involved and $E \in \mathsf{Rep}(S_3)$ is realized as $E = 1 \oplus P$. Fermionizing with respect to $P$, we obtain Arf TFT on which $(-1)^F = \pi$ and $E = 1 \oplus \pi$, reproducing what we discussed above.

From the point of view of the club quiche associated to $\mathcal{A}_E$, this gapped phase is produced by choosing the physical boundary to be $\mathfrak{B}_e$ corresponding to Lagrangian algebra $\mathcal{A}_e = 1 \oplus e$ of the $\mathbb{Z}_2$ DW theory, since ending $e$ corresponds to ending $(b, +)$ according to equation (V.114). Recall that the other boundary of $\mathbb{Z}_2$ DW theory is $\mathfrak{B}' = \mathfrak{B}_f$, and it is known that the compactification of $\mathbb{Z}_2$ DW theory with $\mathfrak{B}_f$ and $\mathfrak{B}_e$ as two ends is indeed the 2d Arf TFT on which the $\mathbb{Z}_2^f$ symmetry is realized by $(-1)^F = \pi$. Combining this with (V.117) we indeed obtain (V.137).

We can also restate these results directly from the point of view of deformations of the gapless phase corresponding to $\mathcal{A}_E$ in which $(1, E)$ is already condensed and $(a, \omega^p)$ are already confined. The gapped phase corresponding to $\mathcal{A}_{E,b}$ is obtained by additionally condensing the charge $(b, +)$, leading to the confinement of remaining charges $(b, -)$ and $(1, P)$. According to the map (V.114), we learn that the $(b, +)$ condensation corresponds to condensing the charge $e$ for the IR $\mathbb{Z}_2^f$ symmetry of the gapless phase. Since $e$ is now condensed, the $\mathbb{Z}_2^f$ symmetry is realized as $\pi(-1)^F = 1$, i.e., $(-1)^F = \pi$. Here, we recall that $e$ is mapped to $\pi(-1)^F$ on the boundary $\mathfrak{B}_f$. Hence we obtain a gapped SPT phase for this $\mathbb{Z}_2^f$ symmetry on which the $\mathbb{Z}_2^f$ symmetry is realized as $(-1)^F = \pi$. This again leads to (V.137).

Concretely this $\mathsf{Rep}(S_3)^f$ symmetric deformation is realized by the deformation of Maj (which realizes the gapless phase $\mathcal{A}_E$) to the Arf phase for $\mathbb{Z}_2^f$ symmetry.

### 6. SSB phase whose underlying TFT is $Triv \oplus Triv \oplus Triv$

Now choose the physical Lagrangian algebra to be

$$\mathcal{A}_{\text{phys}} = \mathcal{A}_{P,a}, \tag{V.138}$$

which corresponds to condensing mutually local charges $(1, P)$ and $(a, 1)$. This condensation forces the rest of the charges

$$Q_C = \{(1, E), (a, \omega), (a, \omega^2), (b, \pm)\} \tag{V.139}$$

to confine as they are mutually non-local with $(1, P)$ or $(a, 1)$. Moreover, since the coefficient of $(a, 1)$ in $\mathcal{A}_{P,a}$ is two, this phase contains two linearly independent multiplets of operators with charge $(a, 1)$. Since a multiplet with charge $(a, 1)$ contains an untwisted sector operator, we obtain two linearly independent non-identity local operators in the IR. Hence the resulting gapped phase has

3 vacua in total. A multiplet with charge $(a, 1)$ also contains a bosonic operator in R sector, and an operator with charge $(1, P)$ is also a bosonic R-sector operator. In total, we have three linearly independent bosonic R-sector operators, implying that each vacuum of the theory is a copy of trivial fermionic 2d TFT in which $(-1)^F$ is realized by the identity line. Labeling the vacua as $v_i$ for $i \in \{0, 1, 2\}$, we can express the realization of $(-1)^F$ as

$$(-1)^F = 1 = 1_{00} \oplus 1_{11} \oplus 1_{22}, \tag{V.140}$$

where $1_{ii}$ is the identity line in vacuum $i$.

The calculation of the realization of the $E$ symmetry follows the same argument as in section 5.3.3 of [3], leading to

$$E = 1_{01} \oplus 1_{02} \oplus 1_{12} \oplus 1_{10} \oplus 1_{20} \oplus 1_{21}, \tag{V.141}$$

where $1_{ij}$ is the unit interface from vacuum $i$ to vacuum $j$. Moreover, this argument also implies that there are no non-trivial relative Euler terms between the three vacua.

This is the fermionization of $\mathsf{Rep}(S_3)/\mathbb{Z}_2$ SSB phase of bosonic $\mathsf{Rep}(S_3)$ symmetry in which $P \in \mathsf{Rep}(S_3)$ acts trivially. Hence fermionization does not change the theory.

From the point of view of the club quiche associated to $\mathcal{A}_P$, this gapped phase is produced by choosing the physical boundary to be $\mathfrak{B}_e$ corresponding to Lagrangian algebra $\mathcal{A}_e = 1 \oplus e \oplus e^2$ of the $\mathbb{Z}_3$ DW theory, since ending $e$ and $e^2$ corresponds to ending $(a, 1)$ according to equation (V.106). Recall that the other boundary of $\mathbb{Z}_3$ DW theory is $\mathfrak{B}' = \mathfrak{B}_e$. It is known that the compactification of $\mathbb{Z}_3$ DW theory with $\mathfrak{B}_e$ on both ends is a 2d TQFT with three vacua along with trivial relative Euler terms between them. This is the completely broken phase for the IR $\mathbb{Z}_3$ symmetry, whose generator $P'$ is realized by

$$P' = 1_{01} \oplus 1_{12} \oplus 1_{20}. \tag{V.142}$$

Combining this with (V.108) and (V.111) we indeed obtain (V.140) and (V.141).

We can also restate these results directly from the point of view of deformations of the gapless phase corresponding to $\mathcal{A}_P$ in which $(1, P)$ is already condensed and $(b, \pm)$ are already confined. The gapped phase corresponding to $\mathcal{A}_{P,a}$ is obtained by additionally condensing the charge $(a, 1)$, leading to the confinement of remaining charges $(1, E), (a, \omega), (a, \omega^2)$. According to the map (V.106), we learn that the $(a, 1)$ condensation corresponds to condensation of the charges $e$ and $e^2$ for the IR $\mathbb{Z}_3$ symmetry of the gapless phase, and hence we obtain a gapped $\mathbb{Z}_3$ SSB phase for this $\mathbb{Z}_3$ symmetry, leading again to (V.140) and (V.141).

Concretely this $\mathsf{Rep}(S_3)^f$ symmetric deformation is realized by the deformation of 3-Potts (which realizes the gapless phase $\mathcal{A}_P$) to the $\mathbb{Z}_3$ SSB phase.

From the point of view of the club quiche associated to $\mathcal{A}_a$, this gapped phase is produced by choosing the physical boundary to be $\mathfrak{B}_e$ corresponding to Lagrangian

algebra $\mathcal{A}_e = 1 \oplus e$ of the $\mathbb{Z}_2$ DW theory, since ending $e$ corresponds to ending $(1, P)$ according to equation (V.120). Recall that the other boundary of $\mathbb{Z}_2$ DW theory is $\mathfrak{B}' = \mathfrak{B}_e \oplus (\mathfrak{B}_f \boxtimes \mathrm{Arf})$ with a non-trivial relative Euler term, and thus the compactification of $\mathbb{Z}_2$ DW theory with $\mathfrak{B}'$ on one end and $\mathfrak{B}_e$ on the other end decomposes into a compactification with $\mathfrak{B}_e$ on both ends and a compactification with $(\mathfrak{B}_f \boxtimes \mathrm{Arf}, \mathfrak{B}_e)$ on the two ends. The first compactification results in two vacua $v_0, v_1$ on which the $\mathbb{Z}_2$ symmetry of $\mathfrak{B}_e \subset \mathfrak{B}'$ is spontaneously broken and hence the lines living along $\mathfrak{B}_e \subset \mathfrak{B}'$ are realized as

$$1_{ee} = 1_{00} \oplus 1_{11},$$
$$P_{ee} = 1_{01} \oplus 1_{10}. \tag{V.143}$$

The second compactification results in a single vacuum $v_2$ on which the symmetry $(-1)^F_{ff}$ of $\mathfrak{B}_f \boxtimes \mathrm{Arf} \subset \mathfrak{B}'$ is realized as

$$(-1)^F_{ff} = 1_{22}. \tag{V.144}$$

The boundary changing lines $S_{ef}, S_{fe}$ in $\mathfrak{B}'$ are realized as

$$S_{ef} = \pi_{02} \oplus 1_{12},$$
$$S_{fe} = \pi_{20} \oplus 1_{21}. \tag{V.145}$$

Combining these with (V.125) we recover (V.140) and (V.141).[24] In order to see that there are no relative Euler terms between the three vacua, note that the linking action of $P_{ee}$ on $v_0 + v_1$ being 1 implies that there is no relative Euler term between $v_0$ and $v_1$. On the other hand, the linking action of $S_{ef}$ on $v_0 + v_1$ from (V.124) has to be (using $\lambda = \sqrt{2}$ as in eq. (V.127))

$$S_{ef} : \ v_0 + v_1 \to 2v_2 \tag{V.146}$$

which implies that $1_{01}$ and $1_{02}$ both have quantum dimensions 1. Thus there are no relative Euler terms between the three vacua.

We can also restate these results directly from the point of view of deformations of the gapless phase corresponding to $\mathcal{A}_a$ in which $(a, 1)$ is already condensed and $(1, E), (a, \omega), (a, \omega^2)$ are already confined. The gapped phase corresponding to $\mathcal{A}_{P,a}$ is obtained by additionally condensing the charge $(1, P)$, leading to the confinement of remaining charges $(b, \pm)$. According to the map (V.120), we learn that the $(1, P)$ condensation corresponds to condensation of the charge $e$ from the perspective of the IR of the gapless phase. This deforms $\mathfrak{T}^{\mathrm{IR}}_e$ in (V.129) to a $\mathbb{Z}_2$ SSB phase with two vacua and $\mathfrak{T}^{\mathrm{IR}}_f$

in (V.129) to an Arf factor. Hence from $\mathfrak{T}^{\mathrm{IR}}_f \boxtimes \mathrm{Arf}$ we obtain another trivial vacuum. As discussed above, this leads to (V.140) and (V.141).

Concretely this $\mathsf{Rep}(S_3)^f$ symmetric deformation is realized by the deformation of $\mathsf{Ising} \oplus \mathsf{Maj}$ (which realizes the gapless phase $\mathcal{A}_a$) in which $\mathsf{Ising}$ is deformed to $\mathbb{Z}_2$ SSB phase and $\mathsf{Maj}$ is deformed to Triv phase for $\mathbb{Z}_2^f$ symmetry.

### 7.  SSB phase whose underlying TFT is Triv $\oplus$ Arf

Finally, choose the physical Lagrangian algebra to be

$$\mathcal{A}_{\mathrm{phys}} = \mathcal{A}_{a,b}, \tag{V.147}$$

which corresponds to condensing mutually local charges $(b, +)$ and $(a, 1)$. This condensation forces the rest of the charges

$$Q_{P,a} = \{(1, P), (1, E), (a, \omega), (a, \omega^2), (b, -)\} \tag{V.148}$$

to confine as they are mutually non-local with $(b, +)$ or $(a, 1)$. Since the coefficient of $(a, 1)$ in $\mathcal{A}_{a,b}$ is one, this phase contains a single multiplet of operators with charge $(a, 1)$, and hence this gapped phase has 2 vacua. Note that a multiplet of operators with charge $(b, +)$ does not contain any untwisted operators. A multiplet with charge $(a, 1)$ also contains a bosonic operator in the R sector, and a multiplet with charge $(b, +)$ contains a fermionic R-sector operator. This implies that one of the vacua, labeled $v_0$, carries a copy of trivial fermionic 2d TFT in which $(-1)^F$ is realized by the identity line, and the other vacuum, labeled $v_1$, carries a copy of the Arf TFT in which $(-1)^F$ is realized by the $\pi$ line. We can thus express the realization of $(-1)^F$ as

$$(-1)^F = 1_{00} \oplus \pi_{11}, \tag{V.149}$$

where $1_{ii}$ is the identity line in vacuum $i$ and $\pi_{ii}$ is the $\pi$ line in vacuum $i$.

The full set of topological lines in this 2d TFT Triv $\oplus$ Arf is

$$1_{ii}, \quad \pi_{ii}, \quad S_{01}, \quad S_{10}, \tag{V.150}$$

where $S_{ij}$ are lines changing vacuum $i$ to another vacuum $j$. The lines $S_{ij}$ are of $q$-type

$$\pi_{ii} \otimes S_{ij} = S_{ij} \otimes \pi_{jj} = S_{ij} \tag{V.151}$$

and their compositions are

$$S_{ij} \otimes S_{ji} = 1_{ii} \oplus \pi_{ii}. \tag{V.152}$$

The linking actions of these lines are

$$S_{01} : \ v_0 \to \sqrt{2}\lambda v_1,$$
$$S_{10} : \ v_1 \to \sqrt{2}\lambda^{-1} v_0, \tag{V.153}$$

---

[24] Actually, from eqs. (V.143)–(V.145) together with (V.125), we obtain $E = \pi_{01} \oplus \pi_{10} \oplus \pi_{02} \oplus \pi_{20} \oplus 1_{12} \oplus 1_{21}$, which agrees with eq. (V.141) upon redefining $1_{0i}$ and $1_{i0}$ by $\pi_{0i}$ and $\pi_{i0}$ for $i = 1, 2$. Such a redefinition is allowed because it preserves the operator algebra $1_{ij} 1_{kl} = \delta_{jk} 1_{il}$, cf. discussions around eq. (V.65).

where $\lambda \in \mathbb{R}^+$ captures the relative Euler term between the vacua $v_0$ and $v_1$, cf. eq. (V.10).

The calculation of the realization of the $E$ line follows the same argument as in section IV.D.3 of [5], leading to

$$E = S_{01} \oplus S_{10} \oplus \pi_{00}, \qquad (V.154)$$

which is consistent with the fusion rule $E^2 = 1 \oplus (-1)^F \oplus E$. Imposing that the quantum dimension of $E$ is 2 determines the relative Euler term to be

$$\lambda = \sqrt{2}. \qquad (V.155)$$

This is the fermionization of $\mathsf{Rep}(S_3)$ SSB phase of bosonic $\mathsf{Rep}(S_3)$ symmetry which carries 3 vacua, which form a $\mathrm{Triv} \oplus \mathbb{Z}_2$ SSB phase from the point of view of $\mathbb{Z}_2$ subsymmetry of $\mathsf{Rep}(S_3)$. Hence fermionization leads to a $\mathrm{Arf} \oplus \mathrm{Triv}$ phase.

From the point of view of the club quiche associated to $\mathcal{A}_a$, this gapped phase is produced by choosing the physical boundary to be $\mathfrak{B}_m$ corresponding to Lagrangian algebra $\mathcal{A}_m = 1 \oplus m$ of the $\mathbb{Z}_2$ DW theory, since ending $m$ corresponds to ending $(b, +)$ according to equation (V.120). Recall that the other boundary of $\mathbb{Z}_2$ DW theory is $\mathfrak{B}' = \mathfrak{B}_e \oplus (\mathfrak{B}_f \boxtimes \mathrm{Arf})$ with a non-trivial relative Euler term (V.127). Thus, the compactification of $\mathbb{Z}_2$ DW theory with $\mathfrak{B}'$ on one end and $\mathfrak{B}_m$ on the other end decomposes into a compactification with $(\mathfrak{B}_e, \mathfrak{B}_m)$ on the two ends and a compactification with $(\mathfrak{B}_f \boxtimes \mathrm{Arf}, \mathfrak{B}_m)$ on the two ends. The first compactification results in a trivial vacuum $v_0$ on which the $\mathbb{Z}_2$ symmetry of $\mathfrak{B}_e \subset \mathfrak{B}'$ is spontaneously unbroken and hence the lines living along $\mathfrak{B}_e \subset \mathfrak{B}'$ are realized as

$$\begin{aligned} 1_{ee} &= 1_{00}, \\ P_{ee} &= 1_{00}. \end{aligned} \qquad (V.156)$$

The second compactification results in a single Arf vacuum $v_1$ on which the symmetry $(-1)^F_{ff}$ of $\mathfrak{B}_f \boxtimes \mathrm{Arf} \subset \mathfrak{B}'$ is realized as

$$(-1)^F_{ff} = \pi_{11}. \qquad (V.157)$$

The boundary changing lines $S_{ef}, S_{fe}$ in $\mathfrak{B}'$ are realized as $S_{01}$ and $S_{10}$ respectively

$$\begin{aligned} S_{ef} &= S_{01}, \\ S_{fe} &= S_{10}. \end{aligned} \qquad (V.158)$$

Combining these with (V.125) we recover (V.149) and (V.154). The relative Euler term (V.155) in the gapped phase simply descends from the relative Euler term (V.127) in the gapless phase.

Concretely this $\mathsf{Rep}(S_3)^f$ symmetric deformation is realized by the deformation of $\mathsf{Ising} \oplus \mathsf{Maj}$ (which realizes the gapless phase $\mathcal{A}_a$) in which $\mathsf{Ising}$ is deformed to $\mathrm{Triv}$ phase for $\mathbb{Z}_2$ symmetry and $\mathsf{Maj}$ is deformed to $\mathrm{Arf}$ phase for $\mathbb{Z}_2^f$ symmetry.

## E. $\mathbb{Z}_2 \times \mathbb{Z}_2^f$ with a Gu-Wen Anomaly

As described in Sec. III E, the SymTFT for systems with a fermionic symmetry $\mathbb{Z}_2 \times \mathbb{Z}_2^f$ with a Gu-Wen anomaly $\nu = 2, 6 \mod 8$ is the $\mathbb{Z}_4$ Dijkgraaf-Witten TFT with the topological action $\omega = 2 \in H^3(\mathbb{Z}_4, U(1)) = \mathbb{Z}_4$, whose anyon content is

$$\mathcal{Z}(\mathsf{Vec}_{Z_4}^\omega) = \{e^a m^b \mid a, b = 0, 1, 2, 3\}. \qquad (V.159)$$

The SymTFT has a single fermionic topological boundary $\mathfrak{B}_{\mathcal{S}_f}^{\mathrm{sym}}$ (up to Arf term) corresponding to the algebra

$$\mathcal{A}_{m^2, e^2} = 1 \oplus \pi m^2 \oplus e^2 \oplus \pi e^2 m^2. \qquad (V.160)$$

The fermionic symmetry on this boundary is generated by $\eta$ and $P$ which are obtained via the bulk-to-boundary functor for the fermionic Lagrangian algebra $\mathcal{A}_{m^2, e^2}$ as

$$F(e) = P, \qquad F(m) = \eta. \qquad (V.161)$$

These lines have the fusion rules

$$P^2 = 1, \qquad \eta^2 = \pi. \qquad (V.162)$$

We may choose $(-1)^F = P$ or $(-1)^F = \pi P$. We choose $(-1)^F = P$ corresponding to which, the generalized charges were described in Sec. IV B 4.

We now describe the different phases by studying the different bosonic algebras and their corresponding condensed charges. There are a total of three algebras

$$\begin{aligned} \mathcal{A}_e &= 1 \oplus e \oplus e^2 \oplus e^3, \\ \mathcal{A}_{em^2} &= 1 \oplus em^2 \oplus e^2 \oplus e^3 m^2, \\ \mathcal{A}_{e^2} &= 1 \oplus e^2, \end{aligned} \qquad (V.163)$$

from which $\mathcal{A}_e$ and $\mathcal{A}_{em^2}$ are Lagrangian and therefore correspond to gapped phases. These condensable algebras can be organized into the following Hasse diagram

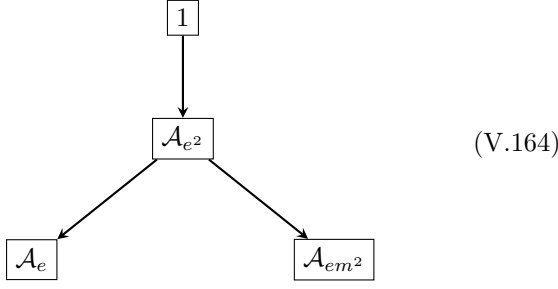

$$(V.164)$$

### 1. $\mathbb{Z}_2$ SSB phase

Let us consider the Lagrangian algebra $\mathcal{A}_e$ as the physical boundary. We denote the IR images of the operators with generalized charge $e^i$ as $\mathcal{O}_{e^i}$. Since the bulk line $e^2$ can end on both boundaries, we obtain two local topological operators in the IR, which are $\mathcal{O}_1 = 1$ and $\mathcal{O}_{e^2}$. Consequently there are two vacua with idempotents

$$v_0 = \frac{1 + \mathcal{O}_{e^2}}{2}, \quad v_1 = \frac{1 - \mathcal{O}_{e^2}}{2}. \qquad (V.165)$$

The IR images of the remaining order parameters, are both in the $P = (-1)^F$ twisted (i.e., R) sector. These carry charges $q_\eta = i$ and $-i$ respectively and are both bosonic. The linear combinations

$$\mathcal{O}_{e,0} = \frac{1}{2}\left(\mathcal{O}_e + \mathcal{O}_{e^3}\right),$$
$$\mathcal{O}_{e,1} = \frac{i}{2}\left(\mathcal{O}_e - \mathcal{O}_{e^3}\right), \tag{V.166}$$

satisfy the property

$$\mathcal{O}_{e,i}v_j = \delta_{ij}\mathcal{O}_{e,i}, \qquad \mathcal{O}_{e,i}\mathcal{O}_{e,j} = \delta_{ij}v_i. \tag{V.167}$$

The operators $\mathcal{O}_{e,i}$ can therefore be regarded as the end of the $(-1)^F$ line in the vacua $v_i$. Since the endoints in both vacua are bosonic, the fermion parity operator acts trivially as

$$(-1)^F = 1_{00} \oplus 1_{11}. \tag{V.168}$$

From the perspective of $\mathbb{Z}_2^f$, this is the decomposable phase

$$\text{Triv} \oplus \text{Triv}. \tag{V.169}$$

As for the $\eta$ symmetry, it acts as

$$\eta : \mathcal{O}_{e^j} \to i^j \mathcal{O}_{e^j}, \tag{V.170}$$

from which it follows that

$$\begin{aligned} \eta : \quad v_i &\longmapsto \quad v_{i+1 \bmod 2}, \\ \eta : \mathcal{O}_{e,i} &\longmapsto (-1)^i \mathcal{O}_{e,i+1 \bmod 2} \end{aligned} \tag{V.171}$$

As the two vacua are exchanged under $\mathbb{Z}_2^\eta$, this phase is referred to as a $\mathbb{Z}_2$ SSB phase. $\eta^2$ acts as -1 on the R sector and +1 on the NS sector, and therefore satisfies the group relation $\eta^2 = \pi$. The $\eta$ symmetry is represented as

$$\eta = 1_{01} \oplus \pi_{10}. \tag{V.172}$$

### 2. $\mathbb{Z}_2^{Arf}$ SSB phase

Let us consider the Lagrangian algebra $\mathcal{A}_{em^2}$ as the physical boundary. The analysis is almost identical to the previous case. Again we denote the IR image of the order parameters as $\mathcal{O}_{(em^2)^i}$ which satisfy the $\mathbb{Z}_4$ composition rules

$$\mathcal{O}_{(em^2)^i} \times \mathcal{O}_{(em^2)^j} = \mathcal{O}_{(em^2)^{i+j}}. \tag{V.173}$$

There are two local operators $\mathcal{O}_1$ and $\mathcal{O}_{e^2}$ in terms of which the vacua are defined as (V.165). Further we analogously define the linear combinations of the $P$-twisted operators

$$\mathcal{O}_{m^2,0} = \frac{1}{2}\left(\mathcal{O}_{em^2} + \mathcal{O}_{e^3m^2}\right),$$
$$\mathcal{O}_{m^2,1} = \frac{i}{2}\left(\mathcal{O}_{em^2} - \mathcal{O}_{e^3m^2}\right), \tag{V.174}$$

that satisfy the properties

$$\begin{aligned} \mathcal{O}_{m^2,i}v_j &= \delta_{ij}\mathcal{O}_{m^2,i}. \\ \mathcal{O}_{m^2,i}\mathcal{O}_{m^2,J} &= \delta_{ij}v_i. \end{aligned} \tag{V.175}$$

and can therefore be regarded as the end of the $(-1)^F$ line in the vacua $v_i$. Unlike the previous case, both $\mathcal{O}_{em^2}$ and $\mathcal{O}_{e^3m^2}$ are fermionic therefore the fermion parity operator acts as

$$(-1)^F = \pi_{00} \oplus \pi_{11}. \tag{V.176}$$

From the perspective of $\mathbb{Z}_2^f$, this is the decomposable phase

$$\text{Arf} \oplus \text{Arf}. \tag{V.177}$$

The $\eta$ symmetry acts as

$$\eta : \mathcal{O}_{(em^2)^j} \to (-i)^j \mathcal{O}_{(em^2)^j}, \tag{V.178}$$

from which it follows that

$$\eta : v_1 \longleftrightarrow v_2, \tag{V.179}$$

and on the twisted sector order parameters, it acts as

$$\eta : \mathcal{O}_{m^2,i} \longmapsto (-1)^{i+1} \mathcal{O}_{m^2,i+1 \bmod 2}. \tag{V.180}$$

Again we find that $\eta^2$ is +1 and -1 in the NS and R sector respectively. In this phase, $\eta$ is represented as

$$\eta = \pi_{01} \oplus 1_{10}. \tag{V.181}$$

### 3. $e^2$-condensed gapless phase

We now describe the gapless phase defined via the non-maximal condensable algebra $\mathcal{A}_{e^2}$ condensed. This algebra implements a bosonic interface $\mathcal{I}_{e^2}$ between $\mathcal{Z}(\text{Vec}_{\mathbb{Z}_4}^\omega)$ and $\mathcal{Z}(\text{Vec}_{\mathbb{Z}_2})$ which is the Toric Code. Notice that since $e^2$ can end on both the interface and on the fermionic boundary $\mathfrak{B}_{\mathcal{S}_f}^{\text{sym}}$ corresponding to the fermionic Lagrangian algebra $\mathcal{A}_{m^2,e^2}$. Therefore the boundary obtained by compactifying the region occupied by $\mathcal{Z}(\text{Vec}_{\mathbb{Z}_4}^\omega)$ (denoted as $\mathfrak{B}'$) contains two local topological operators. In other words, this boundary decomposes into a direct sum of irreducible topological boundary conditions

$$\mathfrak{B}' = \mathfrak{B}_0' \oplus \mathfrak{B}_1'. \tag{V.182}$$

Let us denote the anyon content of the Toric code as $\{1, e', m', f'\}$. Then the interface provides a map of topological lines from $\mathcal{Z}(\text{Vec}_{\mathbb{Z}_2})$ to $\mathcal{Z}(\text{Vec}_{\mathbb{Z}_4}^\omega)$ under which

$$\begin{aligned} 1 &\mapsto 1 \oplus e^2 \\ e' &\mapsto e \oplus e^3, \\ m' &\mapsto em^2 \oplus e^3m^2, \\ f' &\mapsto m^2e^2 \oplus m^2. \end{aligned} \tag{V.183}$$

We denote the bulk lines of the Toric code with primes. In what follows, we also denote the operators on the boundary $\mathfrak{B}'$ with primed labels. From the above map and the form of $\mathcal{A}_{m^2,e^2}$, it follows that $\pi f'$ has two ends on $\mathfrak{B}'$ corresponding to the ends of $m^2$ and $e^2 m^2$ on the $\mathfrak{B}^{\text{sym}}_{\mathcal{S}_f}$. We denote these ends as $\mathcal{E}'_{m^2}$ and $\mathcal{E}'_{e^2 m^2}$ respectively. Additionally, there is a topological local operator $\mathcal{O}'_{e^2}$ obtained from compactifying the $e^2$ extending between $\mathcal{I}_{e^2}$ and $\mathfrak{B}$. The algebra of these operators is

$$\mathcal{E}'_{e^p m^2} \times \mathcal{E}'_{e^q m^2} = \mathcal{O}'_{e^{p+q}} \,,$$
$$\mathcal{O}'_{e^p} \times \mathcal{E}'_{e^q m^2} = \mathcal{E}'_{e^{p+q} m^2} \,, \qquad (\text{V.184})$$
$$\mathcal{O}'_{e^p} \times \mathcal{O}'_{e^q} = \mathcal{O}'_{e^{p+q}} \,,$$

where $\mathcal{O}_{e^0} = 1$. The identity operators on the two decomposed boundaries $\mathfrak{B}'_0$ and $\mathfrak{B}'_1$ are

$$v'_0 = \frac{1 + \mathcal{O}'_{e^2}}{2} \,, \quad v'_1 = \frac{1 - \mathcal{O}'_{e^2}}{2} \,. \qquad (\text{V.185})$$

We also define the fermionic ends on the two decomposed boundaries $\mathfrak{B}'_0$ and $\mathfrak{B}'_1$ as

$$\mathcal{E}'_0 = \frac{\mathcal{E}'_{m^2} + \mathcal{E}'_{e^2 m^2}}{2} \,, \quad \mathcal{E}'_1 = \frac{\mathcal{E}'_{m^2} - \mathcal{E}'_{e^2 m^2}}{2} \,. \qquad (\text{V.186})$$

The local and fermionic ends satisfy

$$v'_i v'_j = \delta_{ij} v'_i \,, \quad \mathcal{E}'_i v'_j = \delta_{ij} \mathcal{E}'_i \,, \quad \mathcal{E}'_i \mathcal{E}'_j = \delta_{ij} v'_i \,, \quad (\text{V.187})$$

implying that both $\mathfrak{B}'_0$ and $\mathfrak{B}'_1$ are indecomposable fermionic topological boundaries of the Toric code. The symmetry category on the fermionic boundary of the Toric code is $\mathbb{Z}_2^f$ such that the bulk to boundary projection is $m \mapsto P$, $e \mapsto \pi P$ and $f \mapsto \pi$. For the present case, we should choose $P = \pi(-1)^F$. Since the boundary $\mathfrak{B}'$ decomposes as (V.182), correspondingly, the bulk projection of all the lines also split as

$$1' \longmapsto v'_0 \oplus v'_1 \,,$$
$$m' \longmapsto m'_0 \oplus m'_1 \,,$$
$$e' \longmapsto e'_0 \oplus e'_1 \,, \qquad (\text{V.188})$$
$$f' \longmapsto \mathcal{E}'_0 \oplus \mathcal{E}'_1 \,.$$

Let the end point of a line $e^a m^b \in \mathcal{Z}(\mathsf{Vec}(\mathbb{Z}_4^\omega))$ on $\mathfrak{B}'$ be denoted as $\mathcal{O}'_{e^a m^b}$. In terms of the lines in these lines, we define the operators

$$\mathcal{O}'_{e_0} = \frac{1}{2} \left( \mathcal{O}'_e + \mathcal{O}'_{e^3} \right) \,,$$
$$\mathcal{O}'_{e_1} = \frac{i}{2} \left( \mathcal{O}'_e - \mathcal{O}'_{e^3} \right) \,,$$
$$\mathcal{O}'_{m_0} = \frac{1}{2} \left( \mathcal{O}'_{e m^2} + \mathcal{O}'_{e^3 m^2} \right) \,, \qquad (\text{V.189})$$
$$\mathcal{O}'_{m_1} = \frac{i}{2} \left( \mathcal{O}'_{e m^2} - \mathcal{O}'_{e^3 m^2} \right) \,.$$

Note that boundary operators are none other than $\mathcal{O}_{e,i}$ (c.f. V.166) and $\mathcal{O}_{m^2,i}$ (c.f V.174) that became the

R-sector order parameters in the two indecomposable gapped phases realized for this symmetry. These operator satisfy the fusion rules

$$\mathcal{O}'_{m_i} \times \mathcal{O}'_{m_j} = \mathcal{O}'_{e_i} \times \mathcal{O}'_{e_j} = \delta_{i,j} v'_i \,,$$
$$\mathcal{O}'_{m_i} \times \mathcal{O}'_{e_j} = \delta_{i,j} \mathcal{E}'_i \,,$$
$$\mathcal{O}'_{m_i} \times \mathcal{E}'_j = \delta_{i,j} \mathcal{O}'_{e_i} \,, \qquad (\text{V.190})$$
$$\mathcal{O}'_{e_i} \times \mathcal{E}'_j = \delta_{i,j} \mathcal{O}'_{m_i} \,.$$

In other words $\{v'_i, \mathcal{O}'_{e_i}, \mathcal{O}'_{m_i}, \mathcal{E}'_i\}$ for $i = 0, 1$ satisfy fusion rules representative of the Toric code. The operators, $\mathcal{O}'_{e'_i}$ are twisted (i.e., R) sector bosons while $\mathcal{O}'_{m_i}$ are twisted sector fermions. Meanwhile $\mathcal{E}'_i$ are untwisted (NS) sector fermions.

Now we may define the symmetry action on $\mathfrak{B}'$. Firstly, $(-1)^F$ is implemented via the projection of the Toric code line $e'$ and is realized on the two decoupled copies diagonally as

$$(-1)^F = (-1)^F_{00} \oplus (-1)^F_{11} \,. \qquad (\text{V.191})$$

The remaining $\mathbb{Z}_2^\eta$ symmetry can be read off form the $\eta$ action on $\mathcal{O}'_{e^a m^b}$. We know from the analysis of the $\mathbb{Z}_2 \times \mathbb{Z}_2^f$ charges that

$$\eta : (\mathcal{O}'_{m^2}, \mathcal{O}'_e) \longmapsto (-\mathcal{O}'_{m^2}, i\mathcal{O}'_e) \,, \qquad (\text{V.192})$$

which is all we require to compute the $\eta$ action on $\mathfrak{B}'$. $\eta$ maps between the decoupled boundaries $\mathfrak{B}'_0$ and $\mathfrak{B}'_1$ such that the different operators transform as

$$\eta : v'_0 \longleftrightarrow v'_1 \,,$$
$$: \mathcal{E}'_i \longleftrightarrow -\mathcal{E}'_i \,,$$
$$: (\mathcal{O}'_{m_0}, \mathcal{O}'_{m_1}) \longleftrightarrow (\mathcal{O}'_{m_1}, -\mathcal{O}'_{m_0}) \,, \qquad (\text{V.193})$$
$$: (\mathcal{O}'_{e_0}, \mathcal{O}'_{e_1}) \longleftrightarrow (-\mathcal{O}'_{e_1}, \mathcal{O}'_{e_0}) \,.$$

Note that $\eta^2$ is $+1$ on the operators in the NS sector and $-1$ on the operators in the R sector, therefore we deduce that $\eta^2 = \pi$ as expected. More precisely the $\eta$ symmetry is represented as

$$\eta = \pi_{01} \oplus 1_{10} \,, \qquad (\text{V.194})$$

with an additional minus sign ossociated with the intersection of the $\eta$ line and $\pi$ line.

$$\qquad (\text{V.195})$$

This gapless phase can be realized by IR CFT

$$\text{Maj} \oplus \text{Maj} \,. \qquad (\text{V.196})$$

and the two gapped phases are realized by either deforming both Maj to Triv or to Arf.

The Hasse diagram (V.164) of condensable algebras is realized as a diagram of phases for $\mathbb{Z}_2 \times \mathbb{Z}_2^f$ symmetry

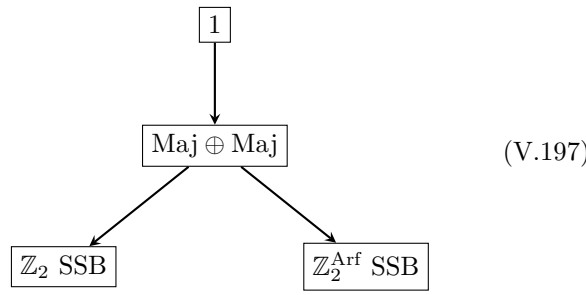

$$(\text{V.197})$$

which can also be realized via a fermionization of bosonic Hasse diagram

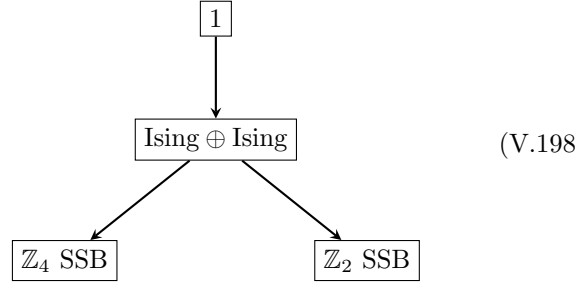

$$(\text{V.198})$$

**F.   $\mathbb{Z}_2 \times \mathbb{Z}_2^f$ with a beyond Gu-Wen Anomaly**

Consider the doubled Ising TFT $\mathcal{Z}(\mathsf{Ising}_+) = \mathsf{Ising} \boxtimes \overline{\mathsf{Ising}}$ as the SymTFT. Its anyon content is

$$
\begin{aligned}
1\bar{1}, \quad & 1\bar{\psi}, \quad 1\bar{\sigma}, \\
\psi\bar{1}, \quad & \psi\bar{\psi}, \quad \psi\bar{\sigma}, \\
\sigma\bar{1}, \quad & \sigma\bar{\psi}, \quad \sigma\bar{\sigma},
\end{aligned}
\tag{V.199}
$$

where $\{1, \psi, \sigma\}$ are the anyons of the Ising TFT and $\{\bar{1}, \bar{\psi}, \bar{\sigma}\}$ are the anyons of its time-reversal. There is a unique fermionic Lagrangian algebra

$$
\mathcal{A}^f_{\psi,\bar{\psi}} = 1\bar{1} \oplus 1\bar{\psi} \oplus \psi\bar{1} \oplus \psi\bar{\psi},
\tag{V.200}
$$

which we choose as the symmetry Lagrangian algebra

$$
\mathcal{A}_{\text{sym}} = \mathcal{A}^f_{\psi,\bar{\psi}}.
\tag{V.201}
$$

The symmetry category on the corresponding fermionic boundary is $\mathbb{Z}_2 \times \mathbb{Z}_2^f$ with a beyond Gu-Wen anomaly $\nu = 1, 7 \bmod 8$. As we discussed in Section III E 5, the bulk-to-boundary functor $F$ for this boundary is given by

$$
F(\psi\bar{1}) = F(1\bar{\psi}) = \pi, \quad F(\sigma\bar{1}) = q, \quad F(1\bar{\sigma}) = q(-1)^F,
\tag{V.202}
$$

where boundary line $q$ obeys the following fusion rules:

$$
q\pi = \pi q = q, \quad q^2 = 1 \oplus \pi.
\tag{V.203}
$$

The non-trivial bosonic condensable algebras in the doubled Ising TFT are

$$
\begin{aligned}
\mathcal{A}_{\sigma\bar{\sigma}} &= 1 \oplus \psi\bar{\psi} \oplus \sigma\bar{\sigma}, \\
\mathcal{A}_{\psi\bar{\psi}} &= 1 \oplus \psi\bar{\psi}.
\end{aligned}
\tag{V.204}
$$

The first one $\mathcal{A}_{\sigma\bar{\sigma}}$ is Lagrangian, while the second one $\mathcal{A}_{\psi\bar{\psi}}$ in non-Lagrangian. The Hasse diagram corresponding to these condensable algebras is

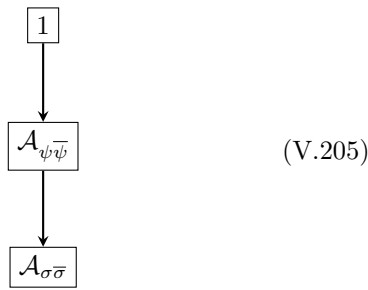

$$(\text{V.205})$$

**1.   SSB phase Triv ⊕ Arf**

Consider first the gapped phase with physical Lagrangian algebra

$$
\mathcal{A}_{\text{phys}} = \mathcal{A}_{\sigma\bar{\sigma}}.
\tag{V.206}
$$

This phase has two vacua because not only 1 but also $\psi\bar{\psi}$ can end on both boundaries. Topological point-like operators in this gapped phase are

$$
\mathcal{O}_{1\bar{1}}, \quad \mathcal{O}_{\psi\bar{\psi}}, \quad \mathcal{O}^0_{\sigma\bar{\sigma}}, \quad \mathcal{O}^1_{\sigma\bar{\sigma}},
\tag{V.207}
$$

where $\mathcal{O}_{1\bar{1}}$ and $\mathcal{O}_{\psi\bar{\psi}}$ are untwisted sector bosonic operators, $\mathcal{O}^0_{\sigma\bar{\sigma}}$ is a $(-1)^F$-twisted sector bosonic operator, and $\mathcal{O}^1_{\sigma\bar{\sigma}}$ is a $(-1)^F$-twisted sector fermionic operator. We can identify the two vacua to be

$$
\begin{aligned}
v_0 &= \frac{1 + \mathcal{O}_{\psi\bar{\psi}}}{2}, \\
v_1 &= \frac{1 - \mathcal{O}_{\psi\bar{\psi}}}{2},
\end{aligned}
\tag{V.208}
$$

which satisfy the condition

$$
v_i v_j = \delta_{ij} v_i.
\tag{V.209}
$$

The twisted sector operators $\mathcal{O}^0_{\sigma\bar{\sigma}}$ and $\mathcal{O}^1_{\sigma\bar{\sigma}}$ belong to different vacua because they carry the different fermion parity. Without loss of generality, we can take $\mathcal{O}^i_{\sigma\bar{\sigma}}$ to be a twisted sector operator in vacuum $v_i$ for $i = 0, 1$. The operator implementing the fermion parity symmetry $(-1)^F$ is identified as

$$
(-1)^F = 1_{00} \oplus \pi_{11},
\tag{V.210}
$$

or in other words, the underlying TFT $\mathfrak{T}^{\text{IR}}$ of this gapped phase is given by

$$
\mathfrak{T}^{\text{IR}} = \text{Triv} \oplus \text{Arf}.
\tag{V.211}
$$

The action of the line operator $q$ on the untwisted sector operators can be computed as

$$q: \ \mathcal{O}_{1\overline{1}} \to \sqrt{2}\mathcal{O}_{1\overline{1}}, \quad \mathcal{O}_{\psi\overline{\psi}} \to -\sqrt{2}\mathcal{O}_{\psi\overline{\psi}}, \qquad \text{(V.212)}$$

which implies that $q$ exchanges the two vacua as follows:

$$q: \ v_0 \to \sqrt{2}v_1, \quad v_1 \to \sqrt{2}v_0. \qquad \text{(V.213)}$$

This action allows us to conclude that the $q$ line is realized as

$$q = S_{01} \oplus S_{10}, \qquad \text{(V.214)}$$

where $S_{ij}$ is the interface between two vacua $v_i$ and $v_j$ that have a relative Arf term. Comparing the action (V.213) with eq. (V.10), we find that there is no relative Euler term between the two vacua. The gapped phase discussed above is the fermionization of the unique bosonic gapped phase with $\mathsf{Ising}_+$ symmetry.

### 2. $\psi\overline{\psi}$-condensed phase

Now let us discuss a gapless phase with $\mathbb{Z}_2 \times \mathbb{Z}_2^f$ symmetry with a beyond Gu-Wen anomaly $\nu = 1, 7 \bmod 8$. We take the physical condensable algebra to be $\mathcal{A}_{\psi\overline{\psi}}$

$$\mathcal{A}_{\text{phys}} = \mathcal{A}_{\psi\overline{\psi}} = 1 \oplus \psi\overline{\psi}. \qquad \text{(V.215)}$$

This defines a bosonic topological interface $\mathcal{I}_{\psi\overline{\psi}}$ from the doubled Ising TFT to the toric code. Thus, the club quiche compactification produces a topological boundary condition $\mathfrak{B}'$ of the toric code. As $\psi\overline{\psi}$ also appears in the symmetry Lagrangian algebra $\mathcal{A}_{\text{sym}} = \mathcal{A}_{\psi,\overline{\psi}}^f$, the boundary $\mathfrak{B}'$ has two topological local operators on it, which implies that $\mathfrak{B}'$ is comprised of two irreducible topological boundary conditions

$$\mathfrak{B}' = \mathfrak{B}'_0 \oplus \mathfrak{B}'_1. \qquad \text{(V.216)}$$

As we pass the interface $\mathcal{I}_{\psi\overline{\psi}}$, the anyons $\{1, e, m, f\}$ of the toric code are converted into the anyons of the doubled Ising TFT, according to the map

$$\begin{aligned} 1 &\mapsto 1 \oplus \psi\overline{\psi}, \\ e &\mapsto \sigma\overline{\sigma}, \\ m &\mapsto \sigma\overline{\sigma}, \\ f &\mapsto 1\overline{\psi} \oplus \psi\overline{1}. \end{aligned} \qquad \text{(V.217)}$$

From this and the Lagrangian algebra $\mathcal{A}_{\text{sym}} = \mathcal{A}_{\psi,\overline{\psi}}^f$, we learn that there are two fermionic topological ends of $f$ along $\mathfrak{B}'$, one coming from the end of $1\overline{\psi}$ and the other coming from the end of $\psi\overline{1}$. We label these ends respectively as $\mathcal{E}_{1\overline{\psi}}$ and $\mathcal{E}_{\psi\overline{1}}$. Furthermore, there are topological local operators $\mathcal{O}_{1\overline{1}} = 1$ and $\mathcal{O}_{\psi\overline{\psi}}$ along $\mathfrak{B}'$ coming from compactifying respectively $1\overline{1}$ and $\psi\overline{\psi}$ in between the symmetry boundary of $\mathcal{Z}(\mathsf{Ising}_+)$ and the interface

$\mathcal{I}_{\psi\overline{\psi}}$. The products of these operators obey the fusion of their subscripts:

$$\begin{aligned} \mathcal{E}_{\psi^i\overline{\psi}^{i+1}}\mathcal{E}_{\psi^j\overline{\psi}^{j+1}} &= \mathcal{O}_{\psi^{i+j}\overline{\psi}^{i+j}}, \\ \mathcal{E}_{\psi^i\overline{\psi}^{i+1}}\mathcal{O}_{\psi^j\overline{\psi}^j} &= \mathcal{E}_{\psi^{i+j}\overline{\psi}^{i+j+1}}, \\ \mathcal{O}_{\psi^i\overline{\psi}^i}\mathcal{O}_{\psi^j\overline{\psi}^j} &= \mathcal{O}_{\psi^{i+j}\overline{\psi}^{i+j}}. \end{aligned} \qquad \text{(V.218)}$$

The identity operators along the two boundaries $\mathfrak{B}'_0$ and $\mathfrak{B}'_1$ are

$$\begin{aligned} v_0 &= \frac{1 + \mathcal{O}_{\psi\overline{\psi}}}{2}, \\ v_1 &= \frac{1 - \mathcal{O}_{\psi\overline{\psi}}}{2}, \end{aligned} \qquad \text{(V.219)}$$

satisfying

$$v_i v_j = \delta_{ij} v_i. \qquad \text{(V.220)}$$

We also have fermionic topological ends

$$\begin{aligned} \mathcal{E}_0 &= \mathcal{E}_{1\overline{\psi}} + \mathcal{E}_{\psi\overline{1}}, \\ \mathcal{E}_1 &= \mathcal{E}_{1\overline{\psi}} - \mathcal{E}_{\psi\overline{1}} \end{aligned} \qquad \text{(V.221)}$$

of $f$ along $\mathfrak{B}'_0$ and $\mathfrak{B}'_1$ respectively, which satisfy

$$\mathcal{E}_i v_j = \delta_{ij} \mathcal{E}_i. \qquad \text{(V.222)}$$

This means that both $\mathfrak{B}'_0$ and $\mathfrak{B}'_1$ are irreducible fermionic topological boundaries of the toric code associated to the fermionic Lagrangian algebra $1 \oplus \pi f$.

In addition to the above local operators, the boundary $\mathfrak{B}'$ hosts two topological point-like operators that convert the $e$ line of the toric code into the fermion parity line $(-1)^F$ on the boundary. These operators come from the compactification of $\sigma\overline{\sigma}$ between the symmetry boundary of $\mathcal{Z}(\mathsf{Ising}_+)$ and the interface $\mathcal{I}_{\psi\overline{\psi}}$. The bulk-to-boundary map $F(\sigma\overline{\sigma}) = (-1)^F \oplus \pi(-1)^F$ implies that one of these operators is bosonic and the other is fermionic. Similarly, the boundary $\mathfrak{B}'$ also hosts two topological point-like operators that convert the $m$ line of the toric code into the fermion parity line $(-1)^F$ on the boundary. For the same reason as above, one of these operators is bosonic and the other is fermionic. This means that the two fermionic boundaries $\mathfrak{B}'_0$ and $\mathfrak{B}'_1$ have a relative Arf term. Thus, we find

$$\mathfrak{B}' = \mathfrak{B}_f \oplus (\mathfrak{B}_f \boxtimes \text{Arf}). \qquad \text{(V.223)}$$

Topological lines on the boundary $\mathfrak{B}'$ are listed as

$$\{1_{00}, 1_{11}, \pi_{00}, \pi_{11}, (-1)^F_{00}, (-1)^F_{11}, S_{01}, S_{10}\}, \qquad \text{(V.224)}$$

where $S_{01}$ and $S_{10}$ are interfaces between two boundaries $\mathfrak{B}'_0$ and $\mathfrak{B}'_1$. These interfaces obey the fusion rules

$$\begin{aligned} S_{01}S_{10} &= 1_{00} \oplus \pi_{00}, \\ S_{10}S_{01} &= 1_{11} \oplus \pi_{11}. \end{aligned} \qquad \text{(V.225)}$$

The action of the $q$ line on the point-like operators $v_i$ and $\mathcal{E}_i$ can be computed as

$$q: \begin{array}{ll} v_0 \to \sqrt{2}v_1, & v_1 \to \sqrt{2}v_0, \\ \mathcal{E}_0 \to \sqrt{2}\mathcal{E}_1, & \mathcal{E}_1 \to \sqrt{2}\mathcal{E}_0. \end{array} \qquad (V.226)$$

This implies that the line operator $q$ is realized on the boundary $\mathfrak{B}'$ as

$$q = S_{01} \oplus S_{10}. \qquad (V.227)$$

In particular, eq. (V.226) shows that there is no relative Euler term between the two boundaries $\mathfrak{B}'_0 = \mathfrak{B}_f$ and $\mathfrak{B}'_1 = \mathfrak{B}_f \boxtimes \mathrm{Arf}$. On the other hand, the fermion parity line $(-1)^F$ on $\mathfrak{B}'$ is realized as

$$(-1)^F = (-1)^F_{00} \oplus (-1)^F_{11}. \qquad (V.228)$$

Completing the club sandwich by a bosonic gapless physical boundary of the toric code, we obtain a gapless SSB phase for $\mathbb{Z}_2 \times \mathbb{Z}_2^f$ symmetry with a beyond Gu-Wen anomaly $\nu = 1, 7 \bmod 8$. The IR theory $\mathfrak{T}^{\mathrm{IR}}$ of this gSSB phase can be expressed as

$$\mathfrak{T}^{\mathrm{IR}} = \mathfrak{T}_f \oplus (\mathfrak{T}_f \boxtimes \mathrm{Arf}), \qquad (V.229)$$

where $\mathfrak{T}_f$ is a fermionic gapless phase with $\mathbb{Z}_2^f$ symmetry. The anomalous $\mathbb{Z}_2 \times \mathbb{Z}_2^f$ symmetry acts on $\mathfrak{T}^{\mathrm{IR}}$ via eqs. (V.227) and (V.228). To summarize, the Hasse diagram in (V.205) is realized concretely on phases with the beyond Gu-Wen anomaly as

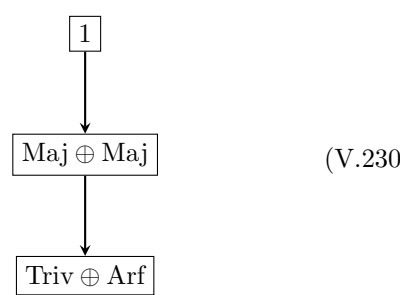

$$(V.230)$$

where we have simply chosen $\mathfrak{T}_f$ in (V.229) to be Maj. Finally, we can compare this with the analogous Hasse diagram for bosonic Ising symmetry, which corresponds to choosing the symmetry boundary to correspond to $\mathcal{A}_{\sigma\bar{\sigma}}$. The gapped phase obtained by choosing the physical boundary to be the unique Lagrangian algebra has three vacua and from the $\mathbb{Z}_2 \subset \mathcal{S}_{\mathrm{Ising}}$ is $\mathrm{Triv} \oplus \mathrm{SSB}$. The bosonic Hasse diagram has the form

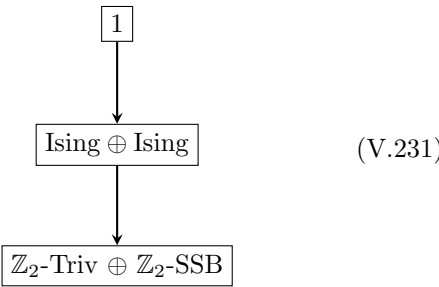

$$(V.231)$$

**Acknowledgements.** We thank Kantaro Ohmori, Matthew Yu and Yunqin Zheng for useful discussions. KI thanks Kantaro Ohmori for collaboration on a related project. We also thank Sheng-Jie Huang for coordinating the submission of the paper [170]. LB thanks Neils Bohr International Academy for hospitality, where a part of this work was completed. LB is funded as a Royal Society University Research Fellow through grant URF\R1\231467. KI was partially supported by FoPM, WINGS Program, the University of Tokyo, and by JSPS Research Fellowship for Young Scientists. AT is funded by Villum Fonden Grant no. VIL60714.

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
