# Peer review of "Fermionic Non-Invertible Symmetries in (1+1)d: Gapped and Gapless Phases, Transitions, and Symmetry TFTs"

_SciPost Physics_

## Round 2 · Referee Report · Anonymous (Referee 1) · 2024-10-24

Report
The paper is interesting, well-written and the exposition is clear. It also contains many explicit examples which will be useful for future studies. Therefore I recommend its publication in SciPost, once the minor comments below are addressed.
Requested changes
1) Throughout the paper, it is assumed that the fermionic parity symmetry is “non-anomalous.” It is clear from context what the authors mean, but it might be helpful to clarify explicitly whether they mean that the anomaly vanishes mod 8 or mod 16. When the anomaly vanishes only mod 8 but not mod 16 (e.g. 8 chiral Majoranas), it is known that there is an obstruction to bosonization, even though the fermion parity is “non-anomalous” in a conventional sense. For instance, see https://arxiv.org/pdf/2403.03953 and references therein.
2) On page 7, it is said that Equation (II.37) is known as the GSO projection. Although it is common to call (II.37) a GSO projection, strictly speaking it is different from the original GSO projection in superstring theory in general. In superstring (say Type II), one performs GSO projections independently for the left-moving and right-moving degrees of freedom, i.e. one gauges “chiral” fermion parities, and furthermore the result is not quite a bosonic theory (due to the previously mentioned obstruction to bosonization). It might be okay to still call Equation (II.37) a GSO projection informally, but the referee thinks that it would be helpful if this impreciseness in terminology is discussed at least briefly, maybe in a footnote.
3) On Page 22, the authors refer to a “canonical gapless phase” which is claimed to exist for any fermionic generalized symmetry. It would be helpful if it is explicitly explained how to construct such a canonical gapless phase for an arbitrary fusion supercategory symmetry.
Recommendation
Publish (surpasses expectations and criteria for this Journal; among top 10%)

---

## Round 2 · Referee Report · Anonymous (Referee 2) · 2024-11-1

Report
(1) The manuscript would benefit from additional references to the condensed matter physics literature, especially concerning the relationship between symTFT and gauged (2+1)D fermionic symmetry-protected topological (SPT) phases or symmetry-enriched topological phases. Over the past few decades, extensive work in condensed matter physics has addressed the classification and properties of these phases. For example, the super pentagon equation for the fermionic toric code, which first appeared in Phys. Rev. B 90, 085140 (2014), represents a specific instance of Ref. 180 in this manuscript. Citing other relevant references would help provide a more comprehensive context for readers.
(2) Is there a notion of quantum dimension for fermionic fusion supercategories? Including a discussion on this would be helpful in fermionization and bosonization, and in identifying fermionic Lagrangian algebras.
(3) What are the conditions or properties required for a fermionic fusion supercategory to take the form of a Deligne product of
sVec with some bosonic fusion category?
(4) The symTFT of beyond Gu-Wen anomaly case can also be constructed as a model of commuting projector Hamiltonian of Majorana chain-decorated fermionic SPT phases, first constructed in Phys. Rev. B 94, 115115 (2016). A superfusion category likely underlies this construction. Could the authors discuss the relationship between this category and the one presented in the manuscript?
(5) Could the authors elaborate on the relation of fermionization discussed here to the approach in JHEP 2017, 172 (2017)?
(6) The manuscript states, "In this paper, we suppose that the fermion parity symmetry generated by (-1)^F is non-anomalous." Are there scenarios in physical systems where this symmetry could indeed be anomalous? Further discussion of this possibility and its implications could be beneficial for readers.
Recommendation
Publish (surpasses expectations and criteria for this Journal; among top 10%)

---

## Editorial Decision

unknown